# DEEP DISTRIBUTED OPTIMIZATION FOR LARGE-SCALE QUADRATIC PROGRAMMING

**Augustinos D. Saravanos, Hunter Kuperman, Alex Oshin, Arshiya Taj Abdul, Vincent Pacelli and Evangelos A. Theodorou**
Georgia Institute of Technology, USA
{asaravanos, kup, alexoshin, aabdul6, vpacelli3, evangelos.theodorou}@gatech.edu

## ABSTRACT

Quadratic programming (QP) forms a crucial foundation in optimization, appearing in a broad spectrum of domains and serving as the basis for more advanced algorithms. Consequently, as the scale and complexity of modern applications continue to grow, the development of efficient and reliable QP algorithms becomes increasingly vital. In this context, this paper introduces a novel deep learning-aided distributed optimization architecture designed for tackling large-scale QP problems. First, we combine the state-of-the-art Operator Splitting QP (OSQP) method with a consensus approach to derive **DistributedQP**, a new method tailored for network-structured problems, with convergence guarantees to optimality. Subsequently, we unfold this optimizer into a deep learning framework, leading to **DeepDistributedQP**, which leverages learned policies to accelerate reaching to desired accuracy within a restricted amount of iterations. Our approach is also theoretically grounded through Probably Approximately Correct (PAC)-Bayes theory, providing generalization bounds on the expected optimality gap for unseen problems. The proposed framework, as well as its centralized version **DeepQP**, significantly outperform their standard optimization counterparts on a variety of tasks such as randomly generated problems, optimal control, linear regression, transportation networks and others. Notably, DeepDistributedQP demonstrates strong generalization by training on small problems and scaling to solve much larger ones (up to 50K variables and 150K constraints) using the same policy. Moreover, it achieves orders-of-magnitude improvements in wall-clock time compared to OSQP. The certifiable performance guarantees of our approach are also demonstrated, ensuring higher-quality solutions over traditional optimizers.

## 1 INTRODUCTION

Quadratic programming (QP) serves as a fundamental cornerstone in optimization with a wide variety of applications in machine learning (Cortes & Vapnik, 1995; Tibshirani, 1996), control and robotics (Garcia et al., 1989; Rawlings et al., 2017), signal processing (Mattingley & Boyd, 2010), finance (Cornuejols et al., 2018), and transportation networks (Mota et al., 2014) among other fields. Beyond its standalone applications, QP also acts as the core component of many advanced non-convex optimization algorithms such as sequential quadratic programming (Nocedal & Wright, 1999), trust-region methods (Conn et al., 2000), augmented Lagrangian approaches (Houska et al., 2016), mixed-integer optimization (Belotti et al., 2013), etc. For these reasons, the pursuit of more efficient QP algorithms remains an ever-evolving area of research from active set (Wolfe, 1959) and interior point methods (Nesterov & Nemirovskii, 1994) during the previous century to first-order methods such as the state-of-the-art Operator Splitting QP (OSQP) algorithm (Stellato et al., 2020).

As the scale of modern decision-making applications rapidly increases, there is an emerging interest in developing effective optimization architectures for addressing high-dimensional problems. Given the fundamental role of QP in optimization, there is a clear demand for algorithms capable of solving large-scale QPs with thousands, and potentially much more, variables and constraints. Such problems arise in diverse applications including sparse linear regression (Mateos et al., 2010) and support vector machines (Navia-Vazquez et al., 2006) with decentralized data, multi-agent control (Van Parys & Pipeleers, 2017), resource allocation (Huang et al., 2014), network flow (Mota et al., 2014), power grids (Lin et al., 2012) and image processing (Soheili & Eftekhari-Moghadam, 2020). Traditional centralized optimization algorithms are inadequate for solving such problems at

scale (see for example Fig. 1), prompting the development of distributed methods that leverage the underlying network/decentralized structure to parallelize computations. In this context, the Alternating Direction Method of Multipliers (ADMM) has gained widespread popularity as an effective approach for deriving distributed algorithms (Boyd et al., 2011; Mota et al., 2013). Nevertheless, as scale increases, such algorithms continue to face significant challenges such as their need for *meticulous tuning*, the absence of *generalization guarantees* and restrictions on the *allowed number of iterations* imposed by computational or communication limitations.

Learning-to-optimize has recently emerged as a methodology for enhancing existing optimizers or developing entirely new ones through training on sample problems (Chen et al., 2022b; Amos et al., 2023). A notable approach within this paradigm is *deep unfolding*, which unrolls optimizer iterations as layers of a deep learning network and learns the optimal parameters for improving performance (Monga et al., 2021; Shlezinger et al., 2022). Our key insight is that deep unfolding is particularly well-suited for overcoming the limitations of *distributed constrained optimization*, as it can eliminate the need for extensive tuning, manage iteration restrictions and enhance generalization. However, its combination with distributed ADMM has only recently been explored in Noah & Shlezinger

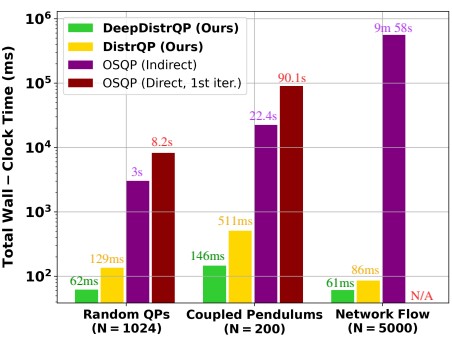

Figure 1: **Wall-clock time comparison:** DeepDistributedQP, DistributedQP (ours) and OSQP on large-scale QPs.

(2024). While this framework shows promising initial results, it relies on a relatively simple setup that studies unconstrained problems, assumes local updates consisting of gradient steps, focuses solely on parameter tuning, and is not accompanied by any formal performance guarantees.

**Contributions.** This paper introduces a novel deep learning-aided distributed optimization architecture for solving large-scale constrained QP problems. Our approach relies on unfolding a newly introduced distributed QP algorithm as a supervised learning framework for a prescribed number of iterations. To our best knowledge, this is the first work to present a deep learning architecture for distributed constrained optimization using ADMM, despite the widespread popularity of the latter. Our framework demonstrates remarkable scalability, being trained on small problems and then effectively applied to much larger ones. Furthermore, its performance is theoretically supported by establishing guarantees based on generalization bounds from statistical learning theory. We believe that this work lays the foundation for developing learned distributed optimizers capable of handling large-scale constrained optimization problems without requiring training at such scales.

Our specific contributions can be summarized as follows:

- First, we introduce **DistributedQP**, a new decentralized method that combines the well-established OSQP solver with a consensus approach. We further prove that the algorithm is guaranteed to converge to optimality, even under varying local algorithm parameters.

- Then, we propose **DeepDistributedQP**, a deep learning-aided distributed architecture that unrolls the iterations of DistributedQP in a supervised manner, learning feedback policies for the underlying algorithm parameters. As a byproduct, we also present **DeepQP**, its centralized counterpart which corresponds to unfolding the standard OSQP solver.

- To certify the performance of the learned solver, we establish generalization guarantees on the optimality gap of the final solution of DeepDistributedQP for unseen problems using Probably Approximately Correct (PAC)-Bayes theory.

- Finally, we present an extensive experimental evaluation that validates the following:
  - For centralized QPs, DeepQP consistently outperforms OSQP requiring 1.5-3 times fewer iterations for achieving the desired accuracy.
  - DeepDistributedQP successfully scales for *high-dimensional* problems (up to 50K variables and 150K constraints) while being *trained exclusively* on *much smaller ones*. Furthermore, both DeepDistributedQP and DistributedQP outperform OSQP in wall-clock time by orders of magnitude as the problem dimensionality increases.
  - The proposed PAC bounds offer valuable guarantees on the quality of solutions produced by DeepDistributedQP for unseen problems from the same class.

## 2 RELATED WORK

This section provides an overview of existing related literature from the angles of distributed optimization and learning-to-optimize. An extended discussion is provided in Appendix A.

**Distributed optimization with ADMM.** Distributed ADMM algorithms have emerged as a scalable approach for addressing large-scale optimization problems (Boyd et al., 2011; Mota et al., 2013). Despite their significant applicability to machine learning (Mateos et al., 2010), robotics (Shorinwa et al., 2024) and many other fields, their successful performance has been shown to be highly sensitive to the proper tuning of underlying parameters (Xu et al., 2017a; Saravanos et al., 2023a). Moreover, tuning parameters for large-scale problems is often tedious and time-consuming, making it desirable to develop effective *learned* optimizers that can be trained on smaller problems instead. Furthermore, even if an distributed optimizer performs well for a specific problem instance, its generalization to new problems remains challenging to verify. These challenges constitute our main motivation for studying learning-aided distributed ADMM architectures. We also note that an ADMM-based distributed QP solver resembling a simpler version of DistributedQP was presented in Pereira et al. (2022), but focusing on multi-robot control and lacking any theoretical analysis.

**Learning-to-optimize.** The area of learning-to-optimize methods has emerged as an effective approach for enhancing existing optimizers or even deriving new algorithmic updates through training on sample problems (Chen et al., 2022a; Shlezinger et al., 2022; Amos et al., 2023). A prominent technique in this paradigm is deep unfolding, which under the realistic assumption of computational budget restrictions, unrolls a fixed number of iterations as layers of a deep learning framework and learns the optimal parameters for improving performance on a specific problem class (Monga et al., 2021; Zhang et al., 2020). Nevertheless, combining deep unfolding with distributed ADMM has only been investigated recently in Noah & Shlezinger (2024). Although this framework demonstrates promising results, it is limited to an unconstrained problem formulation, assumes gradient-based local updates, focuses exclusively on parameter tuning and lacks formal performance guarantees. A reinforcement learning algorithm for accelerating OSQP was presented in Ichnowski et al. (2021). While this approach also explores learning policies for algorithm parameters, it is limited to centralized quadratic programming, lacks guarantees and its training comes at a significant computational cost. In the context of establishing generalization bounds for learned optimizers, Sambharya & Stellato (2024a) recently explored incorporating PAC-Bayes bounds in learned optimizers, yet our approach differs fundamentally, as their method employs a binary error function, whereas ours directly establishes bounds based on the optimality gap of the final solution. The works in Sucker & Ochs (2023) and Sucker et al. (2024) are also investigating generalization bounds for learned optimizers, considering the update function as a gradient step or a multi-layer perceptron, respectively.

## 3 DISTRIBUTED QUADRATIC PROGRAMMING

### 3.1 PROBLEM FORMULATION

A convex (centralized) QP problem is expressed in general as

$$\min \frac{1}{2}\boldsymbol{x}^\top \boldsymbol{Q}\boldsymbol{x} + \boldsymbol{q}^\top \boldsymbol{x} \quad \text{s.t.} \quad \boldsymbol{A}\boldsymbol{x} \le \boldsymbol{b}, \quad (1)$$

where $\boldsymbol{x} \in \mathbb{R}^n$ is the decision vector and $\zeta = \{\boldsymbol{Q} \in \mathbb{S}_{++}^n, \boldsymbol{q} \in \mathbb{R}^n, \boldsymbol{A} \in \mathbb{R}^{m \times n}, \boldsymbol{b} \in \mathbb{R}^m\}$ are the problem data. [1] As the scale of such problems increases to higher dimensions, there is often an underlying networked/decentralized structure that could be

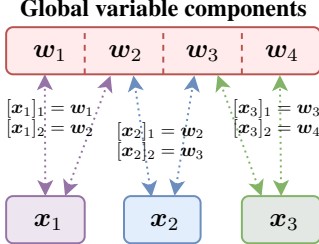

**Global variable components**

**Local variables**

Figure 2: Example of consensus mapping $\mathcal{G}$ in problem (2).

leveraged for achieving distributed computations. This work specifically aims to address problems characterized by such structures. Let $\boldsymbol{w} \in \mathbb{R}^n$ be the main global variable and $\boldsymbol{x}_i \in \mathbb{R}^{n_i}$ be local variables $i \in \mathcal{V} = \{1, \ldots, N\}$. Then, assume a mapping $(i, j) \mapsto \mathcal{G}(i, j)$ from all index pairs $(i, j)$ of local variable components $[\boldsymbol{x}_i]_j$ to indices $l = \mathcal{G}(i, j)$ of global components $\boldsymbol{w}_l$[2] - for an example see Fig. 2. We consider QP problems of the following *distributed consensus* form:

$$\min \sum_{i \in \mathcal{V}} \frac{1}{2}\boldsymbol{x}_i^\top \boldsymbol{Q}_i \boldsymbol{x}_i + \boldsymbol{q}_i^\top \boldsymbol{x}_i \quad \text{s.t.} \quad \boldsymbol{A}_i \boldsymbol{x}_i \le \boldsymbol{b}_i, \quad \boldsymbol{x}_i = \tilde{\boldsymbol{w}}_i, \quad i \in \mathcal{V}, \quad (2)$$

---

[1] Note that equality constraints can also be captured as pairs of inequalities.

[2] This formulation is adopted from the standard consensus ADMM framework (Boyd et al., 2011), wherein local variables are typically associated with their respective computational nodes.

where the problem data are now given by $\zeta = \{\zeta_i\}_{i=1}^N$ with $\zeta_i = \{\boldsymbol{Q}_i \in \mathbb{S}_{++}^{n_i}, \boldsymbol{q}_i \in \mathbb{R}^{n_i}, \boldsymbol{A}_i \in \mathbb{R}^{m_i \times n_i}, \boldsymbol{b}_i \in \mathbb{R}^{m_i}\}$. The vector $\boldsymbol{x} = [\{\boldsymbol{x}_i\}_{i \in \mathcal{V}}]$ is the concatenation of all local variables, while $\tilde{\boldsymbol{w}}_i \in \mathbb{R}^{n_i}$, defined as $\tilde{\boldsymbol{w}}_i = [\{\boldsymbol{w}_l\}_{l \in \mathcal{G}(q,j):q=i}]$, is the selection of global variable components that correspond to the components of $\boldsymbol{x}_i$. This form captures a wide variety of large-scale QPs found in machine learning (Mateos et al., 2010; Navia-Vazquez et al., 2006), optimal control (Van Parys & Pipeleers, 2017), transportation networks, (Mota et al., 2014), power grids (Lin et al., 2012), resource allocation (Huang et al., 2014) and many other fields.

## 3.2 DISTRIBUTEDQP: THE UNDERLYING OPTIMIZATION ALGORITHM

This section introduces a new distributed algorithm named **DistributedQP** for solving problems of the form (2). The proposed method can be viewed as a combination of consensus ADMM (Boyd et al., 2011) and OSQP (Stellato et al., 2020) using local iteration-varying penalty parameters.

Let us introduce the auxiliary variables $\boldsymbol{z}_i, \boldsymbol{s}_i \in \mathbb{R}^{m_i}$, such that problem (2) can be reformulated as

$$\min \sum_{i \in \mathcal{V}} \frac{1}{2} \boldsymbol{x}_i^\top \boldsymbol{Q}_i \boldsymbol{x}_i + \boldsymbol{q}_i^\top \boldsymbol{x}_i \quad \text{s.t.} \quad \boldsymbol{A}_i \boldsymbol{x}_i = \boldsymbol{z}_i, \ \boldsymbol{s}_i \le \boldsymbol{b}_i, \ \boldsymbol{z}_i = \boldsymbol{s}_i, \ \boldsymbol{x}_i = \tilde{\boldsymbol{w}}_i, \ i \in \mathcal{V}. \quad (3)$$

The proposed DistributedQP method is summarized below, where $k = 0, 1, \dots,$ denotes iterations:

1. **Local updates for $\boldsymbol{x}_i, \boldsymbol{z}_i$.** For each node $i \in \mathcal{V}$, solve in parallel:
$$\begin{bmatrix} \boldsymbol{Q}_i + \mu_i^k \boldsymbol{I} & \boldsymbol{A}_i^\top \\ \boldsymbol{A}_i & -1/\rho_i^k \boldsymbol{I} \end{bmatrix} \begin{bmatrix} \boldsymbol{x}_i^{k+1} \\ \boldsymbol{\nu}_i^{k+1} \end{bmatrix} = \begin{bmatrix} -\boldsymbol{q}_i + \mu_i^k \tilde{\boldsymbol{w}}_i - \boldsymbol{y}_i \\ \boldsymbol{z}_i - 1/\rho_i^k \boldsymbol{\lambda}_i \end{bmatrix}, \quad (4)$$
and then update in parallel:
$$\boldsymbol{z}_i^{k+1} = \boldsymbol{s}_i^k + 1/\rho_i^k (\boldsymbol{\nu}_i^{k+1} - \boldsymbol{\lambda}_i^k). \quad (5)$$

2. **Local updates for $\boldsymbol{s}_i$ and global update for $\boldsymbol{w}$.** For each node $i \in \mathcal{V}$, update in parallel:
$$\boldsymbol{s}_i^{k+1} = \Pi_{\boldsymbol{s}_i \le \boldsymbol{b}_i} \left( \alpha^k \boldsymbol{z}_i^{k+1} + (1 - \alpha^k) \boldsymbol{s}_i^k + \boldsymbol{\lambda}_i^k / \rho_i^k \right). \quad (6)$$
In addition, each global variable component $\boldsymbol{w}_l$ is updated through:
$$\boldsymbol{w}_l^{k+1} = \alpha^k \frac{\sum_{\mathcal{G}(i,j)=l} \mu_i^k [\boldsymbol{x}_i]_j}{\sum_{\mathcal{G}(i,j)=l} \mu_i^k} + (1 - \alpha^k) \boldsymbol{w}_l^k. \quad (7)$$

3. **Local updates for dual variables $\boldsymbol{\lambda}_i, \boldsymbol{y}_i$.** For each node $i \in \mathcal{V}$, update in parallel:
$$\boldsymbol{\lambda}_i^{k+1} = \boldsymbol{\lambda}_i^k + \rho_i^k (\alpha^k \boldsymbol{z}_i^{k+1} + (1 - \alpha^k) \boldsymbol{s}_i^k - \boldsymbol{s}_i^{k+1}), \quad (8)$$
$$\boldsymbol{y}_i^{k+1} = \boldsymbol{y}_i^k + \mu_i^k (\alpha^k \boldsymbol{x}_i^{k+1} + (1 - \alpha^k) \tilde{\boldsymbol{w}}_i^k - \tilde{\boldsymbol{w}}_i^{k+1}). \quad (9)$$

The Lagrange multipliers $\boldsymbol{\nu}_i, \boldsymbol{\lambda}_i$ and $\boldsymbol{y}_i$ correspond to the equality constraints $\boldsymbol{A}_i \boldsymbol{x}_i = \boldsymbol{z}_i, \boldsymbol{z}_i = \boldsymbol{s}_i$ and $\boldsymbol{x}_i = \tilde{\boldsymbol{w}}_i$, respectively. The penalty parameters $\rho_i, \mu_i > 0$ correspond to $\boldsymbol{z}_i = \boldsymbol{s}_i$ and $\boldsymbol{x}_i = \tilde{\boldsymbol{w}}_i$, while $\alpha^k \in [1, 2)$ are over-relaxation parameters. A complete derivation is provided in Appendix B.

## 3.3 CONVERGENCE GUARANTEES

Prior to unrolling DistributedQP into a deep learning framework, it is particularly important to establish that the underlying optimization algorithm is well-behaved even for varying parameters, i.e., it is expected to asymptotically converge to the optimal solution. This property is especially important in deep unfolding where parameters are expected to be distinct between different iterations.

In the simpler case of $\alpha^k = 1$, $\rho_i^k = \rho$, $\mu_i^k = \mu$, the standard convergence guarantees of two-block ADMM would apply directly (Deng & Yin, 2016); for a detailed discussion, see Appendix C. Nevertheless, the introduction of local iteration-varying penalty parameters $\rho_i^k, \mu_i^k$, as well as the over-relaxation with varying parameters $\alpha^k$ makes proving the convergence of this algorithm non-trivial. In the following, we provide convergence guarantees to optimality for DistributedQP.

We consider the following assumption for the penalty parameters.

**Assumption 1.** *As $k \to \infty$, the parameters $\rho_i^k = \rho_i^{k-1}$, $\mu_i^k = \mu_i^{k-1}$, for all $i \in \mathcal{V}$.*

The following theorem states the convergence guarantees of DistributedQP to optimality. The proof, as well as necessary intermediate results, are provided in Appendix D.

**Theorem 1** (Convergence guarantees for DistributedQP). *If Assumption 1 holds and $\alpha^k \in [1, 2)$, then the iterates $\boldsymbol{w}^k$ converge to the optimal solution $\boldsymbol{w}^*$ of problem (2), as $k \to \infty$.*

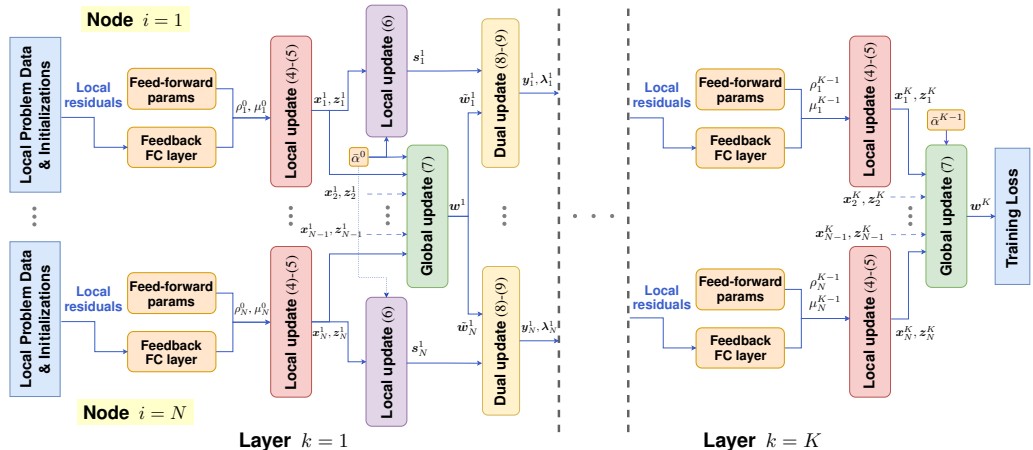

Figure 3: **The DeepDistributedQP architecture.** The proposed framework relies on unrolling the DistributedQP optimizer as a supervised deep learning framework. In particular, we interpret its iterations (4)-(9) as sequential network layers and introduce learnable components (orange blocks) to facilitate reaching the desired accuracy after a predefined number of allowed iterations.

## 4    THE DEEPDISTRIBUTEDQP ARCHITECTURE

The proposed DeepDistributedQP architecture emerges from unfolding the iterations of the DistributedQP optimizer into a deep learning framework. Section 4.1 illustrates the main architecture, key aspects of our methodology, as well as the centralized variant DeepQP. Section 4.2 leverages implicit differentiation during backpropagation to facilitate the training of our framework.

### 4.1    MAIN ARCHITECTURE

**Architecture overview.**    The **DeepDistributedQP** architecture arises from unrolling the DistributedQP optimizer within the supervised learning paradigm. (Fig. 3). This is accomplished through treating the updates (4)-(9) as blocks in sequential layers of a deep learning network. The number of layers is equal to the predefined number of allowed iterations $K$, with each layer corresponding to an iteration $k = 1, \ldots, K$. The inputs of the network are the local problem data $\zeta_i$ and initializations $x_i^0, z_i^0, \tilde{w}_i^0, s_i^0, \lambda_i^0$ and $y_i^0$. These are initially passed to $N$ parallel local blocks corresponding to (4)-(5), which output the new variables $x_i^1$ and $z_i^1$. Then, all $z_i^1$ are fed into $N$ new parallel local blocks (6), yielding the new iterates $s_i^1$. In the meantime, all $x_i^1$ are communicated to a central node that computes the new iterate $w^1$ through the weighted averaging step (7). Subsequently, the global variable components $\tilde{w}_i$ are communicated back to each local node $i$, to perform the updates (8)-(9) which output the updated dual variables $\lambda_i, y_i$. This group of blocks is then repeated $K$ times, yielding the output of the network which is the final global variable iterate $w^K$.

**Learning feedback policies.**    Standard deep unfolding typically leverages data to learn algorithm parameters tailored for a specific problem (Shlezinger et al., 2022). From a control theoretic point of view, this process can be interpreted as seeking *open-loop* policies without the incorporating any feedback. In our setup, this is equivalent with learning the optimal parameters $\bar{\rho}_i^k$, $\bar{\mu}_i^k$, $\bar{\alpha}^k$, with

$$\rho_i^k = \text{SoftPlus}(\bar{\rho}_i^k), \quad \mu_i^k = \text{SoftPlus}(\bar{\mu}_i^k), \quad \alpha^k = \text{Sigmoid}_{1,2}(\bar{\alpha}^k), \tag{10}$$

for all $i = 1, \ldots, N$ and $k = 1, \ldots, K$, where the $\text{SoftPlus}(\cdot)$ function is used to guarantee the positivity of $\rho_i^k, \mu_i^k$, and the sigmoid function $\text{Sigmoid}_{1,2}(\cdot)$ restricts each $\alpha^k$ to lie between $(1, 2)$.

In the meantime, the predominant practice for online adaptation of the ADMM penalty parameters relies on observing the primal and dual residuals every few iterations (Boyd et al., 2011). The widely-used rule suggests that if the ratio of primal-to-dual residuals is high, the penalty parameter $\rho$ should be increased; conversely, if the ratio is low, $\rho$ should be decreased. Despite its heuristic nature, this approach includes a notion of "feedback" since the current state of the optimizer is used to adapt the parameters, and as a result, it can be interpreted as a closed-loop policy. Based on this point of view, our goal is to learn the optimal *closed-loop* policies for the local penalty parameters

$$\rho_i^k = \text{SoftPlus}\Big(\bar{\rho}_i^k + \pi_{i,\rho}^k(r_{i,\rho}^k, s_{i,\rho}^k; \theta_{i,\rho}^k)\Big), \quad \mu_i^k = \text{SoftPlus}\Big(\bar{\mu}_i^k + \pi_{i,\mu}^k(r_{i,\mu}^k, s_{i,\mu}^k; \theta_{i,\mu}^k)\Big), \tag{11}$$

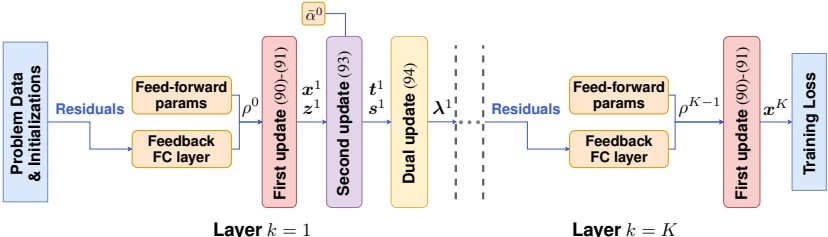

Figure 4: **The DeepQP architecture:** The simplified variant for centralized QP.

where the feedback components are obtained through the policies $\pi_{i,\cdot}^k(r_{i,\cdot}^k, s_{i,\cdot}^k; \theta_{i,\cdot}^k)$, parameterized by fully-connected neural network layers with inputs $r_{i,\cdot}^k, s_{i,\cdot}^k$ and weights $\theta_{i,\cdot}^k$. The terms $r_{i,\cdot}^k$ and $s_{i,\cdot}^k$ represent the local primal and dual residuals of node $i$ at layer $k$ and are detailed in Appendix E.

**Solving the local updates.** The most computationally demanding block in DeepDistributedQP is solving the local updates (4), as this requires solving a linear system of size $n_i + m_i$. Similar to OSQP (Stellato et al., 2020), this can be accomplished using either a direct or an indirect method. The direct method factors the KKT matrix, solving the system via forward and backward substitution. This approach is particularly efficient when penalty parameters remain fixed, as the same factorization can then be reused accross iterations. Nevertheless, at larger scales, this factorization might become impractical. In contrast, with the indirect method, we eliminate $\nu_i^{k+1}$ to solve the linear system:

$$\underbrace{(\boldsymbol{Q}_i + \mu_i^k \boldsymbol{I} + \boldsymbol{A}_i^\top \rho_i^k \boldsymbol{A}_i)}_{\bar{\boldsymbol{Q}}_i^k} \boldsymbol{x}_i^{k+1} = \underbrace{-\boldsymbol{q}_i + \mu_i^k \tilde{\boldsymbol{w}}_i - \boldsymbol{y}_i + \boldsymbol{A}_i^\top \rho_i^k \boldsymbol{z}_i - \boldsymbol{A}_i^\top \boldsymbol{\lambda}_i}_{\bar{\boldsymbol{b}}_i^k}. \tag{12}$$

This new linear system is solved for $\boldsymbol{x}_i^{k+1}$ using an iterative scheme such as the conjugate gradient (CG) method. We then substitute $\nu_i^{k+1} = \rho_i^k(\boldsymbol{A}_i \boldsymbol{x}_i^{k+1} - \boldsymbol{z}_i) + \boldsymbol{\lambda}_i$. The indirect method has three important properties that make it particularly attractive in our setup. First, its computational complexity scales better w.r.t. the dimension of the local problem, while no additional overhead is introduced by changing the penalty parameters. Second, it can be warmstarted using the solution from the previous iteration, greatly reducing the number of iterations required to converge to a solution. The final important property, which is critical for the scalability of the DeepDistributedQP, is that training with the indirect method can be much more memory efficient as shown in Section 4.2.

**Training loss.** Let $\mathcal{S} = \{\zeta^j\}_{j=1}^H$ be a dataset consisting of $H$ problem instances $\zeta^j = \{(\boldsymbol{Q}_i, \boldsymbol{q}_i, \boldsymbol{A}_i, \boldsymbol{b}_i)_{i=1}^N, \boldsymbol{w}^*\}_j$ subject to the known mapping $\mathcal{G}$ of problem (2). The loss we are using for training is the average of the $\gamma_k$-scaled distances of the global iterates $\boldsymbol{w}_1, \ldots, \boldsymbol{w}_N$ from the known optimal solution $\boldsymbol{w}^*$ of each problem instance $\zeta^j$, provided as

$$\ell(\mathcal{S}; \theta) = \frac{1}{H} \sum_{j=1}^H \sum_{k=1}^K \gamma_k \|\boldsymbol{w}^k(\zeta^j; \theta) - \boldsymbol{w}^*(\zeta^j)\|_2, \tag{13}$$

where $\theta$ corresponds to the concatenation of all learnable parameters/weights.

**Centralized version.** While this work primarily focuses on distributed optimization, we also introduce **DeepQP**, the centralized version of our framework, for addressing general QPs of the form (1). In the centralized case, our architecture simplifies to $N = 1$, eliminating the need for distinguishing between local and global variables. Under this simplification, the DistributedQP optimizer coincides with OSQP. Hence, DeepQP consists of unfolding the OSQP updates (see Appendix F) and learning policies for adapting its penalty and over-relaxation parameters. The resulting framework is illustrated in Fig. 4. Additional details on DeepQP are provided in Appendix F.

### 4.2 IMPLICIT DIFFERENTIATION

When solving for the local updates in (12) using the indirect method, it is computationally intractable to backpropagate through all CG iterations. This is especially important in the context of unfolding, as it would become necessary to unroll multiple inner CG optimization loops. To address this, we leverage the implicit function theorem (IFT) to express the solution of (12) as an implicit function

of the local problem data. This allows us to compute gradients in a manner that avoids unrolling the CG iterations and requires solving a linear system with the same coefficient matrix, but with a new RHS, achieved by rerunning the CG method. This result is formalized in the following theorem.

**Theorem 2** (Implicit Differentiation of Indirect Method). *Let $\boldsymbol{x}_i^{k+1}$ be the unique solution to the linear system $\bar{\boldsymbol{Q}}_i^k \boldsymbol{x}_i^{k+1} = \bar{\boldsymbol{b}}_i^k$ in (12). Let $\nabla_{\boldsymbol{x}} L(\boldsymbol{x}_i^{k+1})$ be a backward pass vector computed through reverse-mode automatic differentiation of some loss function $L$. Then, the gradient of $L$ with respect to $\bar{\boldsymbol{Q}}_i^k$ and $\bar{\boldsymbol{b}}_i^k$ is given by*

$$\nabla_{\bar{\boldsymbol{Q}}_i^k} L = \frac{1}{2}(\boldsymbol{x}_i^{k+1} \otimes d\boldsymbol{x}_i^{k+1} + d\boldsymbol{x}_i^{k+1} \otimes \boldsymbol{x}_i^{k+1}),$$

$$\nabla_{\bar{\boldsymbol{b}}_i^k} L = -d\boldsymbol{x}_i^{k+1},$$

*where $d\boldsymbol{x}_i^{k+1}$ is the unique solution to the linear system $\bar{\boldsymbol{Q}}_i^k d\boldsymbol{x}_i^{k+1} = -\nabla_x L(\boldsymbol{x}_i^{k+1})$.*

The proof is provided in Appendix G and is a straightforward application of the IFT, similar to the results established by Amos & Kolter (2017) and Agrawal et al. (2019).

## 5 GENERALIZATION BOUNDS

In this section, we establish guarantees on the expected performance of DeepDistributedQP. To achieve this, we leverage the PAC-Bayes framework (Alquier, 2024), a well-known statistical learning methodology for providing bounds on expected loss metrics that hold with high probability. In our case, we provide bounds on the *expected progress* of the final iterate $\boldsymbol{w}^K$ towards reaching the optimal solution $\boldsymbol{w}^*$ for unseen problems drawn from the same distribution as the training dataset.

**Learning stochastic policies.** PAC-Bayes theory is applicable to frameworks that learn weight distributions rather than fixed weights. For this reason, in order to establish such guarantees, we switch to learning a Gaussian distribution of weights $\mathcal{P} = \mathcal{N}(\mu_\Theta, \Sigma_\Theta)$ based on a prior $\mathcal{P}_0 = \mathcal{N}(\mu_\Theta^0, \Sigma_\Theta^0)$. This choice is motivated by the fact that PAC-Bayes bounds include Kullback–Leibler (KL) divergence terms which can be easily evaluated and optimized for Gaussian distributions.

**Generalization bound for DeepDistributedQP.** To facilitate the exhibition of our performance guarantees, we provide necessary preliminaries on PAC-Bayes theory in Appendix H. To establish a generalization guarantee for DeepDistributedQP, a meaningful loss function must first be selected. This quantity will be denoted $q(\zeta; \theta)$ to differentiate from the loss used for training. To capture the progress the optimizer makes towards optimality, we propose the following *progress metric*:

$$q(\zeta; \theta) = \min \left\{ \frac{\|\boldsymbol{w}^K(\zeta; \theta) - \boldsymbol{w}^*(\zeta)\|_2}{\|\boldsymbol{w}^0(\zeta) - \boldsymbol{w}^*(\zeta)\|_2}, 1 \right\}. \tag{14}$$

This loss function measures progress by comparing the distance between the final iterate $\boldsymbol{w}^K(\zeta; \theta)$ and problem solution $\boldsymbol{w}^*(\zeta)$ with the distance between the initialization $\boldsymbol{w}^0(\zeta; \theta)$ and the solution. This choice satisfies the requirement of being bounded between 0 and 1 while being more informative than the indicator losses used in prior work that simply determine whether the final iterate is within a specified neighborhood of the optimal solution (Sambharya & Stellato, 2024a). Moreover, this loss is invariant to the scale of the problem data since it is a relative measurement.

As in Appendix H, let $q_{\mathcal{D}}(\mathcal{P})$ be the true expected loss and $q_{\mathcal{S}}(\mathcal{P})$ the empirical expected loss. To evaluate the PAC-Bayes bounds in (101), the expectation $\mathbb{E}_{\theta \sim \mathcal{P}}[q(\zeta; \theta)]$ must be computed as part of the definition of $q_{\mathcal{S}}(\mathcal{P})$. Since no closed-form solution is available, an empirical estimate using $M$ sampled weights $(\theta_j)_{j=1}^M$ is required to upper bound $q_{\mathcal{S}}(\mathcal{P})$ with high probability. We adopt a standard approach involving a sample convergence bound (Majumdar et al. (2021), Dziugaite & Roy (2017), Langford & Caruana (2001)). Specifically, define the empirical estimate of $q_{\mathcal{S}}(\mathcal{P})$ as:

$$\hat{q}_{\mathcal{S}}(\mathcal{P}; M) = \frac{1}{MH} \sum_{i=1}^{H} \sum_{j=1}^{M} q(\zeta_i; \theta_j). \tag{15}$$

Then, the following sample convergence bound provides an upper bound on $q_{\mathcal{S}}(\mathcal{P})$,

$$q_{\mathcal{S}}(\mathcal{P}) \leq \bar{q}_{\mathcal{S}}(\mathcal{P}; M, \epsilon) := \mathbb{D}_{\mathrm{KL}}\left(\hat{q}_{\mathcal{S}}(\mathcal{P}; M) \parallel M^{-1} \log(2/\epsilon)\right), \tag{16}$$

with probability $1 - \epsilon$. The following theorem summarizes the PAC-Bayes bound we use to evaluate the generalization capabilities of our framework.

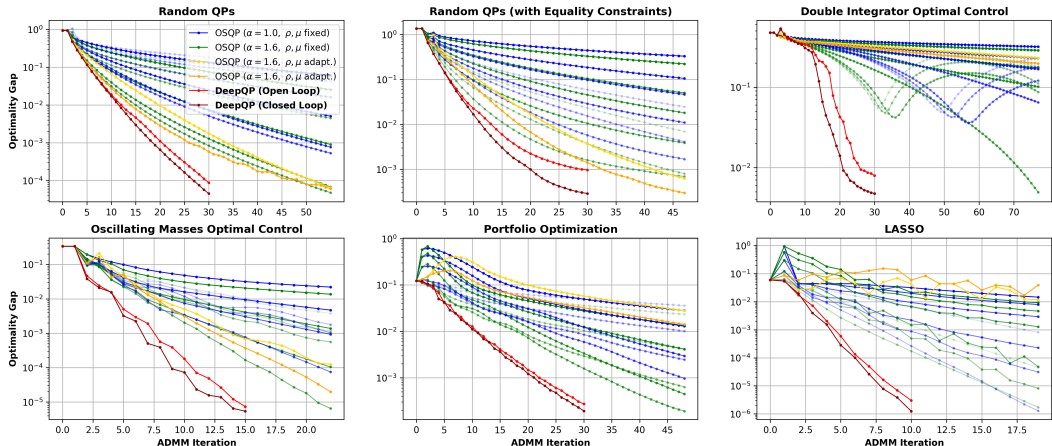

Figure 5: **Small-scale centralized comparison of DeepQP and OSQP.** Across all tested problems, DeepQP consistently outperforms OSQP (same per-iteration complexity using the indirect method).

**Theorem 3** (Generalization bound for DeepDistributedQP). *For problems $\zeta \in \mathcal{Z}$ drawn from distribution $\mathcal{D}$, the true expected progress metric of DeepDistributedQP with policy $\mathcal{P}$, i.e.,*

$$q_{\mathcal{D}}(\mathcal{P}) = \mathbb{E}_{\zeta \sim \mathcal{D}} \, \mathbb{E}_{\theta \sim \mathcal{P}} \left[ \min \left\{ \frac{\|\boldsymbol{w}^K(\zeta; \theta) - \boldsymbol{w}^*(\zeta)\|_2}{\|\boldsymbol{w}^0(\zeta) - \boldsymbol{w}^*(\zeta)\|_2}, 1 \right\} \right], \tag{17}$$

*is bounded with probability at least $1 - \delta - \epsilon$ by:*

$$q_{\mathcal{D}}(\mathcal{P}) \leq \mathbb{D}_{\mathrm{KL}}^{-1} \left( \bar{q}_{\mathcal{S}}(\mathcal{P}; M, \epsilon) \middle\| \left( \mathbb{D}_{\mathrm{KL}}(\mathcal{P} \| \mathcal{P}_0) + \log(2\sqrt{H}/\delta) \right) / H \right), \tag{18}$$

*where $\bar{q}_{\mathcal{S}}(\mathcal{P}; M, \epsilon)$ is the estimate of $q_{\mathcal{S}}(\mathcal{P}; M, \epsilon)$ described in (16).*

We explain in detail how we train for optimizing this bound in Appendix I.

## 6 EXPERIMENTS

We conduct extensive experiments to highlight the effectiveness, scalability and generalizability of the proposed methods. Section 6.1 shows the advantageous performance of DeepQP against OSQP on a variety of centralized QPs. In Section 6.2, we address large-scale problems, showcasing the scalability of DeepDistributedQP despite being trained exclusively on much lower-dimensional instances. Additionally, we discuss the advantages of learning local policies over shared ones and evaluate the proposed generalization bounds, which provide guarantees for the performance of our framework on unseen problems. An overall discussion and potential limitations are provided in Section 6.3. All experiments were performed on an system with an RTX 4090 GPU 24GB, a 13th Gen Intel(R) Core(TM) i9-13900K and 64GB of RAM.

### 6.1 SMALL-SCALE CENTRALIZED EXPERIMENTS: DEEPQP VS OSQP

**Setup.** We begin with comparing DeepQP against OSQP for solving centralized QPs (1). The following problems are considered: i,ii) random QPs without/with equality constraints, iii, iv) optimal control for double integrator and oscillating masses, v) portfolio optimization, and vi) LASSO regression. For all problems, we set a maximum allowed amount of iterations $K$ for DeepQP within $[10, 30]$ and examine how many iterations OSQP requires to reach the same accuracy. We train DeepQP using both open-loop and closed-loop policies and with a dataset of size $H \in [500, 2000]$. For OSQP, we consider both constant and adaptive penalty parameters $\rho$ and we set $\alpha$ to be either $1.0$ or $1.6$. Additional details on DeepQP, OSQP and the problems can be found in Appendix J.

**Performance comparison.** The comparison between DeepQP and OSQP is illustrated in Fig. 5. Note that both methods share the same per-iteration complexity from solving (92). We evaluate their performance by comparing the (normalized) optimality gap $\|\boldsymbol{x}^k - \boldsymbol{x}^*\|_2 / \sqrt{n}$. For all tested problems, DeepQP provides a consistent improvement over OSQP, requiring $1.5 - 3$ times fewer iterations to reach the desired accuracy. Furthermore, the advantage of incorporating feedback in the policies is shown, as closed-loop policies outperform open-loop ones in all cases.

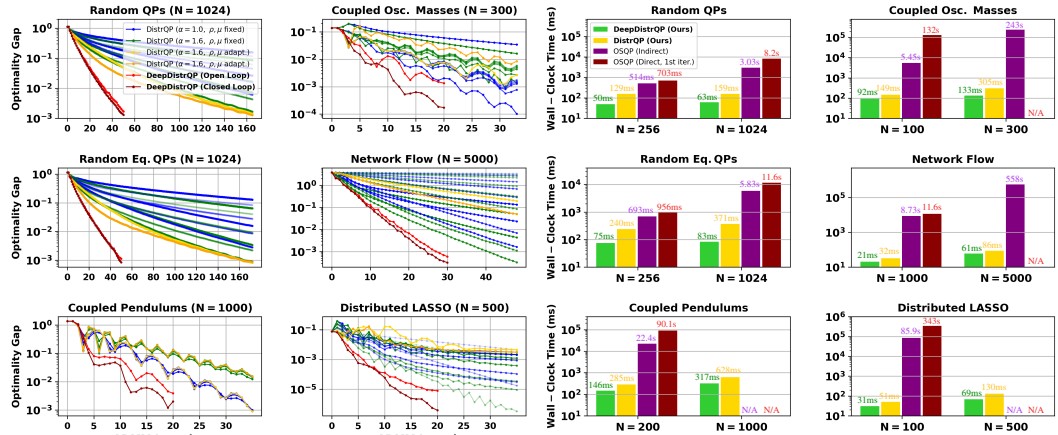

Figure 6: **Scaling DeepDistributedQP to high-dimensional problems.** Left: Comparison between DeepDistributedQP and its traditional optimization counterpart DistributedQP (same per-iteration complexity). Right: Total wall-clock time required by DeepDistributedQP, DistributedQP and OSQP (using indirect or direct method) to achieve the same accuracy.

## 6.2 LARGE-SCALE DISTRIBUTED EXPERIMENTS: SCALING DEEPDISTRIBUTEDQP

**Setup.** The purpose of the following analysis is to compare the performance and scalability of DeepDistributedQP (ours), DistributedQP (ours) and OSQP for large-scale QPs of the form (2). We consider the following six problems: i,ii) random networked QPs without/with equality constraints, iii, iv) multi-agent optimal control for coupled pendulums and oscillating masses, v) network flow, and vi) distributed LASSO. We select a maximum allowed number of iterations $K$ for DeepDistributedQP within $[20, 50]$ and examine what is the computational effort required by DistributedQP and OSQP to achieve the same accuracy measured by the optimality gap $\|\boldsymbol{w}^k - \boldsymbol{w}^*\|_2/\sqrt{n}$. More details about our experimental setup are provided in Appendix J.

**Training on low-dimensional problems.** One of the key advantages of DeepDistributedQP is that it only requires using small-scale problems for training. The training dimensions for each problem are detailed in Table 1. Both open-loop and closed-loop versions are trained using shared policies on datasets of size $H \in [500, 1000]$. We employ the shared policies version of DeepDistributedQP to enable the same policies to be applied to larger problems during testing.

**Scaling to high-dimensional problems.** Subsequently, we evaluate DeepDistributedQP on problems with significantly larger scale than those used during training. The maximum problem dimensions tested are shown in Table 1. On the left side of Fig. 6, we highlight the superior performance of DeepDistributedQP over its standard optimization counterpart DistributedQP (same per-iteration complexity). In all cases, the learned algorithm achieves the same level of accuracy while requiring 1.5-3.5 times fewer iterations. Additionally, the right side of Fig. 6 compares the total wall-clock time between DeepDistributedQP, DistributedQP and OSQP (using indirect or direct method). For a complete illustration, we refer the reader to Table 6 in Appendix J.5. The provided results emphasize the superior scalability of the two proposed distributed methods against OSQP for large-scale QPs, as well as the advantage of our deep learning-aided approach over traditional optimization.

**Local vs shared policies.** When applying a policy to a problem with the same dimensions as used during training, leveraging local policies instead of shared ones can be advantageous for better exploiting the structure of the problem. On the left side of Fig. 7, we compare the performance of local and shared policies on random QPs ($N = 16$) and coupled pendulums ($N = 10$). For the coupled pendulums problem, which exhibits significant underlying structure, local policies demonstrate clear superiority. For the random QPs problem, where structural patterns are less pronounced, the advantage of local policies is smaller but still significant.

**Performance guarantees.** Next, we verify the guarantees of our framework for generalizing on unseen random QPs ($N = 10$) and coupled pendulums ($N = 5$) problems. We switch from learning deterministic weights to learning stochastic ones and follow the procedure described in Appendix I with $H = 15000$ training samples, $M = 30000$ sampled weights for the bounds evaluation, $\delta = 0.009$ and $\epsilon = 0.001$. The resulting generalization bounds, illustrated in Fig. 7 (right), are expressed in terms of the the expected final relative optimality gap - the progress metric used for

Table 1: **Training and maximum testing dimensions for DeepDistributedQP.** The metric $\text{nnz}(\boldsymbol{Q}, \boldsymbol{A})$ denotes the total number of non-zero elements in $\boldsymbol{Q}$ and $\boldsymbol{A}$.

| | Training | | | | Max Testing | | | |
|---|---|---|---|---|---|---|---|---|
| **Problem Class** | $N$ | $n$ | $m$ | $\text{nnz}(\boldsymbol{Q}, \boldsymbol{A})$ | $N$ | $n$ | $m$ | $\text{nnz}(\boldsymbol{Q}, \boldsymbol{A})$ |
| Random QPs | 16 | 160 | 120 | 4,000 | 1,024 | 10,240 | 9,920 | 300,800 |
| Random QPs w/ Eq. Constr. | 16 | 160 | 168 | 4,960 | 1,024 | 10,240 | 9,920 | 300,800 |
| Coupled Pendulums | 10 | 470 | 640 | 3,690 | 1,000 | 47,000 | 64,000 | 380,880 |
| Coupled Osc. Masses | 10 | 470 | 1,580 | 4,590 | 300 | 28,200 | 47,400 | 141,180 |
| Network Flow | 20 | 100 | 140 | 600 | 5,000 | 25,000 | 35,000 | 150,000 |
| Distributed LASSO | 10 | 1,100 | 3,000 | 29,000 | 500 | 50,100 | 150,000 | 1,450,000 |

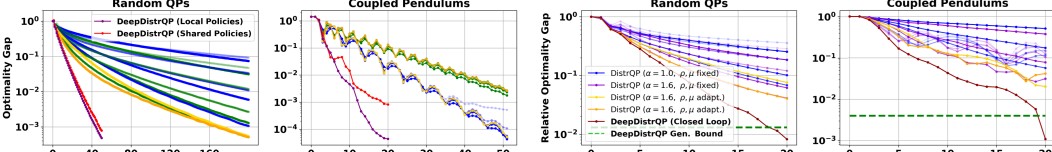

Figure 7: **Left: Local vs shared policies.** We showcase the advantage of learning local policies over shared ones. **Right: Performance guarantees.** The obtained generalization bounds guarantee the performance of DeepDistributedQP and its improvements over DistributedQP.

deriving bounds in Section 5, implying that with $99\%$ probability the average performance of our framework will be bounded by this threshold. The bounds are observed to be tight compared to actual performance, underscoring their significance. Moreover, they outperform the standard optimizers, providing a strong guarantee of improved performance for DeepDistributedQP.

### 6.3 DISCUSSION

**In which cases can we use the direct method?** As illustrated in Fig. 6 and Table 6, and further discussed in Stellato et al. (2020), the indirect method is generally preferred for solving systems of the form (4) - or (90) for DeepQP/OSQP - once the problem reaches a certain scale. In this work, we adopt this approach both for training, due the memory and computational advantages outlined in Section 4.2, and evaluating DeepDistributedQP/DeepQP. However, it is worth considering whether the direct method might be advantageous during evaluation, a choice that depends on the problem scale and capabilities of the available hardware. Overall, the results of this work show that learning policies for the algorithm parameters is significantly beneficial in the context of both distributed and centralized QP assuming the indirect method is used. In future work, we wish to also explore schemes that adapt the parameters less frequently using the direct method and/or designing mechanisms to dynamically switch between the two approaches.

**Limitations.** One limitation of the proposed framework is its reliance on a supervised training loss, requiring a dataset of pre-solved problems. In future work, we aim to explore training through directly minimizing the problem residuals rather than the optimality gaps. Furthermore, while PAC-Bayes theory provides an important probabilistic bound on average performance, stronger guarantees may be necessary for safety-critical applications to ensure reliability and robustness.

## 7 CONCLUSION AND FUTURE WORK

In this work, we introduced DeepDistributedQP, a new deep learning-aided distributed optimization architecture for solving large-scale QP problems. The proposed method relies on unfolding the iterations of a novel optimizer named DistributedQP as layers of a supervised deep learning framework. The expected performance of our learned optimizer on unseen problems is also theoretically established through PAC-Bayes theory. DeepDistributedQP exhibits impressive scalability in effectively tackling large-scale optimization problems while being trained exclusively on much smaller ones. In addition, both DeepDistributedQP and DistributedQP significantly outperform OSQP in terms of required wall-clock time to reach the same accuracy as dimension increases. Furthermore, we showcase that the proposed PAC-Bayes bounds provide meaningful practical guarantees for the performance of the learned optimizer on new problems. In future work, we wish to extend the proposed framework to a semi-supervised version that relies less on pre-solved problems for training. In addition, we wish to explore incorporating more complex learnable components such as LSTMs for feedback within our architecture. Finally, we wish to consider other classes of distributed constrained optimization methods outside of quadratic programming.

ACKNOWLEDGMENTS

This work is supported by the National Aeronautics and Space Administration under ULI Grant 80NSSC22M0070 and the ARO Award #W911NF2010151. Augustinos Saravanos acknowledges financial support by the A. Onassis Foundation Scholarship. The authors also thank Alec Farid for helpful discussions on PAC-Bayes Theory.

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

# APPENDIX

## CONTENTS

## A    EXTENDED RELATED WORK

**Distributed optimization with ADMM.**    The ADMM method was first introduced in Glowinski & Marroco (1975) and Gabay & Mercier (1976), yet it only became widely popular after the highly influential review by Boyd et al. (2011), which highlighted its suitability for deriving distributed algorithms whose computations can be parallelized. Since then, ADMM-based architectures have been introduced in a variety of setups including distributed learning (Huang et al., 2019; Zhou & Li, 2023), multi-agent robotics (Saravanos et al., 2023a; Shorinwa et al., 2024), signal processing (Mateos et al., 2010; Zhang & Kwok, 2014), stochastic systems (Saravanos et al., 2021; 2023b), sensor networks (Mota et al., 2013), power grids (Erseghe, 2014) and many other areas, often exhibiting an impressive scalability for high-dimensional problems. Nevertheless, it is well-known that the convergence speed of distributed ADMM algorithms is heavily dependent on the choice of their underlying tuning parameters. To address this difficulty, several adaptation schemes have been presented, typically relying on the residuals of the current iterates (Song et al., 2016; Xu et al., 2017a;b). Nevertheless, such approaches tend to be myopic and cannot leverage data or provide guarantees on the performance of the algorithms on unseen problems under finite iterations.

**Learning-to-optimize for distributed optimization.**    The concept of integrating learning-to-optimize approaches into distributed optimization is particularly compelling, as algorithms of the latter class typically rely on a significant amount of designing and tuning by experts. Nevertheless, the area of distributed learning-to-optimize methods remains largely unexplored. For instance, although distributed ADMM has achieved widespread success in decentralized constrained optimization, its unfolded extension as a deep learning network has only been recently explored by Noah & Shlezinger (2024). This framework shows promising results, but it is limited to an unconstrained problem formulation, assumes gradient-based local updates, focuses solely on parameter tuning and lacks performance guarantees. Biagioni et al. (2020) presented an ADMM framework which utilizes recurrent neural networks for predicting the converged values of the variables demonstrating substantial improvements in convergence speed. In Zeng et al. (2022), a reinforcement learning (RL) approach for learning the optimal parameters of distributed ADMM was proposed, showing promising speed improvements, but requiring a substantial amount of training effort.

Beyond distributed ADMM, Wang et al. (2021) proposed unrolling two decentralized first-order optimization algorithms (ProxDGD and PG-Extra) as graph neural networks (GNNs) for addressing the decentralized statistical inference problem. Hadou et al. (2023) presented a distributed gradient descent algorithm unrolled as a GNN focusing on the federated learning problem setup. From a different point of view, He et al. (2024) recently introduced a distributed gradient-based learning-to-optimize framework for unconstrained optimization which partially imposes structure on the learnable updates instead of unrolling predefined iterations. A deep RL approach for adapting the local updates of the approximate method of multipliers was recently proposed in Zhu et al. (2023). Finally, Kishida et al. (2020) and Ogawa & Ishii (2021) have presented distributed learned optimization methods for tackling the average consensus problem.

**Learning-to-optimize for (centralized) QP.**    Recent works have focused on accelerating QP through learning; however these efforts have solely concentrated on a centralized setup. In particular, Ichnowski et al. (2021) introduced an RL-based algorithm for accelerating OSQP demonstrating promising reductions in iterations, yet training this algorithm incurs significant computational costs. From a different perspective, Sambharya et al. (2023; 2024) focused on learning-to-initialize fixed-point methods including OSQP, while maintaining constant parameters in the unrolled algorithm iterations. Concurrently with the development of the present work, Sambharya & Stellato (2024b) presented a methodology for selecting the optimal algorithm parameters for various first-order optimization methods. Considering OSQP as the unrolled method coincides with the open-loop version of the proposed DeepQP framework without any notion of feedback policies.

**Generalization guarantees for learning-to-optimize.**    The works in Sucker & Ochs (2023) and Sucker et al. (2024) presented generalization bounds for learned optimizers, considering the update function as a gradient step or a multi-layer perceptron, respectively. Sambharya & Stellato (2024a) recently also explored incorporating PAC-Bayes bounds in learning-to-optimize methods without assuming a specific underlying algorithm structure. However, our approach differs fundamentally,

as their method employs a binary error function, whereas in this work we directly establish bounds based on the optimality gap of the final solution.

# B    COMPLETE DERIVATION OF DISTRIBUTEDQP ALGORITHM

**Problem transformation and augmented Lagrangian.**    Here, we present the detailed derivation of the DistributedQP algorithm presented in Section 3.2. We consider the over-relaxed version of ADMM (Boyd et al., 2011) with $\alpha \in [1, 2)$. First, let us rewrite problem (2) as

$$\min_{\boldsymbol{x}} \sum_{i \in \mathcal{V}} \frac{1}{2} \boldsymbol{x}_i^\top \boldsymbol{Q}_i \boldsymbol{x}_i + \boldsymbol{q}_i^\top \boldsymbol{x}_i \quad \text{s.t.} \quad \boldsymbol{A}_i \boldsymbol{x}_i = \boldsymbol{z}_i, \ \boldsymbol{z}_i \leq \boldsymbol{b}_i, \ \boldsymbol{x}_i = \tilde{\boldsymbol{w}}_i, \quad i \in \mathcal{V}, \qquad (19)$$

where we have introduced the auxiliary variables $\boldsymbol{z}_i$ for each $i = 1, \dots, N$. In addition, let us define the variables $\boldsymbol{s}_i, \ i = 1, \dots, N$, and rewrite problem (19) as

$$\min \sum_{i \in \mathcal{V}} \frac{1}{2} \boldsymbol{x}_i^\top \boldsymbol{Q}_i \boldsymbol{x}_i + \boldsymbol{q}_i^\top \boldsymbol{x}_i \quad \text{s.t.} \quad \boldsymbol{A}_i \boldsymbol{x}_i = \boldsymbol{z}_i, \ \boldsymbol{s}_i \leq \boldsymbol{b}_i, \ \boldsymbol{z}_i = \boldsymbol{s}_i, \ \boldsymbol{x}_i = \tilde{\boldsymbol{w}}_i, \ i \in \mathcal{V}. \qquad (3)$$

The above splitting constitutes the problem suitable for being addressed with a two-block ADMM scheme, where the first block of variables consists of $\{\boldsymbol{x}_i, \boldsymbol{z}_i\}_{i=1,\dots,N}$, and the second one contains $\{\boldsymbol{s}_i\}_{i=1,\dots,N}$ and $\boldsymbol{w}$. The (scaled) augmented Lagrangian (AL) for problem (3) is given by

$$\mathcal{L} = \sum_{i \in \mathcal{V}} \frac{1}{2} \boldsymbol{x}_i^\top \boldsymbol{Q}_i \boldsymbol{x}_i + \boldsymbol{q}_i^\top \boldsymbol{x}_i + \mathcal{I}_{\boldsymbol{A}_i \boldsymbol{x}_i = \boldsymbol{z}_i}(\boldsymbol{x}_i, \boldsymbol{z}_i) + \mathcal{I}_{\boldsymbol{s}_i \leq \boldsymbol{b}_i}(\boldsymbol{s}_i)$$
$$+ \frac{\rho_i}{2} \left\| \boldsymbol{z}_i - \boldsymbol{s}_i + \frac{\boldsymbol{\lambda}_i}{\rho_i} \right\|_2^2 + \frac{\mu_i}{2} \left\| \boldsymbol{x}_i - \tilde{\boldsymbol{w}}_i + \frac{\boldsymbol{y}_i}{\mu_i} \right\|_2^2. \qquad (20)$$

**First block of primal updates.**    The first block of variables is updated through

$$\{\boldsymbol{x}_i, \boldsymbol{z}_i\}_{i \in \mathcal{V}} = \arg \min \mathcal{L}(\boldsymbol{x}, \boldsymbol{z}, \boldsymbol{s}^k, \boldsymbol{w}^k, \boldsymbol{\lambda}^k, \boldsymbol{y}^k).$$

This minimization can be decoupled to the following $N$ subproblems for each $i \in \mathcal{V}$,

$$\{\boldsymbol{x}_i, \boldsymbol{z}_i\} = \arg \min \frac{1}{2} \boldsymbol{x}_i^\top \boldsymbol{Q}_i \boldsymbol{x}_i + \boldsymbol{q}_i^\top \boldsymbol{x}_i + \frac{\rho_i}{2} \left\| \boldsymbol{z}_i - \boldsymbol{s}_i + \frac{\boldsymbol{\lambda}_i}{\rho_i} \right\|_2^2 + \frac{\mu_i}{2} \left\| \boldsymbol{x}_i - \tilde{\boldsymbol{w}}_i + \frac{\boldsymbol{y}_i}{\mu_i} \right\|_2^2$$
$$\text{s.t.} \quad \boldsymbol{A}_i \boldsymbol{x}_i = \boldsymbol{z}_i,$$

where we have temporarily dropped the superscript iteration indices for convenience. Since these problems are equality-constrained QPs, we can obtain a closed-form solution. In particular, the KKT conditions for each subproblem are given by

$$\boldsymbol{Q}_i \boldsymbol{x}_i + \boldsymbol{q}_i + \mu_i(\boldsymbol{x}_i - \tilde{\boldsymbol{w}}_i) + \boldsymbol{y}_i + \boldsymbol{A}_i^\top \boldsymbol{\nu}_i = \boldsymbol{0}, \qquad (21a)$$
$$\rho_i(\boldsymbol{z}_i - \boldsymbol{s}_i) + \boldsymbol{\lambda}_i - \boldsymbol{\nu}_i = \boldsymbol{0}, \qquad (21b)$$
$$\boldsymbol{A}_i \boldsymbol{x}_i - \boldsymbol{z}_i = \boldsymbol{0}, \qquad (21c)$$

where $\boldsymbol{\nu}_i$ is the Lagrange multiplier corresponding to the constraint $\boldsymbol{A}_i \boldsymbol{x}_i = \boldsymbol{z}_i$. Eliminating $\boldsymbol{z}_i$ leads to the following system of equations

$$\begin{bmatrix} \boldsymbol{Q}_i + \mu_i \boldsymbol{I} & \boldsymbol{A}_i^\top \\ \boldsymbol{A}_i & -1/\rho_i \boldsymbol{I} \end{bmatrix} \begin{bmatrix} \boldsymbol{x}_i^{k+1} \\ \boldsymbol{\nu}_i^{k+1} \end{bmatrix} = \begin{bmatrix} -\boldsymbol{q}_i + \mu_i \tilde{\boldsymbol{w}}_i^k - \boldsymbol{y}_i^k \\ \boldsymbol{z}_i - 1/\rho_i \boldsymbol{\lambda}_i^k \end{bmatrix}, \qquad (22)$$

with $\boldsymbol{z}_i^{k+1}$ given by

$$\boldsymbol{z}_i^{k+1} = \boldsymbol{s}_i^k + \rho_i^{-1}(\boldsymbol{\nu}_i^{k+1} - \boldsymbol{\lambda}_i^k). \qquad (23)$$

**Second block of primal updates.**    The second block of updates is given by

$$\{\boldsymbol{s}_i\}_{i \in \mathcal{V}}, \boldsymbol{w} = \arg \min \mathcal{L}(\boldsymbol{x}^{k+1}, \boldsymbol{z}^{k+1}, \boldsymbol{s}, \boldsymbol{w}, \boldsymbol{\lambda}^k, \boldsymbol{y}^k),$$

or more analytically by

$$\{s_i\}_{i \in \mathcal{V}}, w = \arg\min \sum_{i \in \mathcal{V}} \frac{\rho_i}{2} \left\| \alpha z_i^{k+1} + (1-\alpha)s_i^k - s_i + \frac{\lambda_i^k}{\rho_i} \right\|_2^2$$
$$+ \frac{\mu_i}{2} \left\| \alpha x_i^{k+1} + (1-\alpha)\tilde{w}_i^k - \tilde{w}_i + \frac{y_i^k}{\mu_i} \right\|_2^2 \quad \text{s.t.} \quad s_i \leq b_i.$$

Note that this minimization can be decoupled w.r.t. all $s_i$, $i \in \mathcal{V}$, and $w$. In particular, each $s_i$ is updated in parallel through

$$s_i^{k+1} = \Pi_{s_i \leq b_i} \left( \alpha z_i^{k+1} + (1-\alpha^k)s_i^k + \lambda_i^k/\rho_i \right). \tag{24}$$

The global variable $w$ minimization can also be decoupled among its components $l = 1, \ldots, n$, which gives

$$w_l = \arg\min \sum_{\mathcal{G}(i,j)=l} \frac{\mu_i}{2} \left\| \alpha[x_i^{k+1}]_j + (1-\alpha)[\tilde{w}_i^k]_j - [\tilde{w}_i]_j + \frac{[y_i^k]_j}{\mu_i} \right\|_2^2.$$

Setting the gradient to be equal to zero yields

$$\sum_{\mathcal{G}(i,j)=l} \mu_i \left[ \alpha[x_i^{k+1}]_j + (1-\alpha)w_l^k - w_l^{k+1} + \frac{[y_i^k]_j}{\mu_i} \right] = 0,$$

leading to

$$\sum_{\mathcal{G}(i,j)=l} \mu_i w_l^{k+1} = \sum_{\mathcal{G}(i,j)=l} \mu_i \left[ \alpha[x_i^{k+1}]_j + (1-\alpha)w_l^k + \frac{[y_i^k]_j}{\mu_i} \right],$$

which eventually gives the update rule

$$w_l^{k+1} = \frac{\sum_{\mathcal{G}(i,j)=l} \alpha\mu_i[x_i^{k+1}]_j + [y_i^k]_j}{\sum_{\mathcal{G}(i,j)=l} \mu_i} + (1-\alpha)w_l^k. \tag{25}$$

**Dual updates.** Finally, the dual variables are updated through dual ascent steps as follows

$$\lambda_i^{k+1} = \lambda_i^k + \rho_i(\alpha z_i^{k+1} + (1-\alpha)s_i^k - s_i^{k+1}), \tag{26}$$
$$y_i^{k+1} = y_i^k + \mu_i(\alpha x_i^{k+1} + (1-\alpha)\tilde{w}_i^k - \tilde{w}_i^{k+1}). \tag{27}$$

**Simplifying the global update.** It is important to observe that after the first iteration (for $k \geq 1$), the global update can be simplified to

$$w_l^{k+1} = \alpha \frac{\sum_{\mathcal{G}(i,j)=l} \mu_i[x_i^{k+1}]_j}{\sum_{\mathcal{G}(i,j)=l} \mu_i} + (1-\alpha)w_l^k, \tag{28}$$

since the summation

$$
\begin{aligned}
\sum_{\mathcal{G}(i,j)=l} [\boldsymbol{y}_i^{k+1}]_j &= \sum_{\mathcal{G}(i,j)=l} [\boldsymbol{y}_i^k]_j + \mu_i(\alpha[\boldsymbol{x}_i^{k+1}]_j + (1-\alpha)[\tilde{\boldsymbol{w}}_i^k]_j - [\tilde{\boldsymbol{w}}_i^{k+1}]_j) \\
&= \sum_{\mathcal{G}(i,j)=l} [\boldsymbol{y}_i^k]_j + \mu_i(\alpha[\boldsymbol{x}_i^{k+1}]_j + (1-\alpha)\boldsymbol{w}_l^k - \boldsymbol{w}_l^{k+1}) \\
&= \sum_{\mathcal{G}(i,j)=l} [\boldsymbol{y}_i^k]_j + \mu_i\Bigg[\alpha[\boldsymbol{x}_i^{k+1}]_j + \cancel{(1-\alpha)\boldsymbol{w}_l^k} \\
&\qquad - \frac{\sum_{\mathcal{G}(u,v)=l}\alpha\mu_u[\boldsymbol{x}_u^{k+1}]_v + [\boldsymbol{y}_u^k]_v}{\sum_{\mathcal{G}(u,v)=l}\mu_u} - \cancel{(1-\alpha)\boldsymbol{w}_l^k}\Bigg] \\
&= \sum_{\mathcal{G}(i,j)=l} [\boldsymbol{y}_i^k]_j + \mu_i\Bigg[\alpha[\boldsymbol{x}_i^{k+1}]_j - \frac{\sum_{\mathcal{G}(u,v)=l}\alpha\mu_u[\boldsymbol{x}_u^{k+1}]_v + [\boldsymbol{y}_u^k]_v}{\sum_{\mathcal{G}(u,v)=l}\mu_u}\Bigg] \\
&= \sum_{\mathcal{G}(i,j)=l} [\boldsymbol{y}_i^k]_j + \alpha\mu_i[\boldsymbol{x}_i^{k+1}]_j - \frac{\cancel{\sum_{\mathcal{G}(i,j)=l}\mu_i}\Big[\sum_{\mathcal{G}(u,v)=l}\alpha\mu_u[\boldsymbol{x}_u^{k+1}]_v + [\boldsymbol{y}_u^k]_v\Big]}{\cancel{\sum_{\mathcal{G}(u,v)=l}\mu_u}} \\
&= \sum_{\mathcal{G}(i,j)=l} [\boldsymbol{y}_i^k]_j + \alpha\mu_i[\boldsymbol{x}_i^{k+1}]_j - \sum_{\mathcal{G}(u,v)=l}\alpha\mu_u[\boldsymbol{x}_u^{k+1}]_v + [\boldsymbol{y}_u^k]_v = 0.
\end{aligned}
\tag{29}
$$

## C   STANDARD CONVERGENCE GUARANTEES FOR SIMPLIFIED VERSION OF DISTRIBUTEDQP

In the simplified case where $\rho_i^k = \rho$, $\mu_i^k = \mu$ for all $i \in \mathcal{V}$ and for all $k$, as well as $\alpha^k = 1$ for all $k$, it would be straightforward to apply the classical convergence guarantees of two-block ADMM for convex optimization problems (Deng & Yin, 2016) to ensure the convergence of DistributedQP. In the following, we show how DistributedQP would fit under this setup.

Let us define the concatenated variables $\bar{\boldsymbol{x}} = [\{\boldsymbol{x}_i\}_{i \in \mathcal{V}}; \{\boldsymbol{z}_i\}_{i \in \mathcal{V}}]$ and $\bar{\boldsymbol{z}} = [\{\boldsymbol{s}_i\}_{i \in \mathcal{V}}; \boldsymbol{w}]$. Then, we can rewrite problem (3) as

$$
\min f(\bar{\boldsymbol{x}}) + g(\bar{\boldsymbol{z}}) \quad \text{s.t.} \quad \bar{\boldsymbol{A}}\bar{\boldsymbol{x}} + \bar{\boldsymbol{B}}\bar{\boldsymbol{z}} = \bar{\boldsymbol{c}},
\tag{30}
$$

where

$$
f(\bar{\boldsymbol{x}}) = \sum_{i \in \mathcal{V}} \frac{1}{2}\boldsymbol{x}_i^\top \boldsymbol{Q}_i \boldsymbol{x}_i + \boldsymbol{q}_i^\top \boldsymbol{x}_i + \mathcal{I}_{\boldsymbol{A}_i\boldsymbol{x}_i=\boldsymbol{z}_i}(\boldsymbol{x}_i, \boldsymbol{z}_i), \quad g(\bar{\boldsymbol{z}}) = \sum_{i \in \mathcal{V}} \mathcal{I}_{\boldsymbol{s}_i \leq \boldsymbol{b}_i}(\boldsymbol{s}_i),
\tag{31}
$$

and $\bar{\boldsymbol{A}} = \text{bdiag}(\boldsymbol{I}, \boldsymbol{I})$, $\bar{\boldsymbol{B}} = \text{bdiag}(\boldsymbol{I}, \boldsymbol{G})$ and $\boldsymbol{c} = \boldsymbol{0}$, with $\boldsymbol{G} \in \mathbb{R}^{(\sum_i n_i) \times n}$ defined such that $\boldsymbol{x} = \boldsymbol{G}\boldsymbol{w}$. In other words, $\boldsymbol{G}$ is the matrix that represents the local-to-global variable components mapping, formally defined as $\boldsymbol{G} = [\boldsymbol{G}_1; \ldots; \boldsymbol{G}_N]$ with each submatrix $\boldsymbol{G}_i \in \mathbb{R}^{n_i \times n}$ given by

$$
[\boldsymbol{G}_i]_{u,v} = \begin{cases} 1, & \text{if } v = \mathcal{G}(i,u) \\ 0, & \text{else} \end{cases}.
\tag{32}
$$

Given this representation, it becomes clear that our algorithm can be framed as a two-block ADMM. Now, note that $\boldsymbol{G}$ is a full column rank matrix since all global variable components $\boldsymbol{g}_l$ are mapped to at least one local variable component $[\boldsymbol{x}_i]_j$. Then, since the functions $f, g$ are convex and the matrices $\bar{\boldsymbol{A}}, \bar{\boldsymbol{B}}$ are full column rank, it follows from Deng & Yin (2016) that the algorithm is guaranteed to converge to the optimal solution of problem (3).

Nevertheless, this analysis is solely applicable to this simplified version of the DistributedQP algorithm. In Appendix D, we tackle the more complex case of iteration-varying relaxation and local penalty parameters.

## D   PROOF OF DISTRIBUTEDQP ASYMPTOTIC CONVERGENCE

In this section, we prove that DistributedQP is guaranteed to converge to optimality, even in the more challenging case of iteration-varying relaxation and local penalty parameters (Theorem 1).

The following analysis extends the theoretical results presented in Xu et al. (2017a), where the convergence of an adaptive relaxed variant of two-block ADMM is provided. Nevertheless, this analysis is not directly applicable to our case which involves distinct local penalty parameters per computational node.

## D.1 SKETCH OF PROOF

To begin, we outline the following conventions. The points $\boldsymbol{x}^*, \boldsymbol{z}^*, \boldsymbol{s}^*, \boldsymbol{w}^*, \boldsymbol{y}^*, \boldsymbol{\lambda}^*$ are the KKT points of problem (3). We refer to the notion of a distance function at any $(k+1)^{th}$ iteration to be representing a weighted squared norm of the difference between the variables $\boldsymbol{s}^{k+1}, \boldsymbol{w}^{k+1}, \boldsymbol{y}^{k+1}, \boldsymbol{\lambda}^{k+1}$ and their corresponding optimal values $\boldsymbol{s}^*, \boldsymbol{w}^*, \boldsymbol{y}^*, \boldsymbol{\lambda}^*$, indicating the distance from the optimal point.

We prove the convergence of DistributedQP in the following steps:

- **Necessary Lemmas (Section D.2)**. First, we derive the descent relation (65), which establishes a relationship between the values of the distance function for consecutive iterations. To derive this descent relation in Lemma 4, we first introduce the relations (R1)-(R8) in Lemmas 1-3.

- **Convergence (Section D.3)**. Next, we use the derived descent relation from Lemma 4 to prove convergence to optimality (Theorem 1) based on Assumption 1.

## D.2 NECESSARY LEMMAS

In this section, we present some necessary lemmas before proving the convergence of DistributedQP in Section D.3. For notational convenience, let us define

$$f_i(\boldsymbol{x}_i) = \frac{1}{2}\boldsymbol{x}_i^\top \boldsymbol{Q}_i \boldsymbol{x}_i + \boldsymbol{q}_i^\top \boldsymbol{x}_i, \quad \mathcal{C}_i = \{\boldsymbol{s}_i | \boldsymbol{s}_i \leq \boldsymbol{b}_i\}, \quad i \in \mathcal{V}.$$

**Lemma 1.** *For all $i \in \mathcal{V}$, the following four relationships hold at every iteration $k$:*

$$\sum_{i\in\mathcal{V}} \boldsymbol{G}_i^\top \boldsymbol{y}_i^{k+1} = \boldsymbol{0}, \tag{R1}$$

$$\alpha^k \boldsymbol{x}_i^{k+1} = \frac{1}{\mu_i^k}(\boldsymbol{y}_i^{k+1} - \boldsymbol{y}_i^k) - (1-\alpha^k)\boldsymbol{G}_i\boldsymbol{w}^k + \boldsymbol{G}_i\boldsymbol{w}^{k+1}, \tag{R2}$$

$$\alpha^k \boldsymbol{z}_i^{k+1} = \frac{1}{\rho_i^k}(\boldsymbol{\lambda}_i^{k+1} - \boldsymbol{\lambda}_i^k) - (1-\alpha^k)\boldsymbol{s}_i^k + \boldsymbol{s}_i^{k+1}, \tag{R3}$$

$$\boldsymbol{\lambda}_i^{k\top}(\boldsymbol{t}_1 - \boldsymbol{t}_2) = 0, \quad \forall \, \boldsymbol{t}_1, \boldsymbol{t}_2 \in \mathcal{C}_i. \tag{R4}$$

*Proof.* Relationship (R1) is equivalent with the argument proved in (29). Indeed, if we observe that each matrix $\boldsymbol{G}_i^\top \in \mathbb{R}^{n \times n_i}$ indicates the mapping from local indices $(i,j)$ to global indices $l$ for a particular $i$, then we can write

$$\sum_{i\in\mathcal{V}} \boldsymbol{G}_i^\top \boldsymbol{y}_i^{k+1} = \begin{bmatrix} \sum_{\mathcal{G}(i,j)=1}[\boldsymbol{y}_i^{k+1}]_j \\ \vdots \\ \sum_{\mathcal{G}(i,j)=n}[\boldsymbol{y}_i^{k+1}]_j \end{bmatrix} = \boldsymbol{0}, \tag{33}$$

which yields (R1). Relationship (R2) follows by rearranging the dual update (9) and replacing $\tilde{\boldsymbol{w}}_i = \boldsymbol{G}_i\boldsymbol{w}$. Similarly, relationship (R3) follows by rearranging the dual update (8). In the remaining, we focus on proving (R4). Let us repeat the $\boldsymbol{s}_i$-update (6) as

$$\boldsymbol{s}_i^{k+1} = \Pi_{\mathcal{C}_i}\left(\alpha^k \boldsymbol{z}_i^{k+1} + (1-\alpha^k)\boldsymbol{s}_i^k + \boldsymbol{\lambda}_i^k/\rho_i^k\right). \tag{34}$$

We define the closed convex cone $\bar{\mathcal{C}}_i = \{\boldsymbol{p} | \, \boldsymbol{p} \leq \boldsymbol{0}\}$, such that (34) is rewritten as

$$\boldsymbol{s}_i^{k+1} = \Pi_{\bar{\mathcal{C}}_i}\left(\hat{\boldsymbol{s}}_i^{k+1}\right) + \boldsymbol{b}_i, \tag{35}$$

with

$$\hat{\boldsymbol{s}}_i^{k+1} = \alpha^k \boldsymbol{z}_i^{k+1} + (1-\alpha^k)\boldsymbol{s}_i^k + \boldsymbol{\lambda}_i^k/\rho_i^k - \boldsymbol{b}_i. \tag{36}$$

Next, let us rearrange the dual update (8) as

$$\boldsymbol{\lambda}_i^{k+1} = \rho_i^k(\boldsymbol{\lambda}_i^k/\rho_i^k + \alpha^k \boldsymbol{z}_i^{k+1} + (1-\alpha^k)\boldsymbol{s}_i^k - \boldsymbol{s}_i^{k+1}), \tag{37}$$

which can be rewritten through (36) as

$$\boldsymbol{\lambda}_i^{k+1} = \rho_i^k(\hat{\boldsymbol{s}}_i^{k+1} + \boldsymbol{b}_i - \boldsymbol{s}_i^{k+1}) \tag{38}$$

Substituting (35) in the above, we get

$$\boldsymbol{\lambda}_i^{k+1} = \rho_i^k\left(\hat{\boldsymbol{s}}_i^{k+1} - \Pi_{\bar{\mathcal{C}}_i}(\hat{\boldsymbol{s}}_i^{k+1})\right). \tag{39}$$

For convenience, let us also repeat the definition of polar cones.

**Definition 1** (Polar cones). *Two cone sets $\mathcal{D}$ and $\mathcal{D}^o$ are called polar cones if for any $\boldsymbol{d} \in \mathcal{D}$ and $\bar{\boldsymbol{d}} \in \mathcal{D}^o$, if follows that $\boldsymbol{d}^\top \bar{\boldsymbol{d}} = 0$.*

By Moreau's decomposition - refer to Theorems 1.1 and 1.2 from Soltan (2019) - $\hat{\boldsymbol{s}}_i^{k+1}$ can then be expressed as

$$\hat{\boldsymbol{s}}_i^{k+1} = \Pi_{\bar{\mathcal{C}}_i}\left(\hat{\boldsymbol{s}}_i^{k+1}\right) + \Pi_{\bar{\mathcal{C}}_i^o}\left(\hat{\boldsymbol{s}}_i^{k+1}\right), \tag{40}$$

where $\bar{\mathcal{C}}_i^o$ is a polar cone to $\bar{\mathcal{C}}_i$. Thus, using (39) and (40), we get $\boldsymbol{\lambda}_i^{k+1} = \rho_i^k\Pi_{\bar{\mathcal{C}}_i^o}\left(\hat{\boldsymbol{s}}_i^{k+1}\right)$, which implies that $\boldsymbol{\lambda}_i^{k+1}/\rho_i^k \in \bar{\mathcal{C}}_i^o$. Further, since $\bar{\mathcal{C}}_i^o$ is a cone, and $\rho_i^k > 0$, we get

$$\boldsymbol{\lambda}_i^{k+1} \in \bar{\mathcal{C}}_i^o. \tag{41}$$

Now, any vector $\boldsymbol{t} \in \mathcal{C}_i$ satisfies $\boldsymbol{t} - \boldsymbol{b}_i \in \bar{\mathcal{C}}_i$. Since $\bar{\mathcal{C}}_i$ and $\bar{\mathcal{C}}_i^o$ are polar cones, and using (41), the following relation holds true by the definition of polar cones,

$$\boldsymbol{\lambda}_i^{k+1\top}(\boldsymbol{t} - \boldsymbol{b}_i) = 0 \quad \text{for all } \boldsymbol{t} \in \mathcal{C}_i.$$

Thus, for any vectors $\boldsymbol{t}_1, \boldsymbol{t}_2 \in \mathcal{C}_i$ and for all $k$, we have

$$\boldsymbol{\lambda}_i^{k+1\top}(\boldsymbol{t}_1 - \boldsymbol{t}_2) = \boldsymbol{\lambda}_i^{k+1\top}(\boldsymbol{t}_1 - \boldsymbol{b}_i - (\boldsymbol{t}_2 - \boldsymbol{b}_i)) = 0, \tag{42}$$

which proves (R4). $\qquad\square$

**Lemma 2.** *For all $i \in \mathcal{V}$, the following two relationships hold at every iteration $k$:*

$$\left(\nabla f_i(\boldsymbol{x}_i^*) + \boldsymbol{y}_i^*\right)^\top(\boldsymbol{x}_i^* - \boldsymbol{x}_i^{k+1}) + \boldsymbol{\lambda}_i^{*\top}(\boldsymbol{z}_i^* - \boldsymbol{z}_i^{k+1}) = 0, \tag{R5}$$

$$\left[\nabla f_i(\boldsymbol{x}_i^{k+1}) + \boldsymbol{y}_i^{k+1} + \mu_i^k\left((1-\alpha^k)\boldsymbol{x}_i^{k+1} - (2-\alpha^k)\boldsymbol{G}_i\boldsymbol{w}^k + \boldsymbol{G}_i\boldsymbol{w}^{k+1}\right)\right]^\top(\boldsymbol{x}_i^{k+1} - \boldsymbol{x}_i^*)$$

$$+ \left[\boldsymbol{\lambda}_i^{k+1} + \rho_i^k\left((1-\alpha^k)\boldsymbol{z}_i^{k+1} - (2-\alpha^k)\boldsymbol{s}_i^k + \boldsymbol{s}_i^{k+1}\right)\right]^\top(\boldsymbol{z}_i^{k+1} - \boldsymbol{z}_i^*) = 0. \tag{R6}$$

*Proof.* We start with proving (R5) using the KKT conditions of problem (3). The point $(\boldsymbol{x}^*, \boldsymbol{z}^*, \boldsymbol{s}^*, \boldsymbol{w}^*)$ is the optimum of (3) if and only if the following conditions are true:

$$\text{Optimality for } \boldsymbol{x}_i: \qquad \nabla f_i(\boldsymbol{x}_i^*) + \boldsymbol{A}_i^\top\boldsymbol{\nu}_i^* + \boldsymbol{y}_i^* = \boldsymbol{0}, \tag{43a}$$

$$\text{Optimality for } \boldsymbol{z}_i: \qquad -\boldsymbol{\nu}_i^* + \boldsymbol{\lambda}_i^* = \boldsymbol{0}, \tag{43b}$$

$$\text{Optimality for } \boldsymbol{s}_i: \qquad \boldsymbol{\lambda}_i^* \in \mathcal{N}_{\mathcal{C}_i}(\boldsymbol{s}_i^*) \Leftrightarrow \boldsymbol{\lambda}_i^{*\top}(\boldsymbol{s}_i - \boldsymbol{s}_i^*) \leq 0 \ \forall \ \boldsymbol{s}_i \in \mathcal{C}_i, \tag{43c}$$

$$\text{Optimality for } \boldsymbol{w}: \qquad \sum_{i \in \mathcal{V}} \boldsymbol{G}_i^\top\boldsymbol{y}_i^* = \boldsymbol{0}, \tag{43d}$$

$$\text{Constraints feasibility:} \qquad \tilde{\boldsymbol{z}}_i^* = \boldsymbol{s}_i^*, \tag{43e}$$

$$\boldsymbol{x}_i^* = \boldsymbol{G}_i\boldsymbol{w}^*, \tag{43f}$$

$$\boldsymbol{A}_i\boldsymbol{x}_i^* = \boldsymbol{z}_i, \tag{43g}$$

$$\boldsymbol{s}_i^* \in \mathcal{C}_i. \tag{43h}$$

From (43a), we have

$$\left(\nabla f_i(\boldsymbol{x}_i^*) + \boldsymbol{A}_i^\top\boldsymbol{\nu}_i^* + \boldsymbol{y}_i^*\right)^\top(\boldsymbol{x}_i^* - \boldsymbol{x}_i^{k+1}) = 0, \tag{44}$$

and similarly from (43b), we get

$$\big(-\boldsymbol{\nu}_i^* + \boldsymbol{\lambda}_i^*\big)^\top (\boldsymbol{z}_i^* - \boldsymbol{z}_i^{k+1}) = 0. \tag{45}$$

Adding (44) and (45), we get

$$\big(\nabla f_i(\boldsymbol{x}_i^*) + \boldsymbol{A}_i^\top \boldsymbol{\nu}_i^* + \boldsymbol{y}_i^*\big)^\top (\boldsymbol{x}_i^* - \boldsymbol{x}_i^{k+1}) + \big(-\boldsymbol{\nu}_i^* + \boldsymbol{\lambda}_i^*\big)^\top (\boldsymbol{z}_i^* - \boldsymbol{z}_i^{k+1}) = 0,$$

which yields

$$\big(\nabla f_i(\boldsymbol{x}_i^*) + \boldsymbol{y}_i^*\big)^\top (\boldsymbol{x}_i^* - \boldsymbol{x}_i^{k+1}) + \boldsymbol{\lambda}_i^{*\top}(\boldsymbol{z}_i^* - \boldsymbol{z}_i^{k+1}) + \boldsymbol{\nu}_i^{*\top}\big(\boldsymbol{A}_i(\boldsymbol{x}_i^* - \boldsymbol{x}_i^{k+1}) - (\boldsymbol{z}_i^* - \boldsymbol{z}_i^{k+1})\big) = 0. \tag{46}$$

Using (43g) and the fact that $\boldsymbol{A}_i\boldsymbol{x}_i^{k+1} - \boldsymbol{z}_i^{k+1} = \boldsymbol{0}$, we can then simplify (46) to (R5).

Subsequently, we proceed with proving (R6). The KKT conditions for the $(k+1)$-th update of $\boldsymbol{x}_i, \boldsymbol{z}_i$ are given by

Optimality for $\boldsymbol{x}_i$:      $\nabla f_i(\boldsymbol{x}_i^{k+1}) + \boldsymbol{A}_i^\top \boldsymbol{\nu}_i^{k+1} + \mu_i^k(\boldsymbol{x}_i^{k+1} - \boldsymbol{G}_i\boldsymbol{w}^k + \boldsymbol{y}_i^k/\mu_i^k) = \boldsymbol{0},$    (47a)

Optimality for $\boldsymbol{z}_i$:      $-\boldsymbol{\nu}_i^{k+1} + \rho_i^k(\boldsymbol{z}_i^{k+1} - \boldsymbol{s}_i^k + \boldsymbol{\lambda}_i^k/\rho_i^k) = \boldsymbol{0},$                (47b)

Constraints feasibility:      $\boldsymbol{A}_i\boldsymbol{x}_i^{k+1} = \boldsymbol{z}_i^{k+1}.$                             (47c)

From (47a), we have

$$\Big[\nabla f_i(\boldsymbol{x}_i^{k+1}) + \boldsymbol{A}_i^\top \boldsymbol{\nu}_i^{k+1} + \mu_i^k(\boldsymbol{x}_i^{k+1} - \boldsymbol{G}_i\boldsymbol{w}^k + \boldsymbol{y}_i^k/\mu_i^k)\Big]^\top (\boldsymbol{x}_i^{k+1} - \boldsymbol{x}_i^*) = 0. \tag{48}$$

We rewrite the term $\mu_i^k(\boldsymbol{x}_i^{k+1} - \boldsymbol{G}_i\boldsymbol{w}^k + \boldsymbol{y}_i^k/\mu_i^k)$ using (9) as follows

$$\mu_i^k(\boldsymbol{x}_i^{k+1} - \boldsymbol{G}_i\boldsymbol{w}^k + \boldsymbol{y}_i^k/\mu_i^k) =$$

$$= \mu_i^k\Big(\boldsymbol{x}_i^{k+1} - \boldsymbol{G}_i\boldsymbol{w}^k + \boldsymbol{y}_i^{k+1}/\mu_i^k - \big(\alpha^k\boldsymbol{x}_i^{k+1} + (1-\alpha^k)\boldsymbol{G}_i\boldsymbol{w}^k - \boldsymbol{G}_i\boldsymbol{w}^{k+1}\big)\Big)$$

$$= \boldsymbol{y}_i^{k+1} + \mu_i^k\Big(\boldsymbol{x}_i^{k+1} - \boldsymbol{G}_i\boldsymbol{w}^k - \alpha^k\boldsymbol{x}_i^{k+1} - (1-\alpha^k)\boldsymbol{G}_i\boldsymbol{w}^k + \boldsymbol{G}_i\boldsymbol{w}^{k+1}\Big)$$

$$= \boldsymbol{y}_i^{k+1} + \mu_i^k\Big((1-\alpha^k)\boldsymbol{x}_i^{k+1} - (2-\alpha^k)\boldsymbol{G}_i\boldsymbol{w}^k + \boldsymbol{G}_i\boldsymbol{w}^{k+1}\Big)$$

such that (48) is given as

$$\Big[\nabla f_i(\boldsymbol{x}_i^{k+1}) + \boldsymbol{A}_i^\top \boldsymbol{\nu}_i^{k+1} + \boldsymbol{y}_i^{k+1}$$

$$+ \mu_i^k\Big((1-\alpha^k)\boldsymbol{x}_i^{k+1} - (2-\alpha^k)\boldsymbol{G}_i\boldsymbol{w}^k + \boldsymbol{G}_i\boldsymbol{w}^{k+1}\Big)\Big]^\top (\boldsymbol{x}_i^{k+1} - \boldsymbol{x}_i^*) = 0. \tag{49}$$

Similarly, from (47b), we get

$$\Big[-\boldsymbol{\nu}_i^{k+1} + \rho_i^k(\boldsymbol{z}_i^{k+1} - \boldsymbol{s}_i^k + \boldsymbol{\lambda}_i^k/\rho_i^k)\Big]^\top (\boldsymbol{z}_i^{k+1} - \boldsymbol{z}_i^*) = 0. \tag{50}$$

We rewrite the term $\rho_i^k(\boldsymbol{z}_i^{k+1} - \boldsymbol{s}_i^k + \boldsymbol{\lambda}_i^k/\rho_i^k)$ using (8) as follows

$$\rho_i^k(\boldsymbol{z}_i^{k+1} - \boldsymbol{s}_i^k + \boldsymbol{\lambda}_i^k/\rho_i^k) = \rho_i^k\Big(\boldsymbol{z}_i^{k+1} - \boldsymbol{s}_i^k + \boldsymbol{\lambda}_i^{k+1}/\rho_i^k - \big(\alpha^k\boldsymbol{z}_i^{k+1} + (1-\alpha^k)\boldsymbol{s}_i^k - \boldsymbol{s}_i^{k+1}\big)\Big)$$

$$= \boldsymbol{\lambda}_i^{k+1} + \rho_i^k\big(\boldsymbol{z}_i^{k+1} - \boldsymbol{s}_i^k - \alpha^k\boldsymbol{z}_i^{k+1} - (1-\alpha^k)\boldsymbol{s}_i^k + \boldsymbol{s}_i^{k+1}\big)$$

$$= \boldsymbol{\lambda}_i^{k+1} + \rho_i^k\Big((1-\alpha^k)\boldsymbol{z}_i^{k+1} - (2-\alpha^k)\boldsymbol{s}_i^k + \boldsymbol{s}_i^{k+1}\Big),$$

such that (50) is given as

$$\Big[-\boldsymbol{\nu}_i^{k+1} + \boldsymbol{\lambda}_i^{k+1} + \rho_i^k\Big((1-\alpha^k)\boldsymbol{z}_i^{k+1} - (2-\alpha^k)\boldsymbol{s}_i^k + \boldsymbol{s}_i^{k+1}\Big)\Big]^\top (\boldsymbol{z}_i^{k+1} - \boldsymbol{z}_i^*) = 0. \tag{51}$$

Combining (43g), (49), (51) and the fact that $\boldsymbol{A}_i\boldsymbol{x}_i^{k+1} - \boldsymbol{z}_i^{k+1} = \boldsymbol{0}$, we obtain (R6).     $\square$

**Lemma 3.** *For all $i \in \mathcal{V}$, the following two relationships hold at every iteration $k$:*

$$\left(\boldsymbol{y}_i^{k+1} - \boldsymbol{y}_i^* + \mu_i^k\big((1-\alpha^k)\boldsymbol{x}_i^{k+1} - (2-\alpha^k)\boldsymbol{G}_i\boldsymbol{w}^k + \boldsymbol{G}_i\boldsymbol{w}^{k+1}\big)\right)^\top (\boldsymbol{x}_i^{k+1} - \boldsymbol{x}_i^*)$$

$$= \frac{1}{2\alpha^k\mu_i^k}\big(\|\boldsymbol{y}_i^{k+1} - \boldsymbol{y}_i^*\|^2 - \|\boldsymbol{y}_i^k - \boldsymbol{y}_i^*\|^2\big) + \frac{(2-\alpha^k)}{2(\alpha^k)^2\mu_i^k}\|\boldsymbol{y}_i^{k+1} - \boldsymbol{y}_i^k\|^2$$

$$+ \frac{(2-\alpha^k)\mu_i^k}{2(\alpha^k)^2}\|\boldsymbol{G}_i(\boldsymbol{w}^{k+1} - \boldsymbol{w}^k)\|^2 + \frac{\mu_i^k}{2\alpha^k}\big(\|\boldsymbol{G}_i(\boldsymbol{w}^{k+1} - \boldsymbol{w}^*)\|^2$$

$$- \|\boldsymbol{G}_i(\boldsymbol{w}^k - \boldsymbol{w}^*)\|^2\big) + \frac{1}{\alpha^k}(\boldsymbol{y}_i^{k+1} - \boldsymbol{y}_i^*)^\top\boldsymbol{G}_i\big(\boldsymbol{w}^{k+1} - (1-\alpha^k)\boldsymbol{w}^k - \alpha^k\boldsymbol{w}^*\big)$$

$$+ \frac{1}{(\alpha^k)^2}(\boldsymbol{y}_i^{k+1} - \boldsymbol{y}_i^k)^\top\boldsymbol{G}_i\big((2-\alpha^k)\boldsymbol{w}^{k+1} - (1+(1-\alpha^k)^2)\boldsymbol{w}^k - \alpha^k(1-\alpha^k)\boldsymbol{w}^*\big), \quad \text{(R7)}$$

$$\left(\boldsymbol{\lambda}_i^{k+1} - \boldsymbol{\lambda}_i^* + \rho_i^k\big((1-\alpha^k)\boldsymbol{z}_i^{k+1} - (2-\alpha^k)\boldsymbol{s}_i^k + \boldsymbol{s}_i^{k+1}\big)\right)^\top (\boldsymbol{z}_i^{k+1} - \boldsymbol{z}_i^*)$$

$$= \frac{1}{2\alpha^k\rho_i^k}\big(\|\boldsymbol{\lambda}_i^{k+1} - \boldsymbol{\lambda}_i^*\|^2 - \|\boldsymbol{\lambda}_i^k - \boldsymbol{\lambda}_i^*\|^2\big) + \frac{(2-\alpha^k)}{2(\alpha^k)^2\rho_i^k}\|\boldsymbol{\lambda}_i^{k+1} - \boldsymbol{\lambda}_i^k\|^2$$

$$+ \frac{\rho_i^k}{2\alpha^k}\big(\|\boldsymbol{s}_i^{k+1} - \boldsymbol{s}_i^*\|^2 - \|\boldsymbol{s}_i^k - \boldsymbol{s}_i^*\|^2\big) + \frac{(2-\alpha^k)\rho_i^k}{2(\alpha^k)^2}\|\boldsymbol{s}_i^{k+1} - \boldsymbol{s}_i^k\|^2$$

$$+ \frac{1}{\alpha^k}(\boldsymbol{\lambda}_i^{k+1} - \boldsymbol{\lambda}_i^*)^\top\big(-(1-\alpha^k)\boldsymbol{s}_i^k + \boldsymbol{s}_i^{k+1} - \alpha^k\boldsymbol{s}_i^*\big). \quad \text{(R8)}$$

*Proof.* Let us first simplify the individual terms of the LHS of (R7). For that, we start by rewriting the term $\boldsymbol{x}_i^{k+1} - \boldsymbol{x}_i^*$ as follows using (R2),

$$\boldsymbol{x}_i^{k+1} - \boldsymbol{x}_i^* = \frac{1}{\alpha^k}\left(\frac{1}{\mu_i^k}(\boldsymbol{y}_i^{k+1} - \boldsymbol{y}_i^k) - (1-\alpha^k)\boldsymbol{G}_i\boldsymbol{w}^k + \boldsymbol{G}_i\boldsymbol{w}^{k+1} - \alpha^k\boldsymbol{x}_i^*\right).$$

Using (43d), we can rewrite the above as

$$\boldsymbol{x}_i^{k+1} - \boldsymbol{x}_i^* = \frac{1}{\alpha^k}\left(\frac{1}{\mu_i^k}(\boldsymbol{y}_i^{k+1} - \boldsymbol{y}_i^k) - (1-\alpha^k)\boldsymbol{G}_i\boldsymbol{w}^k + \boldsymbol{G}_i\boldsymbol{w}^{k+1} - \alpha^k\boldsymbol{G}_i\boldsymbol{w}^*\right) \quad (52)$$

which can be simplified to

$$\boldsymbol{x}_i^{k+1} - \boldsymbol{x}_i^* = \frac{1}{\alpha^k\mu_i^k}(\boldsymbol{y}_i^{k+1} - \boldsymbol{y}_i^k) + \frac{1}{\alpha^k}\boldsymbol{G}_i\big(\boldsymbol{w}^{k+1} - (1-\alpha^k)\boldsymbol{w}^k - \alpha^k\boldsymbol{w}^*\big). \quad (53)$$

Let us now simplify the following term in the LHS of the relationship (R7)

$$(1-\alpha^k)\boldsymbol{x}_i^{k+1} - (2-\alpha^k)\boldsymbol{G}_i\boldsymbol{w}^k + \boldsymbol{G}_i\boldsymbol{w}^{k+1} = (1-\alpha^k)(\boldsymbol{x}_i^{k+1} - \boldsymbol{G}_i\boldsymbol{w}^k) + \boldsymbol{G}_i(\boldsymbol{w}^{k+1} - \boldsymbol{w}^k). \quad (54)$$

Using (R2), we have

$$\boldsymbol{x}_i^{k+1} - \boldsymbol{G}_i\boldsymbol{w}^k = \frac{1}{\alpha^k}\left(\frac{1}{\mu_i^k}(\boldsymbol{y}_i^{k+1} - \boldsymbol{y}_i^k) - (1-\alpha^k)\boldsymbol{G}_i\boldsymbol{w}^k + \boldsymbol{G}_i\boldsymbol{w}^{k+1}\right) - \boldsymbol{G}_i\boldsymbol{w}^k,$$

which can be written as

$$\boldsymbol{x}_i^{k+1} - \boldsymbol{G}_i\boldsymbol{w}^k = \frac{1}{\mu_i^k\alpha^k}(\boldsymbol{y}_i^{k+1} - \boldsymbol{y}_i^k) + \frac{1}{\alpha^k}\boldsymbol{G}_i(\boldsymbol{w}^{k+1} - \boldsymbol{w}^k). \quad (55)$$

Substituting (55) in (54), we get

$$(1-\alpha^k)\boldsymbol{x}_i^{k+1} - (2-\alpha^k)\boldsymbol{G}_i\boldsymbol{w}^k + \boldsymbol{G}_i\boldsymbol{w}^{k+1} = \frac{(1-\alpha^k)}{\mu_i^k\alpha^k}(\boldsymbol{y}_i^{k+1} - \boldsymbol{y}_i^k) + \frac{1}{\alpha^k}\boldsymbol{G}_i(\boldsymbol{w}^{k+1} - \boldsymbol{w}^k).$$

Using the above result, we then rewrite the following term on the LHS of (R7) as

$$\boldsymbol{y}_i^{k+1} - \boldsymbol{y}_i^* + \mu_i^k\big((1-\alpha^k)\boldsymbol{x}_i^{k+1} - (2-\alpha^k)\boldsymbol{G}_i\boldsymbol{w}^k + \boldsymbol{G}_i\boldsymbol{w}^{k+1}\big)$$

$$= \boldsymbol{y}_i^{k+1} - \boldsymbol{y}_i^* + \frac{(1-\alpha^k)}{\alpha^k}(\boldsymbol{y}_i^{k+1} - \boldsymbol{y}_i^k) + \frac{\mu_i^k}{\alpha^k}\boldsymbol{G}_i(\boldsymbol{w}^{k+1} - \boldsymbol{w}^k). \quad (56)$$

For notational simplicity, let us consider the LHS of (R7) as LHS(R7). Using (56) and (53), we get

$$\text{LHS(R7)} = \left(\boldsymbol{y}_i^{k+1} - \boldsymbol{y}_i^* + \frac{(1-\alpha^k)}{\alpha^k}(\boldsymbol{y}_i^{k+1} - \boldsymbol{y}_i^k) + \frac{\mu_i^k}{\alpha^k}\boldsymbol{G}_i(\boldsymbol{w}^{k+1} - \boldsymbol{w}^k)\right)^\top$$
$$\left(\frac{1}{\alpha^k\mu_i^k}(\boldsymbol{y}_i^{k+1} - \boldsymbol{y}_i^k) + \frac{1}{\alpha^k}\boldsymbol{G}_i(\boldsymbol{w}^{k+1} - (1-\alpha^k)\boldsymbol{w}^k - \alpha^k\boldsymbol{w}^*)\right)$$

which can be further rewritten as

$$\text{LHS(R7)} = \frac{1}{\alpha^k\mu_i^k}(\boldsymbol{y}_i^{k+1} - \boldsymbol{y}_i^*)^\top(\boldsymbol{y}_i^{k+1} - \boldsymbol{y}_i^k) + \frac{1}{\alpha^k}(\boldsymbol{y}_i^{k+1} - \boldsymbol{y}_i^*)^\top\boldsymbol{G}_i(\boldsymbol{w}^{k+1} - (1-\alpha^k)\boldsymbol{w}^k$$
$$- \alpha^k\boldsymbol{w}^*) + \frac{(1-\alpha^k)}{(\alpha^k)^2\mu_i^k}\|\boldsymbol{y}_i^{k+1} - \boldsymbol{y}_i^k\|^2 + \frac{(1-\alpha^k)}{(\alpha^k)^2}(\boldsymbol{y}_i^{k+1} - \boldsymbol{y}_i^k)^\top\boldsymbol{G}_i(\boldsymbol{w}^{k+1}$$
$$- (1-\alpha^k)\boldsymbol{w}^k - \alpha^k\boldsymbol{w}^*) + \frac{1}{(\alpha^k)^2}(\boldsymbol{w}^{k+1} - \boldsymbol{w}^k)^\top\boldsymbol{G}_i^\top(\boldsymbol{y}_i^{k+1} - \boldsymbol{y}_i^k)$$
$$+ \frac{\mu_i^k}{(\alpha^k)^2}\left(\boldsymbol{G}_i(\boldsymbol{w}^{k+1} - \boldsymbol{w}^k)\right)^\top\boldsymbol{G}_i(\boldsymbol{w}^{k+1} - (1-\alpha^k)\boldsymbol{w}^k - \alpha^k\boldsymbol{w}^*). \tag{57}$$

Let us now simplify each term on the RHS of the above equation. We start with the terms including only the variables $\boldsymbol{y}_i^{k+1}$, $\boldsymbol{y}_i^k$ and $\boldsymbol{y}_i^*$. Using the fact that $a^\top b = \frac{1}{2}(\|a\|^2 + \|b\|^2 - \|a-b\|^2)$, we get

$$\frac{1}{\alpha^k\mu_i^k}(\boldsymbol{y}_i^{k+1} - \boldsymbol{y}_i^*)^\top(\boldsymbol{y}_i^{k+1} - \boldsymbol{y}_i^k) = \frac{1}{2\alpha^k\mu_i^k}\left(\|\boldsymbol{y}_i^{k+1} - \boldsymbol{y}_i^*\|^2 + \|\boldsymbol{y}_i^{k+1} - \boldsymbol{y}_i^k\|^2 - \|\boldsymbol{y}_i^k - \boldsymbol{y}_i^*\|^2\right).$$

Using the above result, we can write

$$\frac{1}{\alpha^k\mu_i^k}(\boldsymbol{y}_i^{k+1} - \boldsymbol{y}_i^*)^\top(\boldsymbol{y}_i^{k+1} - \boldsymbol{y}_i^k) + \frac{(1-\alpha^k)}{(\alpha^k)^2\mu_i^k}\|\boldsymbol{y}_i^{k+1} - \boldsymbol{y}_i^k\|^2$$
$$= \frac{1}{2\alpha^k\mu_i^k}\left(\|\boldsymbol{y}_i^{k+1} - \boldsymbol{y}_i^*\|^2 + \|\boldsymbol{y}_i^{k+1} - \boldsymbol{y}_i^k\|^2 - \|\boldsymbol{y}_i^k - \boldsymbol{y}_i^*\|^2\right) + \frac{(1-\alpha^k)}{(\alpha^k)^2\mu_i^k}\|\boldsymbol{y}_i^{k+1} - \boldsymbol{y}_i^k\|^2$$
$$= \frac{1}{2\alpha^k\mu_i^k}\left(\|\boldsymbol{y}_i^{k+1} - \boldsymbol{y}_i^*\|^2 - \|\boldsymbol{y}_i^k - \boldsymbol{y}_i^*\|^2\right) + \frac{(2-\alpha^k)}{2(\alpha^k)^2\mu_i^k}\|\boldsymbol{y}_i^{k+1} - \boldsymbol{y}_i^k\|^2. \tag{58}$$

Next, we consider the following terms in the RHS of (57) involving only the variables $\boldsymbol{w}^{k+1}, \boldsymbol{w}^k$ and $\boldsymbol{w}^*$,

$$\frac{\mu_i^k}{(\alpha^k)^2}\left(\boldsymbol{G}_i(\boldsymbol{w}^{k+1} - \boldsymbol{w}^k)\right)^\top\boldsymbol{G}_i(\boldsymbol{w}^{k+1} - (1-\alpha^k)\boldsymbol{w}^k - \alpha^k\boldsymbol{w}^*)$$
$$= \frac{(1-\alpha^k)\mu_i^k}{(\alpha^k)^2}\|\boldsymbol{G}_i(\boldsymbol{w}^{k+1} - \boldsymbol{w}^k)\|^2 + \frac{\mu_i^k}{\alpha^k}\left(\boldsymbol{G}_i(\boldsymbol{w}^{k+1} - \boldsymbol{w}^k)\right)^\top\left(\boldsymbol{G}_i(\boldsymbol{w}^{k+1} - \boldsymbol{w}^*)\right). \tag{59}$$

Using a similar approach as used to derive (58), we obtain

$$\frac{(1-\alpha^k)\mu_i^k}{(\alpha^k)^2}\|\boldsymbol{G}_i(\boldsymbol{w}^{k+1} - \boldsymbol{w}^k)\|^2 + \frac{\mu_i^k}{\alpha^k}\left(\boldsymbol{G}_i(\boldsymbol{w}^{k+1} - \boldsymbol{w}^k)\right)^\top\left(\boldsymbol{G}_i(\boldsymbol{w}^{k+1} - \boldsymbol{w}^*)\right)$$
$$= \frac{(2-\alpha^k)\mu_i^k}{2(\alpha^k)^2}\|\boldsymbol{G}_i(\boldsymbol{w}^{k+1} - \boldsymbol{w}^k)\|^2 + \frac{\mu_i^k}{2\alpha^k}(\|\boldsymbol{G}_i(\boldsymbol{w}^{k+1} - \boldsymbol{w}^*)\|^2 - \|\boldsymbol{G}_i(\boldsymbol{w}^k - \boldsymbol{w}^*)\|^2). \tag{60}$$

Now, let us consider the following terms from the RHS of (57),

$$\frac{(1-\alpha^k)}{(\alpha^k)^2}(\boldsymbol{y}_i^{k+1} - \boldsymbol{y}_i^k)^\top\boldsymbol{G}_i(\boldsymbol{w}^{k+1} - (1-\alpha^k)\boldsymbol{w}^k - \alpha^k\boldsymbol{w}^*)$$
$$+ \frac{1}{(\alpha^k)^2}(\boldsymbol{w}^{k+1} - \boldsymbol{w}^k)^\top\boldsymbol{G}_i^\top(\boldsymbol{y}_i^{k+1} - \boldsymbol{y}_i^k)$$
$$= \frac{1}{(\alpha^k)^2}(\boldsymbol{y}_i^{k+1} - \boldsymbol{y}_i^k)^\top\boldsymbol{G}_i\left((1-\alpha^k)\boldsymbol{w}^{k+1} - (1-\alpha^k)^2\boldsymbol{w}^k - \alpha^k(1-\alpha^k)\boldsymbol{w}^* + \boldsymbol{w}^{k+1} - \boldsymbol{w}^k\right)$$
$$= \frac{1}{(\alpha^k)^2}(\boldsymbol{y}_i^{k+1} - \boldsymbol{y}_i^k)^\top\boldsymbol{G}_i\left((2-\alpha^k)\boldsymbol{w}^{k+1} - (1+(1-\alpha^k)^2)\boldsymbol{w}^k - \alpha^k(1-\alpha^k)\boldsymbol{w}^*\right). \tag{61}$$

Substituting (58), (59), (60) and (61) into (57), we get

$$
\text{LHS(R7)} = \frac{1}{2\alpha^k \mu_i^k} \big( \|\boldsymbol{y}_i^{k+1} - \boldsymbol{y}_i^*\|^2 - \|\boldsymbol{y}_i^k - \boldsymbol{y}_i^*\|^2 \big) + \frac{(2 - \alpha^k)}{2(\alpha^k)^2 \mu_i^k} \|\boldsymbol{y}_i^{k+1} - \boldsymbol{y}_i^k\|^2
$$

$$
+ \frac{(2 - \alpha^k)\mu_i^k}{2(\alpha^k)^2} \|\boldsymbol{G}_i(\boldsymbol{w}^{k+1} - \boldsymbol{w}^k)\|^2 + \frac{\mu_i^k}{2\alpha^k} \big( \|\boldsymbol{G}_i(\boldsymbol{w}^{k+1} - \boldsymbol{w}^*)\|^2
$$

$$
- \|\boldsymbol{G}_i(\boldsymbol{w}^k - \boldsymbol{w}^*)\|^2 \big) + \frac{1}{\alpha^k}(\boldsymbol{y}_i^{k+1} - \boldsymbol{y}_i^*)^\top \boldsymbol{G}_i\big(\boldsymbol{w}^{k+1} - (1 - \alpha^k)\boldsymbol{w}^k - \alpha^k \boldsymbol{w}^*\big)
$$

$$
+ \frac{1}{(\alpha^k)^2}(\boldsymbol{y}_i^{k+1} - \boldsymbol{y}_i^k)^\top \boldsymbol{G}_i\big((2 - \alpha^k)\boldsymbol{w}^{k+1} - (1 + (1 - \alpha^k)^2)\boldsymbol{w}^k - \alpha^k(1 - \alpha^k)\boldsymbol{w}^*\big) \tag{62}
$$

which proves (R7).

Subsequently, we prove the relationship (R8). Using similar steps as for (R7), we get

$$
\left( \boldsymbol{\lambda}_i^{k+1} - \boldsymbol{\lambda}_i^* + \rho_i^k\big((1 - \alpha^k)\boldsymbol{z}_i^{k+1} - (2 - \alpha^k)\boldsymbol{s}_i^k + \boldsymbol{s}_i^{k+1}\big) \right)^\top (\boldsymbol{z}_i^{k+1} - \boldsymbol{z}_i^*)
$$

$$
= \frac{1}{2\alpha^k \rho_i^k}\big( \|\boldsymbol{\lambda}_i^{k+1} - \boldsymbol{\lambda}_i^*\|^2 - \|\boldsymbol{\lambda}_i^k - \boldsymbol{\lambda}_i^*\|^2 \big) + \frac{(2 - \alpha^k)}{2(\alpha^k)^2 \rho_i^k} \|\boldsymbol{\lambda}_i^{k+1} - \boldsymbol{\lambda}_i^k\|^2
$$

$$
+ \frac{\rho_i^k}{2\alpha^k}\big( \|\boldsymbol{s}_i^{k+1} - \boldsymbol{s}_i^*\|^2 - \|\boldsymbol{s}_i^k - \boldsymbol{s}_i^*\|^2 \big) + \frac{(2 - \alpha^k)\rho_i^k}{2(\alpha^k)^2} \|\boldsymbol{s}_i^{k+1} - \boldsymbol{s}_i^k\|^2 \tag{63}
$$

$$
+ \frac{1}{\alpha^k}(\boldsymbol{\lambda}_i^{k+1} - \boldsymbol{\lambda}_i^*)^\top(\boldsymbol{s}_i^{k+1} - (1 - \alpha^k)\boldsymbol{s}_i^k - \alpha^k \boldsymbol{s}_i^*)
$$

$$
+ \frac{1}{(\alpha^k)^2}(\boldsymbol{\lambda}_i^{k+1} - \boldsymbol{\lambda}_i^k)^\top\big((2 - \alpha^k)\boldsymbol{s}_i^{k+1} - (1 + (1 - \alpha^k)^2)\boldsymbol{s}_i^k - \alpha^k(1 - \alpha^k)\boldsymbol{s}_i^*\big).
$$

Let us now simplify the last term of the RHS of the above equation as follows

$$
(\boldsymbol{\lambda}_i^{k+1} - \boldsymbol{\lambda}_i^k)^\top\big((2 - \alpha^k)\boldsymbol{s}_i^{k+1} - (1 + (1 - \alpha^k)^2)\boldsymbol{s}_i^k - \alpha^k(1 - \alpha^k)\boldsymbol{s}_i^*\big)
$$

$$
= (1 + (1 - \alpha^k)^2)(\boldsymbol{\lambda}_i^{k+1} - \boldsymbol{\lambda}_i^k)^\top(\boldsymbol{s}_i^{k+1} - \boldsymbol{s}_i^k) + \alpha^k(1 - \alpha^k)(\boldsymbol{\lambda}_i^{k+1} - \boldsymbol{\lambda}_i^k)^\top(\boldsymbol{s}_i^{k+1} - \boldsymbol{s}_i^*). \tag{64}
$$

From (6) and (43h), we have that the vectors $\boldsymbol{s}_i^k, \boldsymbol{s}_i^{k+1}, \boldsymbol{s}_i^* \in \mathcal{C}_i$. Using (R4), the above equation gives us

$$
(\boldsymbol{\lambda}_i^{k+1} - \boldsymbol{\lambda}_i^k)^\top\big((2 - \alpha^k)\boldsymbol{s}_i^{k+1} - (1 + (1 - \alpha^k)^2)\boldsymbol{s}_i^k - \alpha^k(1 - \alpha^k)\boldsymbol{s}_i^*\big)
$$

$$
= (\boldsymbol{\lambda}_i^{k+1} - \boldsymbol{\lambda}_i^k)^\top\big((2 - \alpha^k)\boldsymbol{s}_i^{k+1} - (2 + (\alpha^k)^2 - 2\alpha^k)\boldsymbol{s}_i^k + (-\alpha^k + (\alpha^k)^2)\boldsymbol{s}_i^*\big) = 0.
$$

It follows that (63) then simplifies to (R8). $\qquad\square$

**Lemma 4.** *The following inequality holds true at every iteration $k$:*

$$
\sum_{i \in \mathcal{V}} \left( \frac{1}{\mu_i^k}\big( \|\boldsymbol{y}_i^{k+1} - \boldsymbol{y}_i^*\|^2 - \|\boldsymbol{y}_i^k - \boldsymbol{y}_i^*\|^2 \big) + \mu_i^k\big( \|\boldsymbol{G}_i(\boldsymbol{w}^{k+1} - \boldsymbol{w}^*)\|^2 - \|\boldsymbol{G}_i(\boldsymbol{w}^k - \boldsymbol{w}^*)\|^2 \big) \right.
$$

$$
\left. + \frac{1}{\rho_i^k}\big( \|\boldsymbol{\lambda}_i^{k+1} - \boldsymbol{\lambda}_i^*\|^2 - \|\boldsymbol{\lambda}_i^k - \boldsymbol{\lambda}_i^*\|^2 \big) + \rho_i^k\big( \|\boldsymbol{s}_i^{k+1} - \boldsymbol{s}_i^*\|^2 - \|\boldsymbol{s}_i^k - \boldsymbol{s}_i^*\|^2 \big) \right)
$$

$$
\leq -\frac{(2 - \alpha^k)}{\alpha^k} \sum_{i \in \mathcal{V}} \left( \frac{1}{\mu_i^k}\|\boldsymbol{y}_i^{k+1} - \boldsymbol{y}_i^k\|^2 + \mu_i^k\|\boldsymbol{G}_i(\boldsymbol{w}^{k+1} - \boldsymbol{w}^k)\|^2 + \frac{1}{\rho_i^k}\|\boldsymbol{\lambda}_i^{k+1} - \boldsymbol{\lambda}_i^k\|^2 \right. \tag{65}
$$

$$
\left. + \rho_i^k\|\boldsymbol{s}_i^{k+1} - \boldsymbol{s}_i^k\|^2 \right).
$$

*Proof.* We start by combining the relationships (R5) and (R6) to get

$$
\left( \boldsymbol{y}_i^{k+1} - \boldsymbol{y}_i^* + \mu_i^k\big((1 - \alpha^k)\boldsymbol{x}_i^{k+1} - (2 - \alpha^k)\boldsymbol{G}_i\boldsymbol{w}^k + \boldsymbol{G}_i\boldsymbol{w}^{k+1}\big) \right)^\top (\boldsymbol{x}_i^{k+1} - \boldsymbol{x}_i^*)
$$

$$
+ \left( \boldsymbol{\lambda}_i^{k+1} - \boldsymbol{\lambda}_i^* + \rho_i^k\big((1 - \alpha^k)\boldsymbol{z}_i^{k+1} - (2 - \alpha^k)\boldsymbol{s}_i^k + \boldsymbol{s}_i^{k+1}\big) \right)^\top (\boldsymbol{z}_i^{k+1} - \boldsymbol{z}_i^*) \tag{66}
$$

$$
= -(\nabla f_i(\boldsymbol{x}_i^{k+1}) - \nabla f_i(\boldsymbol{x}_i^*))^\top(\boldsymbol{x}_i^{k+1} - \boldsymbol{x}_i^*).
$$

Since $f_i$ is convex, then we have $(\nabla f_i(\boldsymbol{x}_i^{k+1}) - \nabla f_i(\boldsymbol{x}_i^*))^\top (\boldsymbol{x}_i^{k+1} - \boldsymbol{x}_i^*) \geq 0$, which gives

$$
\begin{aligned}
&\left( \boldsymbol{y}_i^{k+1} - \boldsymbol{y}_i^* + \mu_i^k \big((1-\alpha^k)\boldsymbol{x}_i^{k+1} - (2-\alpha^k)\boldsymbol{G}_i\boldsymbol{w}^k + \boldsymbol{G}_i\boldsymbol{w}^{k+1}\big) \right)^\top (\boldsymbol{x}_i^{k+1} - \boldsymbol{x}_i^*) \\
&+ \left( \boldsymbol{\lambda}_i^{k+1} - \boldsymbol{\lambda}_i^* + \rho_i^k \big((1-\alpha^k)\boldsymbol{z}_i^{k+1} - (2-\alpha^k)\boldsymbol{s}_i^k + \boldsymbol{s}_i^{k+1}\big) \right)^\top (\boldsymbol{z}_i^{k+1} - \boldsymbol{z}_i^*) \leq 0.
\end{aligned}
\tag{67}
$$

Summing (67) over all $i \in \mathcal{V}$, we get

$$
\begin{aligned}
&\sum_{i \in \mathcal{V}} \left( \boldsymbol{y}_i^{k+1} - \boldsymbol{y}_i^* + \mu_i^k \big((1-\alpha^k)\boldsymbol{x}_i^{k+1} - (2-\alpha^k)\boldsymbol{G}_i\boldsymbol{w}^k + \boldsymbol{G}_i\boldsymbol{w}^{k+1}\big) \right)^\top (\boldsymbol{x}_i^{k+1} - \boldsymbol{x}_i^*) \\
&+ \sum_{i \in \mathcal{V}} \left( \boldsymbol{\lambda}_i^{k+1} - \boldsymbol{\lambda}_i^* + \rho_i^k \big((1-\alpha^k)\boldsymbol{z}_i^{k+1} - (2-\alpha^k)\boldsymbol{s}_i^k + \boldsymbol{s}_i^{k+1}\big) \right)^\top (\boldsymbol{z}_i^{k+1} - \boldsymbol{z}_i^*) \leq 0.
\end{aligned}
\tag{68}
$$

Now, we use the relationships (R7) and (R8) to rewrite the above inequality as

$$
\begin{aligned}
0 \geq \sum_{i \in \mathcal{V}} \Bigg( & \frac{1}{2\alpha^k \mu_i^k} \big( \|\boldsymbol{y}_i^{k+1} - \boldsymbol{y}_i^*\|^2 - \|\boldsymbol{y}_i^k - \boldsymbol{y}_i^*\|^2 \big) + \frac{(2-\alpha^k)}{2(\alpha^k)^2 \mu_i^k} \|\boldsymbol{y}_i^{k+1} - \boldsymbol{y}_i^k\|^2 \\
& + \frac{(2-\alpha^k)\mu_i^k}{2(\alpha^k)^2} \|\boldsymbol{G}_i(\boldsymbol{w}^{k+1} - \boldsymbol{w}^k)\|^2 + \frac{\mu_i^k}{2\alpha^k} \big( \|\boldsymbol{G}_i(\boldsymbol{w}^{k+1} - \boldsymbol{w}^*)\|^2 \\
& - \|\boldsymbol{G}_i(\boldsymbol{w}^k - \boldsymbol{w}^*)\|^2 \big) + \frac{1}{\alpha^k}(\boldsymbol{y}_i^{k+1} - \boldsymbol{y}_i^*)^\top \boldsymbol{G}_i \big( \boldsymbol{w}^{k+1} - (1-\alpha^k)\boldsymbol{w}^k - \alpha^k\boldsymbol{w}^* \big) \\
& + \frac{1}{(\alpha^k)^2}(\boldsymbol{y}_i^{k+1} - \boldsymbol{y}_i^k)^\top \boldsymbol{G}_i \big( (2-\alpha^k)\boldsymbol{w}^{k+1} - (1+(1-\alpha^k)^2)\boldsymbol{w}^k - \alpha^k(1-\alpha^k)\boldsymbol{w}^* \big) \\
& + \frac{1}{2\alpha^k \rho_i^k} \big( \|\boldsymbol{\lambda}_i^{k+1} - \boldsymbol{\lambda}_i^*\|^2 - \|\boldsymbol{\lambda}_i^k - \boldsymbol{\lambda}_i^*\|^2 \big) + \frac{(2-\alpha^k)}{2(\alpha^k)^2 \rho_i^k} \|\boldsymbol{\lambda}_i^{k+1} - \boldsymbol{\lambda}_i^k\|^2 \\
& + \frac{\rho_i^k}{2\alpha^k} \big( \|\boldsymbol{s}_i^{k+1} - \boldsymbol{s}_i^*\|^2 - \|\boldsymbol{s}_i^k - \boldsymbol{s}_i^*\|^2 \big) + \frac{(2-\alpha^k)\rho_i^k}{2(\alpha^k)^2} \|\boldsymbol{s}_i^{k+1} - \boldsymbol{s}_i^k\|^2 \\
& + \frac{1}{\alpha^k}(\boldsymbol{\lambda}_i^{k+1} - \boldsymbol{\lambda}_i^*)^\top \big( -(1-\alpha^k)\boldsymbol{s}_i^k + \boldsymbol{s}_i^{k+1} - \alpha^k\boldsymbol{s}_i^* \big) \Bigg).
\end{aligned}
\tag{69}
$$

Let us now further simplify the terms on the RHS of the above equation. For that, let us start with the last term on the RHS. We have

$$
\begin{aligned}
(\boldsymbol{\lambda}_i^{k+1} - \boldsymbol{\lambda}_i^*)^\top(-(1-\alpha^k)\boldsymbol{s}_i^k + \boldsymbol{s}_i^{k+1} - \alpha^k\boldsymbol{s}_i^*) &= (\boldsymbol{\lambda}_i^{k+1} - \boldsymbol{\lambda}_i^*)^\top(\boldsymbol{s}_i^{k+1} - \boldsymbol{s}_i^*) \\
&\quad - (1-\alpha^k)(\boldsymbol{\lambda}_i^{k+1} - \boldsymbol{\lambda}_i^*)^\top(\boldsymbol{s}_i^k - \boldsymbol{s}_i^*).
\end{aligned}
\tag{70}
$$

Using (R4), (43c), and the fact that $\boldsymbol{s}_i^k, \boldsymbol{s}_i^{k+1}, \boldsymbol{s}_i^* \in \mathcal{C}_i$, we get

$$
(\boldsymbol{\lambda}_i^{k+1} - \boldsymbol{\lambda}_i^*)^\top(\boldsymbol{s}_i^{k+1} - \boldsymbol{s}_i^*) \geq 0,
\tag{71}
$$

$$
(\boldsymbol{\lambda}_i^{k+1} - \boldsymbol{\lambda}_i^*)^\top(\boldsymbol{s}_i^k - \boldsymbol{s}_i^*) \geq 0.
\tag{72}
$$

Thus, for $\alpha^k \geq 1$, combining (70), (71), and (72), we get

$$
(\boldsymbol{\lambda}_i^{k+1} - \boldsymbol{\lambda}_i^*)^\top(-(1-\alpha^k)\boldsymbol{s}_i^k + \boldsymbol{s}_i^{k+1} - \alpha^k\boldsymbol{s}_i^*) \geq 0.
\tag{73}
$$

Now, the following results hold based on the relationship (R1) and (43d),

$$
\sum_{i \in \mathcal{V}}(\boldsymbol{y}_i^{k+1} - \boldsymbol{y}_i^*)^\top \boldsymbol{G}_i = 0, \quad \sum_{i \in \mathcal{V}}(\boldsymbol{y}_i^{k+1} - \boldsymbol{y}_i^k)^\top \boldsymbol{G}_i = 0.
\tag{74}
$$

By substituting (73) and (74) in (69), and by rearranging the terms, we get

$$
\sum_{i \in \mathcal{V}} \left( \frac{1}{2\alpha^k \mu_i^k} (\|\boldsymbol{y}_i^{k+1} - \boldsymbol{y}_i^*\|^2 - \|\boldsymbol{y}_i^k - \boldsymbol{y}_i^*\|^2) + \frac{\mu_i^k}{2\alpha^k} (\|\boldsymbol{G}_i(\boldsymbol{w}^{k+1} - \boldsymbol{w}^*)\|^2 - \|\boldsymbol{G}_i(\boldsymbol{w}^k - \boldsymbol{w}^*)\|^2) \right.
$$

$$
\left. + \frac{1}{2\alpha^k \rho_i^k} (\|\boldsymbol{\lambda}_i^{k+1} - \boldsymbol{\lambda}_i^*\|^2 - \|\boldsymbol{\lambda}_i^k - \boldsymbol{\lambda}_i^*\|^2) + \frac{\rho_i^k}{2\alpha^k} (\|\boldsymbol{s}_i^{k+1} - \boldsymbol{s}_i^*\|^2 - \|\boldsymbol{s}_i^k - \boldsymbol{s}_i^*\|^2) \right)
$$

$$
\leq -\sum_{i \in \mathcal{V}} \left( \frac{(2 - \alpha^k)}{2(\alpha^k)^2 \mu_i^k} \|\boldsymbol{y}_i^{k+1} - \boldsymbol{y}_i^k\|^2 + \frac{(2 - \alpha^k)\mu_i^k}{2(\alpha^k)^2} \|\boldsymbol{G}_i(\boldsymbol{w}^{k+1} - \boldsymbol{w}^k)\|^2 \right.
$$

$$
\left. + \frac{(2 - \alpha^k)}{2(\alpha^k)^2 \rho_i^k} \|\boldsymbol{\lambda}_i^{k+1} - \boldsymbol{\lambda}_i^k\|^2 + \frac{(2 - \alpha^k)\rho_i^k}{2(\alpha^k)^2} \|\boldsymbol{s}_i^{k+1} - \boldsymbol{s}_i^k\|^2 \right).
$$

Since, $\alpha^k \geq 1$, we can multiply the above equation with $2\alpha^k$ to obtain (65). $\qquad \square$

### D.3 Proof of Theorem 1

Let us first rearrange the inequality (65) derived in Lemma 4, as

$$
\frac{(2 - \alpha^k)}{\alpha^k} \sum_{i \in \mathcal{V}} \left( \frac{1}{\mu_i^k} \|\boldsymbol{y}_i^{k+1} - \boldsymbol{y}_i^k\|^2 + \mu_i^k \|\boldsymbol{G}_i(\boldsymbol{w}^{k+1} - \boldsymbol{w}^k)\|^2 + \frac{1}{\rho_i^k} \|\boldsymbol{\lambda}_i^{k+1} - \boldsymbol{\lambda}_i^k\|^2 + \rho_i^k \|\boldsymbol{s}_i^{k+1} - \boldsymbol{s}_i^k\|^2 \right)
$$

$$
\leq \sum_{i \in \mathcal{V}} \left( \frac{1}{\mu_i^k} (\|\boldsymbol{y}_i^k - \boldsymbol{y}_i^*\|^2 - \|\boldsymbol{y}_i^{k+1} - \boldsymbol{y}_i^*\|^2) + \mu_i^k (\|\boldsymbol{G}_i(\boldsymbol{w}^k - \boldsymbol{w}^*)\|^2 - \|\boldsymbol{G}_i(\boldsymbol{w}^{k+1} - \boldsymbol{w}^*)\|^2) \right.
$$

$$
\left. + \frac{1}{\rho_i^k} (\|\boldsymbol{\lambda}_i^k - \boldsymbol{\lambda}_i^*\|^2 - \|\boldsymbol{\lambda}_i^{k+1} - \boldsymbol{\lambda}_i^*\|^2) + \rho_i^k (\|\boldsymbol{s}_i^k - \boldsymbol{s}_i^*\|^2 - \|\boldsymbol{s}_i^{k+1} - \boldsymbol{s}_i^*\|^2) \right).
\tag{75}
$$

For convenience, let us define for each iteration $k$, the terms $\eta_i^k$, $i \in \mathcal{V}$, and $\eta^k$ such that

$$
\eta_i^k + 1 = \max \left( \frac{\rho_i^k}{\rho_i^{k-1}}, \frac{\rho_i^{k-1}}{\rho_i^k}, \frac{\mu_i^k}{\mu_i^{k-1}}, \frac{\mu_i^{k-1}}{\mu_i^k} \right), \quad \eta_{\max}^k = \max_{i \in \mathcal{V}} \eta_i^k,
$$

and the term $V^k$ as

$$
V^k = \sum_{i \in \mathcal{V}} \left( \frac{1}{\mu_i^{k-1}} \|\boldsymbol{y}_i^k - \boldsymbol{y}_i^*\|^2 + \mu_i^{k-1} \|\boldsymbol{G}_i(\boldsymbol{w}^k - \boldsymbol{w}^*)\|^2 + \frac{1}{\rho_i^{k-1}} \|\boldsymbol{\lambda}_i^k - \boldsymbol{\lambda}_i^*\|^2 + \rho_i^{k-1} \|\boldsymbol{s}_i^k - \boldsymbol{s}_i^*\|^2 \right).
$$

Based on the definition of $\eta_i^k$, we can write

$$
\frac{1}{\mu_i^k} \|\boldsymbol{y}_i^k - \boldsymbol{y}_i^*\|^2 + \mu_i^k \|\boldsymbol{G}_i(\boldsymbol{w}^k - \boldsymbol{w}^*)\|^2 + \frac{1}{\rho_i^k} \|\boldsymbol{\lambda}_i^k - \boldsymbol{\lambda}_i^*\|^2 + \rho_i^k \|\boldsymbol{s}_i^k - \boldsymbol{s}_i^*\|^2
$$

$$
\leq (\eta_i^k + 1) \left( \frac{1}{\mu_i^{k-1}} \|\boldsymbol{y}_i^k - \boldsymbol{y}_i^*\|^2 + \mu_i^{k-1} \|\boldsymbol{G}_i(\boldsymbol{w}^k - \boldsymbol{w}^*)\|^2 + \frac{1}{\rho_i^{k-1}} \|\boldsymbol{\lambda}_i^k - \boldsymbol{\lambda}_i^*\|^2 + \rho_i^{k-1} \|\boldsymbol{s}_i^k - \boldsymbol{s}_i^*\|^2 \right).
$$

By adding the above result over all $i \in \mathcal{V}$, and using the fact that $\eta_{\max}^k \geq \eta_i^k$ for all $i$, we get

$$
\sum_{i \in \mathcal{V}} \left( \frac{1}{\mu_i^k} \|\boldsymbol{y}_i^k - \boldsymbol{y}_i^*\|^2 + \mu_i^k \|\boldsymbol{G}_i(\boldsymbol{w}^k - \boldsymbol{w}^*)\|^2 + \frac{1}{\rho_i^k} \|\boldsymbol{\lambda}_i^k - \boldsymbol{\lambda}_i^*\|^2 + \rho_i^k \|\boldsymbol{s}_i^k - \boldsymbol{s}_i^*\|^2 \right)
$$

$$
\leq \sum_{i \in \mathcal{V}} (\eta_i^k + 1) \left( \frac{1}{\mu_i^{k-1}} \|\boldsymbol{y}_i^k - \boldsymbol{y}_i^*\|^2 + \mu_i^{k-1} \|\boldsymbol{G}_i(\boldsymbol{w}^k - \boldsymbol{w}^*)\|^2 + \frac{1}{\rho_i^{k-1}} \|\boldsymbol{\lambda}_i^k - \boldsymbol{\lambda}_i^*\|^2 \right.
$$

$$
\left. + \rho_i^{k-1} \|\boldsymbol{s}_i^k - \boldsymbol{s}_i^*\|^2 \right)
\tag{76}
$$

$$
\leq (\eta_{\max}^k + 1) \sum_{i \in \mathcal{V}} \left( \frac{1}{\mu_i^{k-1}} \|\boldsymbol{y}_i^k - \boldsymbol{y}_i^*\|^2 + \mu_i^{k-1} \|\boldsymbol{G}_i(\boldsymbol{w}^k - \boldsymbol{w}^*)\|^2 + \frac{1}{\rho_i^{k-1}} \|\boldsymbol{\lambda}_i^k - \boldsymbol{\lambda}_i^*\|^2 \right.
$$

$$
\left. + \rho_i^{k-1} \|\boldsymbol{s}_i^k - \boldsymbol{s}_i^*\|^2 \right)
$$

$$
= (\eta_{\max}^k + 1) V^k.
$$

Substituting the above result in (75), we get

$$\frac{(2-\alpha^k)}{\alpha^k} \sum_{i\in\mathcal{V}} \left( \frac{1}{\mu_i^k} \|\boldsymbol{y}_i^{k+1} - \boldsymbol{y}_i^k\|^2 + \mu_i^k \|\boldsymbol{G}_i(\boldsymbol{w}^{k+1} - \boldsymbol{w}^k)\|^2 + \frac{1}{\rho_i^k} \|\boldsymbol{\lambda}_i^{k+1} - \boldsymbol{\lambda}_i^k\|^2 \right.$$
$$\left. + \rho_i^k \|\boldsymbol{s}_i^{k+1} - \boldsymbol{s}_i^k\|^2 \right) \leq (\eta_{\max}^k + 1)V^k - V^{k+1}. \tag{77}$$

Now that we have derived the above relationship, we need to prove that $V^k$ is bounded. By the definition of $V^k$, we have that $V^k$ is lower bounded by zero. Thus, we now prove that $V^k$ is upper bounded. From (77), we have

$$V^{k+1} \leq (\eta_{\max}^k + 1)V^k, \tag{78}$$

which leads to the following relationship

$$V^{k+1} \leq \prod_{l=1}^{k} (\eta_{\max}^l + 1)V^1. \tag{79}$$

It should be noted that based on Assumption 1, we have $(\eta_{\max}^k + 1) \to 1$, as $k \to \infty$. Therefore, (79) implies that $V^{k+1}$ is upper bounded for all $k$, and there exists $V_{\max}$ such that

$$V^k \leq V_{\max} < \infty, \quad \text{for all } k. \tag{80}$$

Let us now consider summing (77) over $k$ as follows

$$\sum_{k=1}^{\infty} \frac{(2-\alpha^k)}{\alpha^k} \sum_{i\in\mathcal{V}} \left( \frac{1}{\mu_i^k} \|\boldsymbol{y}_i^{k+1} - \boldsymbol{y}_i^k\|^2 + \mu_i^k \|\boldsymbol{G}_i(\boldsymbol{w}^{k+1} - \boldsymbol{w}^k)\|^2 + \frac{1}{\rho_i^k} \|\boldsymbol{\lambda}_i^{k+1} - \boldsymbol{\lambda}_i^k\|^2 \right.$$
$$\left. + \rho_i^k \|\boldsymbol{s}_i^{k+1} - \boldsymbol{s}_i^k\|^2 \right) \leq \sum_{k=1}^{\infty} (\eta_{\max}^k + 1)V^k - V^{k+1}. \tag{81}$$

The term on the RHS of the above equation can be further simplified as follows

$$\sum_{k=1}^{\infty} (\eta_{\max}^k + 1)V^k - V^{k+1} = \sum_{k=1}^{\infty} \eta_{\max}^k V^k + \sum_{k=1}^{\infty} (V^k - V^{k+1}) = V^1 - V^\infty + \sum_{k=1}^{\infty} \eta_{\max}^k V^k.$$

Based on Assumption 1, we have $\eta_{\max}^k \to 0$ as $k \to \infty$, which implies that

$$\sum_{k=1}^{\infty} \eta_{\max}^k < \infty. \tag{82}$$

Using the above fact and (80), we can upper bound $\sum_{k=1}^{\infty} \eta_{\max}^k V^k$ as follows

$$\sum_{k=1}^{\infty} \eta_{\max}^k V^k \leq \left( \sum_{k=1}^{\infty} \eta_{\max}^k \right) V_{\max} < \infty. \tag{83}$$

Using the facts that $V^1$ is upper bounded, and $V^\infty$ is lower bounded by zero, and using the above equation, we get

$$V^1 - V^\infty + \sum_{k=1}^{\infty} \eta_{\max}^k V^k \leq V^1 + \sum_{k=1}^{\infty} \eta_{\max}^k V^k < \infty.$$

Thus, we can rewrite (81) as

$$\sum_{k=1}^{\infty} \frac{(2-\alpha^k)}{\alpha^k} \sum_{i\in\mathcal{V}} \left( \frac{1}{\mu_i^k} \|\boldsymbol{y}_i^{k+1} - \boldsymbol{y}_i^k\|^2 + \mu_i^k \|\boldsymbol{G}_i(\boldsymbol{w}^{k+1} - \boldsymbol{w}^k)\|^2 + \frac{1}{\rho_i^k} \|\boldsymbol{\lambda}_i^{k+1} - \boldsymbol{\lambda}_i^k\|^2 \right.$$
$$\left. + \rho_i^k \|\boldsymbol{s}_i^{k+1} - \boldsymbol{s}_i^k\|^2 \right) < \infty. \tag{84}$$

Since $\alpha^k \in [1, 2)$, we have $\frac{(2-\alpha^k)}{\alpha^k} > 0$ for all $k$. Further, we have $0 < \mu_i^k, \rho_i^k < \infty$ for all $k$. Thus, (84) implies that as $k \to \infty$,

$$(\boldsymbol{y}_i^{k+1} - \boldsymbol{y}_i^k) \to \boldsymbol{0}, \quad \boldsymbol{G}_i(\boldsymbol{w}^{k+1} - \boldsymbol{w}^k) \to \boldsymbol{0}, \quad (\boldsymbol{\lambda}_i^{k+1} - \boldsymbol{\lambda}_i^k) \to \boldsymbol{0}, \quad \boldsymbol{s}_i^{k+1} - \boldsymbol{s}_i^k \to \boldsymbol{0}, \tag{85}$$

for all $i \in \mathcal{V}$. This proves the convergence of the variables $\boldsymbol{y}_i, \boldsymbol{\lambda}_i$ and $\boldsymbol{s}_i$. Further, it follows that $\boldsymbol{G}(\boldsymbol{w}^{k+1} - \boldsymbol{w}^k) \to \boldsymbol{0}$. Since $\boldsymbol{G}$ is full column rank, this implies that as $k \to \infty$,

$$(\boldsymbol{w}^{k+1} - \boldsymbol{w}^k) \to \boldsymbol{0}, \tag{86}$$

which proves the convergence of the global variable $\boldsymbol{w}$. Subsequently, combining (R2), (R3), and the convergence result (85), we also obtain that as $k \to \infty$,

$$(\boldsymbol{x}_i^{k+1} - \boldsymbol{x}_i^k) \to \boldsymbol{0}, \quad (\boldsymbol{z}_i^{k+1} - \boldsymbol{z}_i^k) \to \boldsymbol{0}, \tag{87}$$

for all $i \in \mathcal{V}$. Hence, we have proved the convergence of the DistributedQP algorithm.

Now that we have proved the convergence of all variables, we proceed with verifying that the limit point of convergence is the optimal solution to problem (3). For that, we need to check if the limit point satisfies the KKT conditions (43) for the problem (3). The convergence of the dual variables $\boldsymbol{y}_i$ and $\boldsymbol{\lambda}_i$, and the update steps verify that the limit points have constraint feasibility (43e - 43h). The constraint feasibility of the limit points and the optimality conditions of $(k+1)$-th update of $\boldsymbol{x}_i, \boldsymbol{z}_i$ (47) imply that the limit points satisfy the optimality conditions (43a - 43b). Further, using relations (R1) and (R4), we can prove that the limit points also satisfy (43c - 43d). Consequently, the DistributedQP algorithm converges to the optimal point of problem (3) which is equivalent to problem (2).

## E  DETAILS ON DEEPDISTRIBUTEDQP FEEDBACK POLICIES

In DeepDistributedQP, the penalty parameters are given by

$$\rho_i^k = \mathrm{SoftPlus}\Big(\bar{\rho}_i^k + \pi_{i,\rho}^k(r_{i,\rho}^k, s_{i,\rho}^k; \theta_{i,\rho}^k)\Big), \quad \mu_i^k = \mathrm{SoftPlus}\Big(\bar{\mu}_i^k + \pi_{i,\mu}^k(r_{i,\mu}^k, s_{i,\mu}^k; \theta_{i,\mu}^k)\Big) \tag{88}$$

where $\bar{\rho}_i^k, \bar{\mu}_i^k$ are learnable feed-forward parameters and $\pi_{i,\cdot}^k(r_{i,\cdot}^k, s_{i,\cdot}^k; \theta_{i,\cdot}^k)$ are learnable feedback policies parameterized by fully-connected neural network layers with inputs $r_{i,\cdot}^k, s_{i,\cdot}^k$ and weights $\theta_{i,\cdot}^k$. The analytical expressions for $r_{i,\cdot}^k, s_{i,\cdot}^k$ are provided as follows:

$$r_{i,\rho}^k = \begin{bmatrix} \|\boldsymbol{z}_i^k - \boldsymbol{s}_i^k\|_2 \\ \|\boldsymbol{A}_i \boldsymbol{x}_i^k - \boldsymbol{s}_i^k\|_2 \end{bmatrix}, \quad s_{i,\rho}^k = \begin{bmatrix} \|\boldsymbol{s}_i^k - \boldsymbol{s}_i^{k-1}\|_2 \\ \|\boldsymbol{Q}_i \boldsymbol{x}_i^k + \boldsymbol{q}_i + \boldsymbol{A}_i^\top \boldsymbol{\lambda}_i^k\|_2 \end{bmatrix}, \tag{89a}$$

$$r_{i,\mu}^k = \|\boldsymbol{x}_i^k - \tilde{\boldsymbol{w}}_i^k\|_2, \qquad s_{i,\mu}^k = \|\tilde{\boldsymbol{w}}_i^k - \tilde{\boldsymbol{w}}_i^{k-1}\|_2, \tag{89b}$$

being motivated by the primal and dual residuals of ADMM (Boyd et al., 2011, Section 3) and the ones used in the OSQP algorithm (Stellato et al., 2020).

## F  THE CENTRALIZED VERSION: DEEPQP

The centralized version of DeepDistributedQP boils down to simply unfolding the iterates of the standard OSQP algorithm for solving centralized QPs (1), while applying the same principles as in Section 4.1 for DeepDistributedQP.

For convenience, we repeat the OSQP updates from Stellato et al. (2020) here:

1. *Update for $(\boldsymbol{x}, \boldsymbol{z})$:* Solve linear system

$$\begin{bmatrix} \boldsymbol{Q} + \sigma \boldsymbol{I} & \boldsymbol{A}^\top \\ \boldsymbol{A} & -1/\rho^k \boldsymbol{I} \end{bmatrix} \begin{bmatrix} \boldsymbol{x}^{k+1} \\ \boldsymbol{\nu}^{k+1} \end{bmatrix} = \begin{bmatrix} \sigma \boldsymbol{t}^k - \boldsymbol{q} \\ \boldsymbol{s}^k - 1/\rho^k \boldsymbol{\lambda}^k \end{bmatrix} \tag{90}$$

   and update

$$\boldsymbol{z}^{k+1} = \boldsymbol{s}^k + 1/\rho^k (\boldsymbol{\nu}^{k+1} - \boldsymbol{\lambda}^k). \tag{91}$$

   As explained in Stellato et al. (2020), as the scale of the system (90) increases, it is often preferable to solve the following system instead,

$$(\boldsymbol{Q} + \sigma \boldsymbol{I} + \rho^k \boldsymbol{A}^\top \boldsymbol{A}) \boldsymbol{x}^{k+1} = \sigma \boldsymbol{x}^k - \boldsymbol{q} + \boldsymbol{A}^\top (\rho^k \boldsymbol{z}^k - \boldsymbol{y}^k), \tag{92}$$

   using a method such as conjugate gradient.

2. *Update for $(\boldsymbol{t}, \boldsymbol{s})$:*

$$\boldsymbol{t}^{k+1} = \alpha^k \boldsymbol{x}^{k+1} + (1 - \alpha^k)\boldsymbol{t}^k \tag{93a}$$

$$\boldsymbol{s}^{k+1} = \Pi_{\mathcal{C}}\left(\alpha^k \boldsymbol{z}^{k+1} + (1 - \alpha^k)\boldsymbol{s}^k + \boldsymbol{\lambda}^k/\rho^k\right) \tag{93b}$$

3. *Dual update for $\boldsymbol{\lambda}$:*

$$\boldsymbol{\lambda}^{k+1} = \boldsymbol{\lambda}^k + \rho^k(\alpha^k \boldsymbol{z}^{k+1} + (1 - \alpha^k)\boldsymbol{s}^k - \boldsymbol{s}^{k+1}) \tag{94}$$

The DeepQP framework then emerges through unfolding the OSQP updates following the same methodology as in DeepDistributedQP. In particular, its iterations are unrolled for a prescribed amount of $K$ iterations as shown in Fig. 4.

Similar to DeepDistributedQP, the penalty and relaxation parameters are given by

$$\rho^k = \text{SoftPlus}\left(\bar{\rho}^k + \pi^k(r^k, s^k; \theta^k)\right), \quad \alpha^k = \text{Sigmoid}_{1,2}(\bar{\alpha}^k) \tag{95}$$

where $\bar{\rho}^k$ and $\bar{\alpha}^k$ are learnable feed-forward parameters. The feedback components of the parameters $\rho^k$ are obtained through the learnable policies $\pi^k(r^k, s^k; \theta^k)$ parameterized by fully-connected neural network layers with inputs $r^k, s^k$ and weights $\theta^k$. The expressions for $r^k, s^k$ are also motivated by the ADMM and OSQP residuals, given by

$$r^k = \begin{bmatrix} \|\boldsymbol{z}^k - \boldsymbol{s}^k\|_2 \\ \|\boldsymbol{A}\boldsymbol{x}^k - \boldsymbol{s}^k\|_2 \end{bmatrix}, \quad s^k = \begin{bmatrix} \|\boldsymbol{s}^k - \boldsymbol{s}^{k-1}\|_2 \\ \|\boldsymbol{Q}\boldsymbol{x}^k + \boldsymbol{q} + \boldsymbol{A}^\top\boldsymbol{\lambda}^k\|_2 \end{bmatrix} \tag{96}$$

# G  PROOF OF INDIRECT METHOD IMPLICIT DIFFERENTIATION

We start by restating the implicit function theorem, whose proof is given in Krantz & Parks (2002).

**Lemma 5** (Implicit Function Theorem). *Let $r : \mathbb{R}^n \times \mathbb{R}^m \to \mathbb{R}^n$ be a continuously differentiable function. Let $(\boldsymbol{x}_0, \boldsymbol{\theta}_0)$ be a point such that $r(\boldsymbol{x}_0, \boldsymbol{\theta}_0) = 0$. If the Jacobian matrix $\frac{\partial r}{\partial \boldsymbol{x}}(\boldsymbol{x}_0, \boldsymbol{\theta}_0)$ is invertible, then there exists a function $\boldsymbol{x}^*(\cdot)$ defined in a neighborhood of $\boldsymbol{\theta}_0$ such that $\boldsymbol{x}^*(\boldsymbol{\theta}_0) = \boldsymbol{x}_0$, and*

$$\frac{\partial \boldsymbol{x}^*}{\partial \boldsymbol{\theta}}(\boldsymbol{\theta}) = -\left(\frac{\partial r}{\partial \boldsymbol{x}}(\boldsymbol{x}^*(\boldsymbol{\theta}), \boldsymbol{\theta})\right)^{-1}\frac{\partial r}{\partial \boldsymbol{\theta}}(\boldsymbol{x}^*(\boldsymbol{\theta}), \boldsymbol{\theta}). \tag{97}$$

*Proof of Theorem 2.* Let $\boldsymbol{\theta} = (\bar{\boldsymbol{Q}}_i^k, \bar{\boldsymbol{b}}_i^k)$ be the concatenation of all the parameters in Eq. (12). $\bar{\boldsymbol{Q}}_i^k$ is always positive definite since $\boldsymbol{Q}_i$ is positive definite and the penalty parameters are always non-negative. Therefore, Eq. (12) has a unique solution $\boldsymbol{x}_i^{k+1}$ satisfying $r(\boldsymbol{x}_i^{k+1}, \boldsymbol{\theta}) := \bar{\boldsymbol{Q}}_i^k \boldsymbol{x}_i^{k+1} - \bar{\boldsymbol{b}}_i^k = 0$. Applying Lemma 5 to this residual function yields the relationship $\frac{\partial \boldsymbol{x}_i^{k+1}}{\partial \boldsymbol{\theta}}(\boldsymbol{\theta}) = -(\bar{\boldsymbol{Q}}_i^k)^{-1}\frac{\partial r}{\partial \boldsymbol{\theta}}(\boldsymbol{x}_i^{k+1}(\boldsymbol{\theta}), \boldsymbol{\theta})$.

Now, for any downstream loss function $L(\boldsymbol{x}_i^{k+1}(\boldsymbol{\theta}))$, we have that

$$\begin{aligned}
\nabla_{\boldsymbol{\theta}} L(\boldsymbol{x}_i^{k+1}(\boldsymbol{\theta})) &= \frac{\partial \boldsymbol{x}_i^{k+1}}{\partial \boldsymbol{\theta}}(\boldsymbol{\theta})\nabla_{\boldsymbol{x}} L(\boldsymbol{x}_i^{k+1}(\boldsymbol{\theta})) \\
&= -\frac{\partial r}{\partial \boldsymbol{\theta}}(\boldsymbol{x}_i^{k+1}(\boldsymbol{\theta}), \boldsymbol{\theta})^\top(\bar{\boldsymbol{Q}}_i^k)^{-1}\nabla_{\boldsymbol{x}} L(\boldsymbol{x}_i^{k+1}(\boldsymbol{\theta})) \\
&= \frac{\partial r}{\partial \boldsymbol{\theta}}(\boldsymbol{x}_i^{k+1}(\boldsymbol{\theta}), \boldsymbol{\theta})^\top d\boldsymbol{x}_i^{k+1},
\end{aligned} \tag{98}$$

where $d\boldsymbol{x}_i^{k+1}$ is the unique solution to the linear system

$$\bar{\boldsymbol{Q}}_i^k d\boldsymbol{x}_i^{k+1} = -\nabla_{\boldsymbol{x}} L(\boldsymbol{x}_i^{k+1}(\boldsymbol{\theta})).$$

Expanding the matrix multiplication in (98) yields

$$\begin{aligned}
\nabla_{\bar{\boldsymbol{Q}}_i^k} L &= \frac{1}{2}(\boldsymbol{x}_i^{k+1} \otimes d\boldsymbol{x}_i^{k+1} + d\boldsymbol{x}_i^{k+1} \otimes \boldsymbol{x}_i^{k+1}), \\
\nabla_{\bar{\boldsymbol{b}}_i^k} L &= -d\boldsymbol{x}_i^{k+1}.
\end{aligned}$$

$\square$

## H    BACKGROUND ON PAC-BAYES THEORY

Here, we provide a brief overview of PAC-Bayes theory (Alquier, 2024). Consider a bounded loss function $\ell(\zeta; \theta)$. Without loss of generality, we assume that this loss is uniformly bounded between 0 and 1. PAC-Bayes theory aims to providing a probabilistic bound for the true expected loss

$$\ell_{\mathcal{D}}(\mathcal{P}) = \mathbb{E}_{\zeta \sim \mathcal{D}} \, \mathbb{E}_{\theta \sim \mathcal{P}} \left[ \ell(\zeta; \theta) \right], \tag{99}$$

where $\mathcal{D}$ is the data distribution — in our case, this is the distribution optimization problems are drawn from. The empirical expected loss is given by,

$$\ell_{\mathcal{S}}(\mathcal{P}) = \mathbb{E}_{\theta \sim \mathcal{P}} \left[ \frac{1}{H} \sum_{j=1}^{H} \ell(\zeta^j; \theta) \right], \tag{100}$$

where $\mathcal{S} = \{\zeta^j\}_{j=1}^{H}$ is the training dataset consisting of $H$ problem instances.

The PAC-Bayes framework operates by forming a bound that holds in high probability on the true loss $\ell_{\mathcal{D}}(\mathcal{P})$ in terms of the empirical loss and the deviation between the learned policy $\mathcal{P}$ and a prior policy $\mathcal{P}_0$ used to as an initial guess for $\mathcal{P}$. This deviation is measured using the KL divergence. Importantly, $\mathcal{P}_0$ need not be a Bayesian prior but can be any distribution independent of the data used to train $\mathcal{P}$ and evaluate the sample loss. Moreover, $\ell(\zeta; \theta)$ need not be the loss used to train $\mathcal{P}$, but can be any bounded function. This observation is useful because, both in the literature and in the sequel, it is common to use a loss function modified for practicality during training before evaluating the bound using the loss function of interest.

Specifically, the following PAC-Bayes bounds hold with probability $1 - \delta$,

$$\ell_{\mathcal{D}}(\mathcal{P}) \le \mathbb{D}_{\mathrm{KL}}^{-1} \left( \ell_{\mathcal{S}}(\mathcal{P}) \| \frac{\mathbb{D}_{\mathrm{KL}}(\mathcal{P} \| \mathcal{P}_0) + \log \frac{2\sqrt{H}}{\delta}}{H} \right) \le \ell_{\mathcal{S}}(\mathcal{P}) + \sqrt{\frac{\mathbb{D}_{\mathrm{KL}}(\mathcal{P} \| \mathcal{P}_0) + \log \frac{2\sqrt{H}}{\delta}}{2H}}, \tag{101}$$

where the $\mathbb{D}_{\mathrm{KL}}^{-1}(p \| c)$ is the *inverse of the KL divergence* for Bernoulli random variables $\mathcal{B}(p), \mathcal{B}(q)$:

$$\mathbb{D}_{\mathrm{KL}}^{-1}(p \| c) = \sup\{q \in [0, 1] \mid \mathbb{D}_{\mathrm{KL}}(\mathcal{B}(p) \| \mathcal{B}(q)) \le c\}. \tag{102}$$

The probability $\delta$ captures the failure case that the data set $\mathcal{S}$ is not sufficiently representative of the data distribution $\mathcal{D}$. In the sequel, both of the above inequalities will be used. As the first bound is tighter, it is used to evaluate the generalization capabilities of the learned optimizer. The benefit of the second, looser, bound is that its form is convenient to use during training as a regularizer. Using both bounds in this manner is a common technique in the PAC-Bayes literature (Majumdar et al., 2021; Dziugaite & Roy, 2017).

## I    OPTIMIZING AND EVALUATING GENERALIZATION BOUND

Two important requirements for establishing a tight PAC-Bayes bound are selecting an informative prior and optimizing the PAC-Bayes bounds in (101) instead of simply minimizing the loss function. The choice of prior $\mathcal{P}_0$ is particularly important because the KL divergence is unbounded and can produce a vacuous result (Dziugaite et al., 2021). While the distribution $\mathcal{P}_0$ need not be a Bayesian prior, it must be selected independently from the data used to optimize $\mathcal{P}$ and evaluate the bound. To select $\mathcal{P}_0$, we follow a common approach in the literature and split our training set $\mathcal{S}$ into two disjoint subsets $\mathcal{S}_0, \mathcal{S}_1$. The prior $\mathcal{P}_0$ is first trained using the data set $\mathcal{S}_0$ and the loss $\ell(\mathcal{D}; \Theta)$ discussed in Section 4.

Subsequently, the posterior $\mathcal{P}$ is trained by minimizing the looser (i.e., rightmost) PAC-Bayes bound in (101). This bound is used for training because it is straightforward to evaluate in comparison to computing the inverse of the KL divergence, and this objective is easily interpreted as minimizing an expected loss function with a regularizer. To evaluate the loss function in the PAC-Bayes bound, parameters are sampled from $\mathcal{P}$ using the current network weights and an empirical average is used. Once training is complete, the PAC-Bayes bound is evaluated as described in Theorem 3, i.e., by using the tighter PAC-Bayes bound in (101) and the sample convergence bound in (16).

## J   DETAILS ON EXPERIMENTS

This section provides further details on the problems considered in the experiments, the training of the learned optimizers, as well as the evaluation of both learned and traditional methods.

### J.1   PROBLEM TYPES IN CENTRALIZED EXPERIMENTS

**Random QPs.**   We consider randomly generated problems of the following form

$$\min_{\boldsymbol{x}} \frac{1}{2}\boldsymbol{x}^\top \boldsymbol{Q}\boldsymbol{x} + \boldsymbol{q}^\top \boldsymbol{x} \quad \text{s.t.} \quad \boldsymbol{A}\boldsymbol{x} \le \boldsymbol{b}, \; \boldsymbol{C}\boldsymbol{x} = \boldsymbol{d}. \tag{103}$$

For each generated problem, the cost Hessian is given by $\boldsymbol{Q} = \boldsymbol{F}^\top \boldsymbol{F} + \gamma \boldsymbol{I}$, where each element of $\boldsymbol{F} \in \mathbb{R}^{n\times n}$ is sampled through $\boldsymbol{F}_{ij} \sim \mathcal{N}(0,1)$ and $\gamma = 1.0$. The coefficients of $\boldsymbol{q}$ are also sampled as $\boldsymbol{q}_i \sim \mathcal{N}(0,1)$. The elements of the inequality constraints matrix $\boldsymbol{A} \in \mathbb{R}^{m\times n}$ are given by $\boldsymbol{A}_{ij} \sim \mathcal{N}(0,1)$, while $\boldsymbol{b} = \boldsymbol{A}\boldsymbol{\theta}$, where each element of $\boldsymbol{\theta} \in \mathbb{R}^n$ is sampled through $\boldsymbol{\theta}_i \sim \mathcal{N}(0,1)$. Similarly, the elements of the equality constraints matrix $\boldsymbol{C} \in \mathbb{R}^{p\times n}$ are given by $\boldsymbol{C}_{ij} \sim \mathcal{N}(0,1)$, while $\boldsymbol{d} = \boldsymbol{C}\boldsymbol{\xi}$, where each element of $\boldsymbol{\xi} \in \mathbb{R}^n$ is $\boldsymbol{\xi}_i \sim \mathcal{N}(0,1)$.

For random QPs without equality constraints, we set $n = 50$, $m = 40$ and $p = 0$. For random QPs with equality constraints, we set $n = 50$, $m = 25$ and $p = 20$.

**Optimal control.**   We consider linear optimal control problems of the following form

$$\min_{\boldsymbol{x},\boldsymbol{u}} \sum_{t=0}^{T-1} \boldsymbol{x}_t^\top \boldsymbol{Q}\boldsymbol{x}_t + \boldsymbol{u}_t^\top \boldsymbol{R}\boldsymbol{u}_t + \boldsymbol{x}_T^\top \boldsymbol{Q}_T \boldsymbol{x}_T \tag{104a}$$

$$\text{s.t.} \quad \boldsymbol{x}_{t+1} = \boldsymbol{A}_\mathrm{d}\boldsymbol{x}_t + \boldsymbol{B}_\mathrm{d}\boldsymbol{u}_t, \quad t = 0,\dots,T-1, \tag{104b}$$

$$\boldsymbol{A}_u \boldsymbol{u}_t \le \boldsymbol{b}_u, \quad \boldsymbol{A}_x \boldsymbol{x}_t \le \boldsymbol{b}_x, \quad t = 0,\dots,T, \tag{104c}$$

$$\boldsymbol{x}_0 = \bar{\boldsymbol{x}}_0. \tag{104d}$$

where $\boldsymbol{x} = \{\boldsymbol{x}_0,\dots,\boldsymbol{x}_T\}$ is the state trajectory, $\boldsymbol{u} = \{\boldsymbol{u}_0,\dots,\boldsymbol{u}_{T-1}\}$ is the control trajectory, $\bar{\boldsymbol{x}}_0$ is the given initial state condition, $\boldsymbol{Q}$ and $\boldsymbol{R}$ are the running state and control cost matrices, $\boldsymbol{Q}_T$ is the terminal state cost matrix, $\boldsymbol{A}_\mathrm{d}$ and $\boldsymbol{B}_\mathrm{d}$ are the dynamics matrices, and finally $\boldsymbol{A}_u, \boldsymbol{b}_u$ and $\boldsymbol{A}_x, \boldsymbol{b}_x$ are the control and state constraints coefficients, respectively.

Both the double integrator and the mass-spring problem setups are drawn from Chen et al. (2022a). For the double integrator system, we have $x_t \in \mathbb{R}^2$ and $u_t \in \mathbb{R}$, with time horizon $T = 20$. The dynamics matrices are given by

$$\boldsymbol{A}_\mathrm{d} = \begin{bmatrix} 1 & 1 \\ 0 & 1 \end{bmatrix}, \quad \boldsymbol{B}_\mathrm{d} = \begin{bmatrix} 0.5 \\ 0.1 \end{bmatrix} \tag{105}$$

The cost matrices are $\boldsymbol{Q} = \boldsymbol{Q}_T = \boldsymbol{I}_2$ and $R = 1.0$. The state and control constraint coefficients are given by

$$\boldsymbol{A}_x = \begin{bmatrix} \boldsymbol{I}_2 \\ -\boldsymbol{I}_2 \end{bmatrix}, \quad \boldsymbol{b}_x = [5 \quad 1 \quad 5 \quad 1]^\top, \quad \boldsymbol{A}_u = \begin{bmatrix} 1 \\ -1 \end{bmatrix}, \quad \boldsymbol{b}_u = [0.1 \quad 0.1]^\top. \tag{106}$$

Finally, the initial state conditions are sampled from the uniform distribution $\mathcal{U}[[-1; -0.3], [1; 0.3]]$.

For the oscillating masses, we have $x_t \in \mathbb{R}^{12}$ and $u_t \in \mathbb{R}^3$, with time horizon $T = 10$. The discrete-time dynamics matrices are obtained from the continuous-time ones through Euler discretization,

$$\boldsymbol{A}_\mathrm{d} = \boldsymbol{I} + \boldsymbol{A}_\mathrm{c}\Delta t, \quad \boldsymbol{B}_\mathrm{d} = \boldsymbol{A}_\mathrm{c}\Delta t. \tag{107}$$

The continuous-time dynamics matrices are given by

$$\boldsymbol{A}_\mathrm{c} = \begin{bmatrix} \boldsymbol{0}_{6\times 6} & \boldsymbol{I}_6 \\ a\boldsymbol{I}_6 + c\boldsymbol{L}_6 + c\boldsymbol{L}_6^\top & b\boldsymbol{I}_6 + d\boldsymbol{L}_6 + d\boldsymbol{L}_6^\top \end{bmatrix}, \quad \boldsymbol{B}_\mathrm{c} = \begin{bmatrix} \boldsymbol{0}_{6\times 3} \\ \boldsymbol{F} \end{bmatrix} \tag{108}$$

with $c = 1.0$, $d = 0.1$, $a = -2c$, $b = -2.0$. $\boldsymbol{L}_6$ is the $6 \times 6$ lower shift matrix and

$$\boldsymbol{F} = [\boldsymbol{e}_1 \quad -\boldsymbol{e}_1 \quad \boldsymbol{e}_2 \quad \boldsymbol{e}_3 \quad -\boldsymbol{e}_2 \quad \boldsymbol{e}_3]^\top \tag{109}$$

where $e_1, e_2, e_3$ are the standard basis vectors in $\mathbb{R}^3$.

The timestep is set as $\Delta t = 0.5$. The cost matrices are $Q = Q_T = I_{12}$ and $R = I_3$. The state and control constraints are defined through

$$A_x = \begin{bmatrix} I_{12} \\ -I_{12} \end{bmatrix}, \quad b_x = 4 \cdot \mathbf{1}_{24}, \quad A_u = \begin{bmatrix} I_3 \\ -I_3 \end{bmatrix}, \quad b_u = 0.5 \cdot \mathbf{1}_6. \tag{110}$$

The initial conditions $\bar{x}_0$ are sampled from $\mathcal{U}\left[[-1, 1]^{12}\right]$.

**Portfolio optimization.** We consider the same portfolio optimization problem setup as in Stellato et al. (2020). For completeness, we briefly repeat it here,

$$\max_{x} \; \boldsymbol{\mu}^\top \boldsymbol{x} - \gamma(\boldsymbol{x}^\top \boldsymbol{\Sigma} \boldsymbol{x}) \quad \text{s.t.} \quad x_1 + \cdots + x_n = 1, \quad \boldsymbol{x} \geq \mathbf{0}, \tag{111}$$

where $\boldsymbol{x} \in \mathbb{R}^n$ is the assets allocation vector, $\boldsymbol{\mu} \in \mathbb{R}^n$ is the expected returns vector, $\boldsymbol{\Sigma} \in \mathbb{R}_+^N$ is the risk covariance matrix and $\gamma > 0$ is the risk aversion parameter. The matrix $\boldsymbol{\Sigma}$ is of the form $\boldsymbol{\Sigma} = \boldsymbol{F}\boldsymbol{F}^\top + \boldsymbol{D}$ with $\boldsymbol{F} \in \mathbb{R}^{d \times n}$ is the factors matrix and $\boldsymbol{D} \in \mathbb{R}^{n \times n}$ is a diagonal matrix involving individual asset risks. Using an auxiliary variable $\boldsymbol{t} = \boldsymbol{F}^\top \boldsymbol{x}$, then problem equation 111 is rewritten as

$$\min_{\boldsymbol{x}, \boldsymbol{t}} \; \boldsymbol{x}^\top \boldsymbol{D}\boldsymbol{x} + \boldsymbol{t}^\top \boldsymbol{t} - \frac{1}{\gamma}\boldsymbol{\mu}^\top \boldsymbol{x} \quad \text{s.t.} \quad \boldsymbol{t} = \boldsymbol{F}^\top \boldsymbol{x}, \quad \mathbf{1}^\top \boldsymbol{x} = 1, \quad \boldsymbol{x} \geq \mathbf{0}. \tag{112}$$

For the problems we are generating, we use $n = 250$, $k = 25$ and $\gamma = 1.0$. Each element of the expected return vector $\boldsymbol{\mu}$ is sampled through $\mu_i \sim \mathcal{N}(0, 1)$. The matrix $\boldsymbol{F}$ consists of $50\%$ non-zero elements sampled through $F_{ij} \sim \mathcal{N}(0, 1)$. Finally, the diagonal elements of $\boldsymbol{D}$ are sampled with $\mathcal{D}_{ii} \sim \mathcal{U}[0, \sqrt{k}]$.

**LASSO.** The least absolute shrinkage and selection operator (LASSO) is a linear regression technique with an added $\ell_1$-norm regularization term to promote sparsity in the parameters (Tibshirani, 1996). We again consider the same problem setup as in Stellato et al. (2020), where the initial optimization problem

$$\min_{x} \; \|\boldsymbol{A}\boldsymbol{x} - \boldsymbol{b}\|_2^2 + \lambda\|\boldsymbol{x}\|_1 \tag{113}$$

is rewritten as

$$\min_{\boldsymbol{x}, \boldsymbol{t}} \; (\boldsymbol{A}\boldsymbol{x} - \boldsymbol{b})^\top (\boldsymbol{A}\boldsymbol{x} - \boldsymbol{b}) + \lambda \mathbf{1}^\top \boldsymbol{t} \quad \text{s.t.} \quad -\boldsymbol{t} \leq \boldsymbol{x} \leq \boldsymbol{t}, \tag{114}$$

where $\boldsymbol{x} \in \mathbb{R}^n$ is the vector of parameters, $\boldsymbol{A} \in \mathbb{R}^{m \times n}$ is the data matrix, $\lambda$ is the weighting parameter, and $\boldsymbol{t} \in \mathbb{R}^n$ are newly introduced variables. The matrix $\boldsymbol{A}$ consists of $15\%$ non-zero elements sampled through $A_{ij} \sim \mathcal{N}(0, 1)$. The true sparse vector $\boldsymbol{v} \in \mathbb{R}^n$ to be learned consists of $50\%$ non-zero elements sampled through $v_i \sim \mathcal{N}(0, 1/n)$. We then construct $\boldsymbol{b} = \boldsymbol{A}\boldsymbol{v} + \boldsymbol{\xi}$ where $\xi_i \sim \mathcal{N}(0, 1)$ represents noise in the data. Finally, we set $\lambda = (1/5)\|\boldsymbol{A}^\top \boldsymbol{b}\|_\infty$. For the problems we are generating, we set $n = 100$ and $m = 10^4$.

## J.2 PROBLEM TYPES IN DISTRIBUTED EXPERIMENTS

**Random Networked QPs.** In this family of problems, we generate random QPs with an underlying network structure. Consider an undirected graph $\mathcal{G}(\mathcal{V}, \mathcal{E})$, where $\mathcal{V}$ and $\mathcal{E}$ are the nodes and edges sets, respectively. Each node $i$ is associated with a decision variable $\boldsymbol{x}_i \in \mathbb{R}^{n_i}$. Then, we generate problems of the following form

$$\min_{\{\boldsymbol{x}_i\}_{i \in \mathcal{V}}} \; \sum_{i \in \mathcal{V}} \frac{1}{2}\boldsymbol{x}_i^\top \boldsymbol{Q}_i \boldsymbol{x}_i + \boldsymbol{q}_i^\top \boldsymbol{x}_i \tag{115a}$$

$$\text{s.t.} \quad \boldsymbol{A}_{ij} \begin{bmatrix} \boldsymbol{x}_i \\ \boldsymbol{x}_j \end{bmatrix} \leq \boldsymbol{b}_{ij}, \quad \boldsymbol{C}_{ij} \begin{bmatrix} \boldsymbol{x}_i \\ \boldsymbol{x}_j \end{bmatrix} = \boldsymbol{d}_{ij}, \quad (i, j) \in \mathcal{E}. \tag{115b}$$

For each generated problem, a cost Hessian is constructed as $\boldsymbol{Q}_i = \boldsymbol{F}_i^\top \boldsymbol{F}_i + \gamma \boldsymbol{I}$, where each element of $\boldsymbol{F}_i \in \mathbb{R}^{n_i \times n_i}$ is sampled through $\boldsymbol{F}_i^{kl} \sim \mathcal{N}(0, 1)$ and $\gamma = 1.0$. The elements of the cost coefficients vectors $\boldsymbol{q}_i$ are also sampled through $\boldsymbol{q}_i^k \sim \mathcal{N}(0, 1)$. The elements of the inequality

constraints matrix $\boldsymbol{A}_{ij} \in \mathbb{R}^{m_{ij} \times (n_i + n_j)}$ are given by $\boldsymbol{A}_{ij}^{kl} \sim \mathcal{N}(0, 1)$. The vectors $\boldsymbol{b}_{ij} \in \mathbb{R}^{m_{ij}}$ are obtained through $\boldsymbol{b}_{ij} = \boldsymbol{A}_{ij}\boldsymbol{\theta}_{ij}$, where each element of $\boldsymbol{\theta}_{ij} \in \mathbb{R}^{n_i+n_j}$ is sampled through $\boldsymbol{\theta}_{ij}^k \sim \mathcal{N}(0, 1)$. In a similar manner, the elements of the equality constraints matrices $\boldsymbol{C}_{ij} \in \mathbb{R}^{p_{ij} \times (n_i+n_j)}$ are generated through $\boldsymbol{C}_{ij}^{kl} \sim \mathcal{N}(0, 1)$, while the vectors $\boldsymbol{d}_{ij} \in \mathbb{R}^{p_{ij}}$ are acquired through $\boldsymbol{d}_{ij} = \boldsymbol{C}_{ij}\boldsymbol{\xi}_{ij}$, where each element of $\boldsymbol{\xi}_{ij} \in \mathbb{R}^{n_i+n_j}$ is generated with $\boldsymbol{\xi}_{ij}^k \sim \mathcal{N}(0, 1)$.

It is straightforward to observe that problems of the form (115) can be cast in the form (2) by introducing the augmented node variables $\boldsymbol{x}_i^{aug} = [x_i, \{x_j\}_{j \in \mathcal{N}_i}]^\top$. The problem data can then be augmented based on this new $\boldsymbol{x}_i^{aug}$ to yield the desired problem structure. Most notably, the constraints can be rewritten as $\boldsymbol{A}_i^{aug} \boldsymbol{x}_i^{aug} \leq b_i^{aug}$ and $\boldsymbol{C}_i^{aug} x_i^{aug} = d_i^{aug}$, respectively.

In our experiments, the underlying graph structure is a square grid. For random QPs without equality constraints, we set $n_i = 10$, $m_{ij} = 5$, and $p_{ij} = 0$. For random QPs with equality constraints, we set $n_i = 10$, $m_{ij} = 3$, and $p_{ij} = 2$ for the $N = 16$ training experiment and $p_{ij} = 1$ for the rest of the testing experiments until $N = 1,024$.

**Multi-agent optimal control.** We adapt the distributed MPC problem from (Conte et al., 2012a;b), which generalizes to different systems based on the choice of dynamics matrices, as described below. The optimization problem is given as

$$\min_{\boldsymbol{x},\boldsymbol{u}} \sum_{i \in V} \sum_{t=0}^{T-1} (\boldsymbol{x}_i^t)^\top \boldsymbol{Q}_i \boldsymbol{x}_i^t + (\boldsymbol{u}_i^t)^\top \boldsymbol{R}_i \boldsymbol{u}_i^t + (\boldsymbol{x}_i^T)^\top \boldsymbol{P}_i \boldsymbol{x}_i^T, \tag{116a}$$

$$\text{s.t.} \quad \boldsymbol{x}_i^{t+1} = \boldsymbol{A}_{ii}\boldsymbol{x}_i^t + \boldsymbol{B}_i\boldsymbol{u}_i^t + \sum_{j \in \mathcal{N}_i} \boldsymbol{A}_{ij}\boldsymbol{x}_j^t, \quad t = 0, \ldots, T-1, \quad i \in \mathcal{V} \tag{116b}$$

$$\boldsymbol{G}_x^i \boldsymbol{x}_i^t \leq \boldsymbol{f}_x^i, \boldsymbol{G}_u^i \boldsymbol{u}_i^t \leq \boldsymbol{f}_u^i, \quad t = 0, \ldots, T, \quad i \in \mathcal{V} \tag{116c}$$

$$\boldsymbol{x}_i^0 = \bar{\boldsymbol{x}}_i^0, \quad i \in \mathcal{V}, \tag{116d}$$

where $\boldsymbol{x}_i^t$ and $\boldsymbol{u}_i^t$ are the state and control for agent $i$ at time $t$. Eq. (116b) describes the dynamics and the coupling between the agents, Eq. (116c) describe local inequality constraints, and Eq. (116d) describes the initial condition for each of the agents.

For the coupled pendulums, the individual state $\boldsymbol{x}_i^t \in \mathbb{R}^2$ for each agent consists of the angle and angular velocity of the pendulum and the control $\boldsymbol{u}_i^t \in \mathbb{R}^1$ is the torque. The dynamics matrices are given as

$$\boldsymbol{A}_{ii} = \begin{bmatrix} 1 & dt \\ -(\frac{g}{\ell} + \frac{\text{nn}(i)k}{m})dt & 1 - \frac{\text{nn}(i)c}{m}dt \end{bmatrix}, \quad \boldsymbol{A}_{ij} = \begin{bmatrix} 0 & 0 \\ \frac{k}{m}dt & \frac{c}{m}dt \end{bmatrix}, \quad \boldsymbol{B}_i = \begin{bmatrix} 0 \\ \frac{1}{m\ell^2}dt \end{bmatrix},$$

where $dt = 0.1$ is the discretization step size, $g = 9.81$ is the gravitational constant, $m = 1.0$ is the mass of each pendulum, $\ell = 0.5$ is the length of each pendulum, $\text{nn}(i)$ is the number of neighbors of agent $i$, $k = 0.1$ is the spring constant between each pendulum, and $c = 0.1$ is the damping constant between each pendulum. We have used the small angle assumption $\sin\theta \approx \theta$ so the dynamics are linear and therefore the optimization is convex. There are no inequality constraints for the coupled pendulums. The initial states are sampled uniformly from $\mathcal{U}[-\pi, \pi]$. Finally, we considered $N = 10$ and $T = 30$.

For the coupled oscillating masses, we adapt the same benchmark system from Chen et al. (2022a) used in the non-distributed experiments. The individual state $\boldsymbol{x}_i^t \in \mathbb{R}^2$ for each agent consists of the displacement and velocity of the mass and the control $\boldsymbol{u}_i^t \in \mathbb{R}^1$ is the force acting on the mass. The dynamics matrices are

$$\boldsymbol{A}_{ii} = \begin{bmatrix} 1 & dt \\ -\frac{2k}{m}dt & 1 - \frac{2c}{m}dt \end{bmatrix}, \quad \boldsymbol{A}_{ij} = \begin{bmatrix} 0 & 0 \\ \frac{k}{m}dt & \frac{c}{m}dt \end{bmatrix}, \quad \boldsymbol{B}_i = \begin{bmatrix} 0 \\ \frac{1}{m}dt \end{bmatrix},$$

where $dt = 0.5$ is the discretization step size, $m = 1.0$ is the mass, $k = 0.4$ is the spring constant between each mass, and $c = 0.1$ is the damping constant between each mass. The initial states are sampled uniformly from $\mathcal{U}[-2.0, 2.0]$. Inequality constraints $-4 \leq \boldsymbol{x}_i^t \leq 4$ and $-0.5 \leq \boldsymbol{u}_i^t \leq 0.5$ are represented as

$$\boldsymbol{G}_x^i = \begin{bmatrix} \boldsymbol{I}_2 \\ -\boldsymbol{I}_2 \end{bmatrix}, \quad \boldsymbol{f}_x^i = 4 \cdot \boldsymbol{1}_4, \quad \boldsymbol{G}_u^i = \begin{bmatrix} 1 \\ -1 \end{bmatrix}, \quad \boldsymbol{f}_u^i = 0.5 \cdot \boldsymbol{1}_2,$$

For both the distributed MPC problems described above, the cost matrices are taken to be identity matrices: $\boldsymbol{Q}_i = \boldsymbol{I}_2$, $\boldsymbol{R}_i = \boldsymbol{I}_1$, and $\boldsymbol{P}_i = \boldsymbol{I}_2$, for all $i \in \mathcal{V}$.

The optimization (116) can be expressed in the form of (2) by defining an augmented vector consisting of the individual agent's states and controls, as well as the states and controls of its neighbors. Letting $\boldsymbol{z}_i = [\boldsymbol{x}_i^0, \boldsymbol{u}_i^0, \ldots, \boldsymbol{x}_i^T]^\top$, the augmented optimization vector for each agent $i$ is given as $\boldsymbol{x}_i^{\text{aug}} = [\boldsymbol{z}_i, \{\boldsymbol{z}_j\}_{j \in \mathcal{N}_i}]^\top$. The cost, dynamics, and constraint matrices can be augmented straightforwardly based on this new $\boldsymbol{x}_i^{\text{aug}}$. For all problems, we considered $T = 15$.

**Network flow.** The network flow problem is adapted from Mota (2013); Mota et al. (2014). We consider a directed regular graph with 200 nodes and 1000 directed edges $x_{ij} \in \mathcal{E}$. Each edge has an associated quadratic cost function $\phi_{ij}(x_{ij}) = \frac{1}{2}(x_{ij} - a_{ij})^2$, where $a_{ij}$ is sampled from $[1.0, 2.0, 3.0, 4.0, 5.0, 10.0]$ with probabilities $[0.2, 0.2, 0.2, 0.2, 0.1, 0.1]$. The objective is to optimize the flow through the graph subject to equality constraints on the flow into and out of each node. Namely, the flow into each node should be equal to the flow out of the node. For node $i$, the flow conservation constraint is $\sum_{j \in \mathcal{E}_i^-} x_{ji} = \sum_{k \in \mathcal{E}_i^+} x_{ik}$, where $\mathcal{E}_i^-$ is the set of all incoming edges to node $i$, and similarly $\mathcal{E}_i^+$ is the set of all outgoing edges from node $i$. 100 nodes are randomly selected and injected with an external flow $f_k$ sampled identically to $a_{ij}$. For each of these nodes, a reachable descendant is randomly selected and an equivalent amount of flow $f_k$ is removed from those nodes.

This problem is straightforward to express in the form (2) by considering each node as an individual agent and defining the local state vector for each agent as

$$\boldsymbol{x}_i = \begin{bmatrix} \{x_{ji}\}_{j \in \mathcal{E}_i^-} \\ \{x_{ik}\}_{k \in \mathcal{E}_i^+} \end{bmatrix}, \tag{117}$$

consisting of all the incoming and outgoing edges for node $i$. Each agent is responsible for its own flow constraint defined by

$$\boldsymbol{A}_i = \begin{bmatrix} \{1\}_{j \in \mathcal{E}_i^-} & \{-1\}_{k \in \mathcal{E}_i^+} \\ \{-1\}_{j \in \mathcal{E}_i^-} & \{1\}_{k \in \mathcal{E}_i^+} \end{bmatrix}, \quad \boldsymbol{b}_i = \boldsymbol{0}, \tag{118}$$

where $\boldsymbol{b}_i$ might instead contain the external injected or removed flow $f_i$ for that node $i$. The augmented cost matrix $\boldsymbol{Q}_i$ is zero for all incoming edges and has entries $1/2$ on the diagonal of the outgoing edges. The augmented cost vector $\boldsymbol{q}_i$ contains each of the quadratic cost offsets $a_{ik}$:

$$\boldsymbol{Q}_i = \begin{bmatrix} \{0\}_{j \in \mathcal{E}_i^-} & \\ & \{\frac{1}{2}\}_{k \in \mathcal{E}_i^+} \end{bmatrix}, \quad \boldsymbol{q}_i = \begin{bmatrix} \{0\}_{j \in \mathcal{E}_i^-} \\ \{-a_{ik}\}_{k \in \mathcal{E}_i^+} \end{bmatrix}. \tag{119}$$

Finally, we impose the constraint $-f_{\max} \cdot \boldsymbol{1} \leq \boldsymbol{x}_i \leq f_{\max} \cdot \boldsymbol{1}$ on the maximum allowed flow of all edges, with $f_{\max} = 5$.

**Distributed LASSO.** Distributed LASSO (Mateos et al., 2010) extends LASSO to situations where the training data are distributed across different agents and agents cannot share training data with each other. It can be formulated as

$$\min_{\{\boldsymbol{x}_i\}_{i=1}^N, \boldsymbol{w}} \sum_{i=1}^N \|\boldsymbol{A}_i \boldsymbol{x}_i - \boldsymbol{b}_i\|_2^2 + \frac{\lambda}{N} \|\boldsymbol{x}_i\|_1 \quad \text{s.t.} \quad \boldsymbol{x}_i = \boldsymbol{w}, \quad i = 1, ..., N \tag{120}$$

where $\boldsymbol{w} \in \mathbb{R}^{n_i}$ is a global vector of regression coefficients, $\boldsymbol{x}_i \in \mathbb{R}^{n_i}$ is a local copy of $\boldsymbol{w}$, $\boldsymbol{A}_i \in \mathbb{R}^{m_i \times n_i}$ and $\boldsymbol{b} \in \mathbb{R}^{m_i}$ are the training data available to agent $i$, and $\lambda$ is the weighting parameter. Similarly to non-distributed LASSO, this formulation is rewritten as

$$\min \sum_{i=1}^N (\boldsymbol{A}_i \boldsymbol{x}_i - \boldsymbol{b}_i)^\top (\boldsymbol{A}_i \boldsymbol{x}_i - \boldsymbol{b}_i) + \frac{\lambda}{N} \boldsymbol{1}^\top \boldsymbol{t}_i \tag{121a}$$

$$\text{s.t.} \quad \boldsymbol{t}_i \leq \boldsymbol{x}_i \leq \boldsymbol{t}_i, \quad \boldsymbol{x}_i = \boldsymbol{w}, \quad \boldsymbol{t}_i = \boldsymbol{g}, \quad i = 1, ..., N \tag{121b}$$

where $\boldsymbol{t}_i \in \mathbb{R}^{n_i}$ are newly-introduced variables and $\boldsymbol{g}$ is the global copy of $\boldsymbol{t}_i$.

The matrix $\boldsymbol{A}_i$ consists of 15% non-zero elements sampled through $\boldsymbol{A}_i^{kl} \sim \mathcal{N}(0, 1)$. The true sparse vector $\boldsymbol{v} \in \mathbb{R}^n$ to be learned consists of 50% non-zero elements sampled through $\boldsymbol{v}_i \sim \mathcal{N}(0, 1/n)$. We then construct $\boldsymbol{b} = \boldsymbol{A}\boldsymbol{v} + \boldsymbol{\xi}$ where $\boldsymbol{\xi}_i \sim \mathcal{N}(0, 1)$ represents noise in the data.

Finally, we set $\lambda = (1/5) \max_i(\|\boldsymbol{A}_i^\top \boldsymbol{b}_i\|_\infty)$. For the problems, we have $n_i = 50$ and $m_i = 5 \cdot 10^3$.

Table 2: Training and testing details for DeepQP.

| Problem Class | No of layers $K$ | Training dataset size | Epochs | Training time | Test dataset size |
|---|---|---|---|---|---|
| Random QPs | 30 | 2,000 | 500 | 21min | 1,000 |
| Random QPs with Eq. Constraints | 30 | 2,000 | 500 | 23min | 1,000 |
| Double Integrator | 30 | 500 | 600 | 28min | 1,000 |
| Osc. Masses | 15 | 500 | 600 | 48min | 1,000 |
| Portfolio Optimization | 30 | 500 | 600 | 1h 14min | 1,000 |
| LASSO | 10 | 500 | 600 | 20min | 1,000 |

Table 3: Training and testing details for DeepDistributedQP.

| Problem Class | No of layers $K$ | Training dataset size | Epochs | Training time | Test dataset size |
|---|---|---|---|---|---|
| Random QPs | 50 | 1,000 | 300 | 3h 21min | 500 |
| Random QPs with Eq. Constraints | 50 | 500 | 600 | 3h 29min | 500 |
| Coupled Pendulums | 20 | 500 | 400 | 1h 49min | 500 |
| Coupled Osc. Masses | 20 | 500 | 600 | 2h 29min | 500 |
| Network Flow | 30 | 500 | 600 | 2h 8min | 500 |
| Distributed LASSO | 20 | 500 | 600 | 56min | 500 |

## J.3 DETAILS ON TRAINING AND TESTING

Here, we discuss details regarding the training and testing of DeepQP and DeepDistributedQP in the presented experiments.

**Centralized experiments.** Table 2 shows the number of layers $K$, training dataset size, number of epochs, total training time and testing dataset size for DeepQP in every centralized problem. The increased dataset size and number of epochs for RandomQPs is motivated by the fact that the structure in these problems is less clear; learning policies that exploit this structure therefore requires more examples and takes longer. In all experiments, DeepQP was trained with a batch size of $50$ using the Adam optimizer with learning rate $10^{-3}$. The feedback layers are set as $2 \times 16$ MLPs. DeepQP and OSQP always start with zero initializations in all comparisons. The weights of the training loss were set to $\gamma_k = \exp\left((k - K)/5\right)$ in all experiments. Both the training and testing datasets are contructed after letting OSQP running until optimality.

**Distributed experiments.** Table 3 shows the number of layers $K$, training dataset size, number of epochs, total training time and testing dataset size for DeepDistributedQP in every distributed problem. In all experiments, DeepDistributedQP was trained with a batch size of $50$ using the Adam optimizer with learning rate $10^{-3}$. The feedback layers are set as $2 \times 16$ MLPs. DeepDistributedQP and DistributedQP always start with zero initializations in all comparisons. In all experiments, the weights of the training loss were set to $\gamma_k = \exp\left((k - K)/5\right)$. For the low-dimensional testing datasets, these datasets are constructed using OSQP. For larger scales, the testing dataset is constructed with DistributedQP instead as it is much faster (see Table 6), after ensuring convergence to optimality.

**Generalization bounds experiments.** These experiments were performed on a networked random QPs problem with $N = 10, n_i = 10, m_{ij} = 5, p_{ij} = 0$ and on a coupled pendulums problem with $N = 5$ and the same parameters as described in the previous section. The prior was obtained through training on a small separate dataset of $500$ problems for $50$ epochs. The posterior was then acquired through optimizing for the generalization bound with a dataset of $15,000$ problems for $50$ epochs.

## J.4 DETAILS ON STANDARD OPTIMIZERS

**Details on OSQP.** When comparing with OSQP using fixed penalty parameters, we selected the best-performing subsequence of $\{..., 0.1, 0.3, 0.5, 1.0, 3.0, 5.0, ...\}$ as the penalty parameters to plot against. Table 4 shows these parameters for every centralized problem in our experiments. For equality constraints, we scaled $\rho$ by $10^3$, as in Stellato et al. (2020). For the adaptive version, we preferred the standard heuristic adaptation rule shown in Boyd et al. (2011) with $\tau = 2.0$ and $\mu = 10.0$, instead of the OSQP adaptation scheme (Stellato et al., 2020), as it performed better in our problem instances. We hypothesize that this might be due to the fact that as scale increases the infinity norm is ignoring more information that the 2-norm. The initial $\rho^0$ was initialized as the median of the range of fixed penalty parameters.

Table 4: List of OSQP penalty parameters used in centralized experiments.

| Problem Class | List of penalty parameters $\rho$ |
|---|---|
| Random QPs | $0.1, 0.3, \ldots, 3, 10$ |
| Random QPs with Eq. Constraints | $0.1, 0.3, \ldots, 3, 10$ |
| Double Integrator | $3, 5, \ldots, 100, 300$ |
| Osc. Masses | $0.1, 0.3, \ldots, 3, 10$ |
| Portfolio Optimization | $3, 5, \ldots, 100, 300$ |
| LASSO | $30, 50, \ldots, 1000, 3000$ |

Table 5: List of DistributedQP penalty parameters used in distributed experiments.

| Problem Class | List of penalty parameters $\rho$ and $\mu$ |
|---|---|
| Random QPs | $0.1, 0.3, \ldots, 3, 10$ |
| Random QPs with Eq. Constraints | $0.1, 0.3, \ldots, 3, 10$ |
| Coupled Pendulums | $0.1, 0.3, \ldots, 3, 10$ |
| Coupled Osc. Masses | $0.1, 0.3, \ldots, 3, 10$ |
| Network Flow | $0.1, 0.3, \ldots, 3, 10$ |
| Distributed LASSO | $30, 50, \ldots, 1000, 3000$ |

**Details on DistributedQP.** The range of fixed penalty parameters to compare with was chosen using the same methodology as with OSQP. Table 5 shows these parameters for every distributed problem in our experiments. For the adaptive version, we used the standard heuristic adaptation rule shown in Boyd et al. (2011) with $\tau = 2.0$ and $\mu = 10.0$. The initial value was again always chosen as the median value of the above lists.

## J.5 DETAILS ON WALL-CLOCK TIMES

In Table 6, we list the observed wall-clock times for DeepDistributedQP (ours), DistributedQP (ours) and OSQP using either the indirect or the direct method. The table presents all six studied problems with an increasing dimension. As clearly observed, DeepDistributedQP and DistributedQP demonstrate a substantially more favorable scalability than OSQP. In fact, the two algorithms can efficiently solve problems that OSQP cannot handle due to memory overflow on our system. Finally, DeepDistributedQP also maintains a clear advantage over its standard optimization counterpart DistributedQP across all experiments which signifies the importance of learning policies for the algorithm parameters.

## K ADDITIONAL EXPERIMENTS

The following experiments are dedicated into providing additional insight on exploring the performance of DeepDistributedQP and DeepQP in various testing scenarios.

## K.1 VARYING TRAINING DATASET SIZE

This section provides additional insight on the amount of training data required for the proposed learned optimizers to perform well.

**Centralized experiments.** In Fig. 8, we compare the performance of DeepQP on the centralized problems using a varying training dataset size 500, 1000 or 2000. To ensure an "equivalent total training effort", we train these three cases for $4e$, $2e$ and $e$ epochs, respectively, where $e = 500$ for random QPs and $e = 150$ for all other problems. This comparison highlights the robust performance of DeepQP even with a limited amount of training data. Interestingly, we also observe that training with less data but over more epochs had a beneficial effect on two out of six problems. We hypothesize that this could be attributed to the non-convex nature of training in deep learning, as well as the possibility that additional epochs might have allowed for further improvements in cases where the training of the model had not yet fully converged. Overall, we conclude that DeepQP maintains reliable performance even when training data is limited.

**Distributed experiments.** For the training of DeepDistributedQP, a limited training dataset of 500 sample problems was used for all problems except for the random QPs without equality constraints where we used 1000 problems (see Table 3). For completeness, Fig. 9 presents a performance comparison of the learned optimizer when trained with 500 sample problems (600 epochs) and 1000

Table 6: **Wall-clock times and iterations for DeepDistributedQP, DistributedQP, OSQP (indirect) and OSQP (direct).** This comparison shows the total wall-clock times for DistributedQP and OSQP (indirect or direct method) required to reach the same accuracy as DeepDistributedQP. For OSQP with direct method, we only report the time for the first iteration, assuming the best-case scenario in which the factorized KKT matrix can be reused for all subsequent iterations. Both DeepDistributedQP and DistributedQP demonstrate orders-of-magnitude improvements compared to OSQP as scale increases. In addition, DeepDistributedQP maintains a significant advantage over its standard optimization counterpart in all cases.

| | | | | DeepDistrQP (ours) | | DistrQP (ours) | | OSQP (Indirect) | | OSQP (Direct) | |
|---|---|---|---|---|---|---|---|---|---|---|---|
| | | | | | | | | | | | |
| $N$ | $n$ | $m$ | nnz($Q,A$) | Time | Iters | Time | Iters | Time | Iters | Time (1st iter.) | Iters |
| **Networked Random QPs** | | | | | | | | | | | |
| 16 | 160 | 120 | 4,000 | 33.05 ms | 50 | 141.9 ms | 208 | 46.16 ms | 29 | **0.86 ms** | 29 |
| 64 | 640 | 560 | 17,600 | 39.11 ms | 50 | 129.2 ms | 192 | 185.1 ms | 28 | **23.8 ms** | 28 |
| 256 | 2,560 | 2,400 | 73,600 | 50.21 ms | 50 | 128.8 ms | 168 | 514 ms | 23 | 703.5 ms | 23 |
| 1,024 | 10,240 | 9,920 | 300,800 | **62.68 ms** | 50 | 158.9 ms | 165 | 3.03s | 23 | 8.20 s | 23 |
| **Networked Random QPs with Equality Constraints** | | | | | | | | | | | |
| $N$ | $n$ | $m$ | nnz($Q,A$) | Time | Iters | Time | Iters | Time | Iters | Time (1st iter.) | Iters |
| 16 | 160 | 168 | 4,960 | 37.21 ms | 50 | 138.9 ms | 170 | 36.52 ms | 19 | **0.76 ms** | 19 |
| 64 | 640 | 560 | 17,600 | 57.76 ms | 50 | 238.1 ms | 172 | 109.0 ms | 17 | **26.9 ms** | 17 |
| 256 | 2,560 | 2,400 | 73,600 | 74.54 ms | 50 | 239.5 ms | 164 | 692.5 ms | 17 | 956.0 ms | 17 |
| 1,024 | 10,240 | 9,920 | 300,800 | **82.55 ms** | 50 | 371.0 ms | 172 | 5.83 s | 16 | 11.60 s | 16 |
| **Coupled Pendulums Optimal Control** | | | | | | | | | | | |
| $N$ | $n$ | $m$ | nnz($Q,A$) | Time | Iters | Time | Iters | Time | Iters | Time (1st iter.) | Iters |
| 10 | 470 | 640 | 3,690 | 50.99 ms | 20 | 89.81 ms | 35 | 49.46 ms | 8 | **4.95 ms** | 8 |
| 20 | 940 | 1,200 | 7,500 | **66.44 ms** | 20 | 116.7 ms | 35 | 372.0 ms | 8 | 199.7 ms | 8 |
| 50 | 2,350 | 3,200 | 18,930 | **75.9 ms** | 20 | 142.1 ms | 34 | 948.8 ms | 8 | 4.38 s | 8 |
| 100 | 4,700 | 6,400 | 37,980 | **101.9 ms** | 20 | 201.9 ms | 35 | 3.97 s | 9 | 19.91 s | 9 |
| 200 | 9,400 | 12,800 | 76,080 | **146.0 ms** | 20 | 284.8 ms | 34 | 22.41 s | 8 | 90.07 s | 8 |
| 500 | 23,500 | 32,000 | 190,380 | **204.3 ms** | 20 | 379.8 ms | 36 | 112.9 s | 9 | Out of memory | |
| 1,000 | 47,000 | 64,000 | 380,880 | **317.2 ms** | 20 | 628.2 ms | 34 | Out of memory | | Out of memory | |
| **Coupled Oscillating Masses Optimal Control** | | | | | | | | | | | |
| $N$ | $n$ | $m$ | nnz($Q,A$) | Time | Iters | Time | Iters | Time | Iters | Time (1st iter.) | Iters |
| 10 | 470 | 1,580 | 4,590 | **48.22 ms** | 20 | 73.58 ms | 33 | 79.1 ms | 9 | 178.4 ms | 9 |
| 20 | 940 | 3,160 | 9,300 | **67.93 ms** | 20 | 91.53 ms | 33 | 641.9 ms | 9 | 2.37 s | 9 |
| 50 | 2,350 | 7,900 | 23,430 | **73.92 ms** | 20 | 97.34 ms | 32 | 1.07 s | 8 | 28.1 s | 8 |
| 100 | 4,700 | 15,800 | 46,980 | **91.93 ms** | 20 | 148.8 ms | 33 | 5.45 s | 8 | 132 s | 8 |
| 200 | 9,400 | 31,600 | 94,080 | **109.4 ms** | 20 | 194.4 ms | 34 | 31.8 s | 8 | 614 s | 8 |
| 300 | 28,200 | 47,400 | 141,180 | **132.8 ms** | 20 | 304.8 ms | 33 | 243 s | 8 | Out of memory | |
| **Network Flow** | | | | | | | | | | | |
| $N$ | $n$ | $m$ | nnz($Q,A$) | Time | Iters | Time | Iters | Time | Iters | Time (1st iter.) | Iters |
| 20 | 100 | 140 | 600 | 6.80 ms | 30 | 10.68 ms | 50 | 9.51 ms | 15 | **0.59 ms** | 15 |
| 50 | 250 | 350 | 1,500 | 7.81 ms | 30 | 13.17 ms | 48 | 14.81 ms | 16 | **1.30 ms** | 16 |
| 200 | 1,000 | 1,400 | 6,000 | **12.08 ms** | 30 | 17.61 ms | 42 | 208.19 ms | 17 | 61.93 ms | 17 |
| 500 | 2,500 | 3,500 | 15,000 | **13.63 ms** | 30 | 19.73 ms | 40 | 425.7 ms | 17 | 745.2 ms | 17 |
| 1,000 | 5,000 | 7,000 | 30,000 | **20.51 ms** | 30 | 31.62 ms | 40 | 8.73 s | 18 | 11.59 s | 18 |
| 2,000 | 10,000 | 14,000 | 60,000 | **29.86 ms** | 30 | 47.22 ms | 40 | 51.6 s | 18 | 73.9 s | 18 |
| 5,000 | 25,000 | 35,000 | 150,000 | **61.23 ms** | 30 | 85.99 ms | 39 | 558 s | 18 | Out of memory | |
| **Distributed LASSO** | | | | | | | | | | | |
| $N$ | $n$ | $m$ | nnz($Q,A$) | Time | Iters | Time | Iters | Time | Iters | Time (1st iter.) | Iters |
| 10 | 1,100 | 3,000 | 29,000 | **15.06 ms** | 20 | 28.57 ms | 37 | 2.04 s | 33 | 148.2 ms | 33 |
| 50 | 5,500 | 15,000 | 145,000 | **24.92 ms** | 20 | 44.27 ms | 38 | 13.74 s | 31 | 49.21 s | 31 |
| 100 | 10,100 | 30,000 | 290,000 | **30.51 ms** | 20 | 51.44 ms | 35 | 85.92 s | 32 | 342.9 s | 32 |
| 200 | 20,100 | 60,000 | 580,000 | **40.88 ms** | 20 | 76.21 ms | 36 | 418.9 s | 32 | Out of memory | |
| 500 | 50,100 | 150,000 | 1,450,000 | **69.19 ms** | 20 | 130.24 ms | 35 | Out of memory | | Out of memory | |

sample problems (300 epochs). While additional training data provides some improvement, the model trained with less sample problems still significantly outperforms the standard optimization counterparts.

## K.2 CAN POLICIES TRAINED FOR SPECIFIC PROBLEMS BE APPLIED TO OTHER PROBLEMS?

The field of learning-to-optimize primarily focuses on improving the performance of an underlying optimizer for problems drawn from the same distribution as the training data (Shlezinger et al., 2022). However, this prompts an interesting question: How would a policy perform if trained on a specific problem class and then evaluated on a different one?

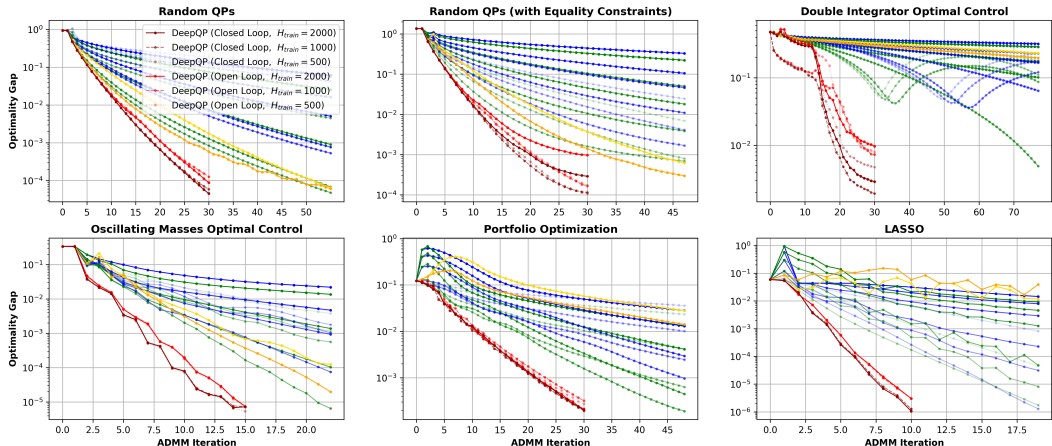

Figure 8: **Varying training dataset size for DeepQP.** The performance of DeepQP remains robust (for both open-loop and closed-loop policies) even as the training dataset size is reduced.

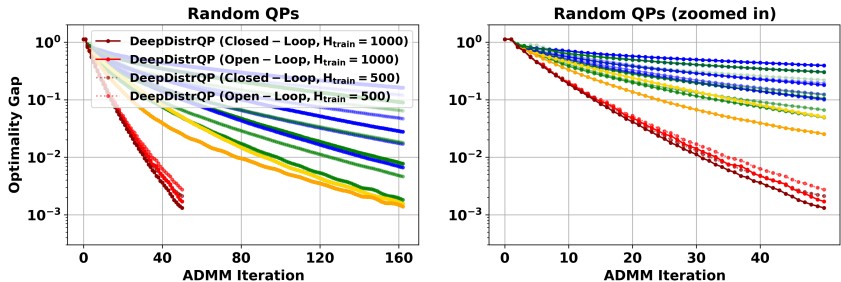

Figure 9: **Performance of DeepDistributedQP on random QPs using training dataset size** 500 **and** 1000**.** While using more training data results in a slight improvement in the performance of DeepDistributedQP, in both scenarios, the proposed learned optimizer consistently outperforms the traditional one. The right figure illustrates only the first 50 iterations.

At this point, we wish to emphasize the following fact:

*The proposed DeepDistributedQP framework already surpasses the expected capabilities of typical learning-to-optimize algorithms in the literature, as it is **trained on small-scale problems** and then successfully **deployed on much higher-dimensional ones**.*

For completeness, we also conduct curiosity-driven experiments, where we apply the learned policies to different classes of problems than the ones used for training. In Fig. 10, we test a policy trained on small-scale random equality-constrained QPs on large-scale random QPs without equality constraints and large-scale coupled pendulums problems. In the first case, DeepDistributedQP maintains remarkable performance compared to DistributedQP due to the existing similarity between the two classes. In the second setup, where the training and testing problems are entirely different, the performance is suboptimal. Overall, these results highlight that when there is a degree of similarity between the training and testing setups, DeepDistributedQP is expected to still perform very well. In future work, we plan to explore extensions trained on a broader variety of problem classes to improve generalization on entirely different setups.

### K.3    VARYING THE NUMBER OF LAYERS IN TESTING DEEPDISTRIBUTEDQP

Another natural question that arises is how DeepDistributedQP can be adapted to run for more iterations than the number of layers it was originally trained for. A straightforward modification is to repeat the last layer of the framework for the additional iterations. In Fig. 11, we add 30 extra iterations for the random QPs and 20 for the other problems. For all cases, the closed-loop policies

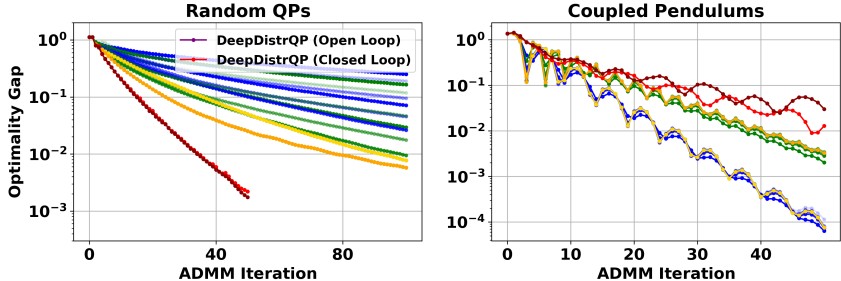

Figure 10: **Testing policies on different classes of problems.** We evaluate the policy trained on small-scale random equality-constrained QP problems ($N = 16$) on two large-scale scenarios of different problem types: random QPs without equality constraints ($N = 1,024$) and coupled pendulums ($N = 1,000$). Notably, in the first case (left), the policy demonstrates strong performance which is attributed on the fact that there is still some similarity between the training and testing setups. In the second case (right), where the testing problems differ entirely from the training setup, the performance of the learned optimizer becomes suboptimal.

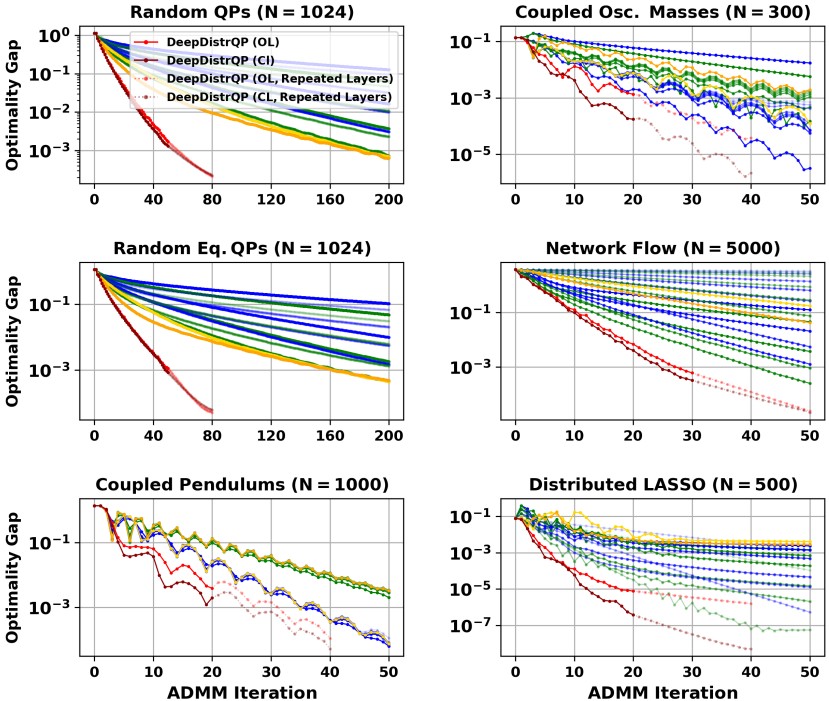

Figure 11: **Varying the number of layers while testing DeepDistributedQP.** If additional iterations are needed, DeepDistributedQP maintains strong performance by repeating its last layer for these extra iterations. Specifically, we explore adding 30 iterations for the random QPs and 20 for the rest of the problems. In all cases, the closed-loop policies continue to demonstrate superior performance, while in 4 out of 6 problems, the open-loop policies also remain advantageous.

continue to outperform the standard optimizers. Additionally, the open-loop policies maintain strong performance in 4 out of 6 problems. In future work, we plan to incorporate the repetition of the last layer during training to further ensure robust performance when additional iterations are required.

