# OpenReview forum: "Deep Distributed Optimization for Large-Scale Quadratic Programming"
_ICLR.cc/2025/Conference — ICLR 2025 Poster_

### Official Review · Reviewer_EEUg · 2024-10-27

**Soundness:** 3
**Presentation:** 3
**Contribution:** 1
**Rating:** 5
**Confidence:** 5

**Summary:**

This paper proposes two frameworks -- Distributed Quadratic Programming (DistributedQP) and Deep Distributed Quadratic Programming (DeepDistributedQP). The former combines the state-of-the-art operator splitting method in optimization with distributed optimization, and the latter further introduces deep neural networks into the framework. The authors theoretically prove the convergence guarantees and the generalization bounds of the algorithms. Some numerical simulations are provided to empirically verify the effectiveness of the algorithms.

**Strengths:**

1. The organization of the paper is clear -- the authors first introduce distributed version of operator splitting, and then combine explain how to incorporate deep learning techniques into the proposed framework to solve optimization problems more effectively.

2. The theoretical proof is well-structured and easy to follow.

**Weaknesses:**

Major:

1. (Lack of Motivation) In DEEPDISTRQP framework, the authors aimed to combine distributed optimization, in particular distributed quadratic programming, with deep learning. However, it lacks motivation of why one should study this type of problems. It is true that distributed QP part has the advantage of interpretability, while the deep learning part has stronger generalization capabilities. However, the experiments are only limited to relatively small-scale problems like optimal control, portfolio optimization, etc. It is unclear if this framework can handle large-scale deep learning problems effectively, and thus a optimization journal, instead of a machine learning conference, might be a better fit for this paper.

2. (Lack of Novelty) The design of the frameworks as well as the theoretical analysis are more like a combination of existing techniques, such as local and global updates, primal dual updates, the generalization bounds, etc. It is unclear if there is any novelty of simply combining these techniques, especially provided that the experiments are only conducted on some simple ones.

3. (Unfair Comparison in Experiments) All figures of the experiments section have the iteration number as the horizontal axis, and the curves showcase better performance of DeepDistributedQP. However, different model architectures have different per-iteration complexities, and thus it is unfair to simply plot the optimality gap vs iteration numbers.

Minor:

1. The term 'OSQP' is first introduced in the abstract without additional context. It is unclear what OS is for the audience outside of optimization community.

**Questions:**

I am wondering if the authors could explain the motivation and the novelty of this paper.

---

> ### Author Response · Authors · 2024-11-29
> **Response to Reviewer EEUg (part 1/4)**
>
> We wish to sincerely thank the reviewer for their valuable feedback, which has highlighted areas where the motivation and novelty of our work required further clarification. In response, we have significantly revised the introduction of our paper (Section 1) to express more accurately the motivation and critical importance of developing effective large-scale quadratic programming methods, as the scale of modern applications increases in all fields. In addition, we have addressed the reviewer’s concerns regarding the scale of the illustrated problems, by significantly scaling our experiments to much higher dimensional problems with up to 150K variables and 50K constraints, while training exclusively on much lower-dimensional ones.
>
> Overall, the reviewer’s feedback has been instrumental in enhancing the quality of our paper and in enabling us to express the core concepts of our work with greater clarity. In the revised version of our paper, we have carefully taken into account the reviewer’s feedback and have thoroughly addressed all their points and questions.
>
> ## Weaknesses
>
> 1. Given that this point raised by the reviewer critiques both the motivation for this work and the scale of the problems addressed in the results, we have structured our response into the following two parts:
>
>    **Motivation.** We really thank the reviewer for pointing out that the motivation of this work was not clearly conveyed in the introduction. In the revised version, we have significantly restructured this section to better highlight the critical importance of effectively addressing large-scale QP problems in modern applications, the challenges distributed constrained optimization methods face at such scales, and how our novel deep unfolding methodology effectively tackles these challenges. Similarly, we have also updated the first two sentences of the abstract to better motivate the importance of developing methods for large-scale QP problems.
>
>    For convenience, we also provide a high-level presentation of the key ideas presented in the updated introduction section here:
>
>    - **Why is quadratic programming (QP) so important?** QP serves a fundamental role in optimization with a wide variety of applications including machine learning, robotics and control, signal processing, finance, transportation networks and many other fields (please see references in first paragraph). On top of its direct applications, QP is also the core component of many other advanced non-convex optimization algorithms such as sequential quadratic programming, trust region methods, augmented Lagrangian methods, mixed-integer optimization and others. Consequently, the development of effective QP methods is of paramount importance across a vast array of applications.
>
>     - **Why is it vital to develop effective methods for large-scale quadratic programming?** Due to the fundamental role of QP in optimization and the growing demand for applying optimization methods to large-scale problems, this immediately creates a necessity for developing QP methods capable of handling problems with thousands of variables and constraints, and potentially much more, within limited computational time. Such applications include sparse regression problems, decentralized multi-agent control, power grids, resource allocation, image processing and many other fields (please see references in second paragraph).
>
>     - **What are the current challenges distributed constrained optimization algorithms are facing?** Distributed optimization methods have emerged as effective algorithms for handling large-scale problems by leveraging underlying network/decentralized structure to enable parallel computations. Within this context, distributed ADMM has become a widely adopted approach for designing distributed optimization algorithms. Nevertheless, such algorithms typically require a significant amount tuning by experts, which must also factor in the computational budget (number of iterations), resulting in often being infeasible for large-scale optimization problems. Furthermore, manually tuned algorithms frequently fail to generalize to new, unseen problems, even within the same problem class.

---

> ### Author Response · Authors · 2024-11-29
> **Response to Reviewer EEUg (part 2/4)**
>
> 1. (continued) **Why are distributed constrained optimization and deep unfolding such a good fit?** Our key insight is that **deep unfolding is particularly well-suited for overcoming the limitations of distributed constrained optimization, as it can eliminate the need for extensive tuning, manage iteration restrictions and enhance generalization.** However, to our best knowledge, its combination with distributed ADMM has only recently been explored in [R1], relying on a relatively simple setup that studies unconstrained problems, assumes local updates consisting of gradient steps, focuses solely on parameter tuning, and is not accompanied by any formal performance guarantees. Our work tackles some of the main challenges of distributed constrained optimization, by introducing a novel deep learning-aided distributed optimization architecture for solving large-scale constrained QP problems. To our best knowledge, this is the first work to propose a learning-based architecture for distributed constrained optimization using ADMM, despite its widespread popularity.
>
> **Ability to scale for large-scale problems.** We wish to really thank the reviewer for prompting us to place more emphasis on scaling our framework to larger problem sizes. In the revised version of our paper, **we have effectively applied DeepDistributedQP to problems with up to 50K variables and 150K constraints.** The detailed dimensions of the problems are provided in Tables and 6. For convenience, we also repeat Table 1 here:
>
> | Problem Class            | N (train)   | n (train)    | m (train)    | nnz(Q, A) (train)  | N (test) | n (test) | m (test) | nnz(Q, A) (test) |
> |--------------------------|-----|------|-------|-------------|---------------|---------------|---------------|-----------------------|
> | Random QPs              | 16  | 160  | 120   | 4,000       | 1,024         | 10,240        | 9,920         | 300,800              |
> | Random QPs w/ Eq. Constr.| 16  | 160  | 168   | 4,960       | 1,024         | 10,240        | 9,920         | 300,800              |
> | Coupled Pendulums        | 10  | 470  | 640   | 3,690       | 1,000         | 47,000        | 64,000        | 380,880              |
> | Coupled Osc. Masses      | 10  | 470  | 1,580 | 4,590       | 300           | 28,200        | 47,400        | 141,180              |
> | Network Flow             | 20  | 100  | 140   | 600         | 5,000         | 25,000        | 35,000        | 150,000              |
> | Distributed LASSO        | 10  | 1,100| 3,000 | 29,000      | 500           | 50,100        | 150,000       | 1,450,000            |
>
> where on the left, we illustrate the dimensions used for training and on the right, the maximum dimensions used for testing. The terms N, n and m correspond to the total number of nodes, variables and constraints, respectively. The metric nnz(Q,A) represents the total amount of non-zero elements in the cost matrix Q and constraint matrix A.
>
> **These results highlight the ability of DeepDistributedQP to generalize well for large-scale problems, while being trained exclusively on much lower-dimensional ones.**
>
> We have also included a wall-clock time comparison in Fig. 1, 6 and Table 6 comparing the total computational time needed from DeepDistributedQP, DistributedQP and OSQP for reaching the same level of accuracy. For convenience, we repeat a part of Table 6 here, e.g. for the network flow problem:
>
> | N    | n      | m      | nnz(Q, A)   | DeepDistributedQP (Ours)        | DistributedQP (Ours)        | OSQP (indirect)      | OSQP (direct, 1st iter.) |
> |------|--------|--------|-------------|-------------|-------------|------------|------------------|
> | 20   | 100    | 140    | 600         | 6.80 ms     | 10.68 ms    | 9.51 ms    | **0.59 ms**      |
> | 50   | 250    | 350    | 1,500       | 7.81 ms     | 13.17 ms    | 14.81 ms   | **1.30 ms**      |
> | 200  | 1,000  | 1,400  | 6,000       | **12.08 ms**| 17.61 ms    | 208.19 ms  | 61.93 ms         |
> | 500  | 2,500  | 3,500  | 15,000      | **13.63 ms**| 19.73 ms    | 425.7 ms   | 745.2 ms         |
> | 1,000| 5,000  | 7,000  | 30,000      | **20.51 ms**| 31.62 ms    | 8.73 s     | 11.59 s          |
> | 2,000| 10,000 | 14,000 | 60,000      | **29.86 ms**| 47.22 ms    | 51.6 s     | 73.9 s           |
> | 5,000| 25,000 | 35,000 | 150,000     | **61.23 ms**| 85.99 ms    | 558 s      | *Out of memory*  |
>
> **The provided exhibition clearly demonstrates the superior scalability of DeepDistributed method as the dimension of the problems increases.** In particular, the proposed distributed learning-to-optimize framework can effectively address problems that OSQP would require a substantially higher amount of time or memory for reaching the same accuracy.
>
> Again, we are truly thankful to the reviewer for prompting us to scale our framework to much higher-dimensional problems. We believe that their suggestion and the newly added results have been instrumental in improving the quality of our paper.

---

> ### Author Response · Authors · 2024-11-29
> **Response to Reviewer EEUg (part 3/4)**
>
> 2. We sincerely thank the reviewer for their feedback and for encouraging us to further emphasize the significant novelty of our work. To address their concerns, we have provided greater clarity in articulating the contributions of our work in Section 1 and have highlighted their distinctions from the existing literature in Section 2.
>
>    For the reviewer’s convenience, we also summarize the novelty of our work through the following key points:
>
>      - **We introduce DistributedQP, a novel decentralized method for tackling large-scale QP problems.** The proposed new method combines the state-of-the-art OSQP solver with a consensus approach towards extending the capabilities of the popular solver in effectively handling high-dimensional problems. Additionally, we have rigorously established convergence guarantees to optimality for this new method, ensuring its reliability.
>
>      - **We then introduce DeepDistributedQP, a novel deep learning-aided distributed architecture for large-scale QP.** The proposed methodology relies on the idea of unrolling the iterations of the new DistributedQP method in a supervised manner, learning feedback policies for the underlying algorithm parameters. In addition, we effectively integrate implicit differentiation into our framework by recognizing that using the indirect method in OSQP/DistributedQP leads to a fixed-point equation that can be exploited for enhancing computational time and memory efficiency during training. **The presented results in Fig. 1, 6 and Table 6 highlight that DeepDistributedQP can effectively handle large-scale QP problems with up to 50K variables and 150K constraints, that the state-of-the-art OSQP method struggles to solve due to high computational time or memory overflow issues. Another key novelty of our work that needs to be emphasized is that DeepDistributedQP is trained exclusively on small-scale problems. Yet, as shown in Tables 1, 6 and Fig. 6, the proposed framework can tackle much-higher dimensional problems than the ones used for training. This fact constitutes a key step in providing effective distributed learning-to-optimize methods for large-scale optimization.**
>
>     - **This is the first work to explore the deep unfolding of distributed ADMM for large-scale constrained optimization.** As also explained earlier in our response, despite the widespread popularity of distributed ADMM for large-scale constrained optimization (the highly infliential ADMM tutorial by Boyd et al. [R2] has over 23K citations), to our best knowledge, its interpretation as a deep unfolded network was only been recently explored in [R1]. Yet this work explored a much simpler setup than ours, studying unconstrained problems, assuming local updates consisting of gradient steps, focusing solely on parameter tuning, and not providing any formal performance guarantees. As a result, our work is the first to propose a distributed deep learning-aided architecture based on ADMM for large-scale constrained optimization.
>
>     - **We establish novel performance guarantees for learned optimizers through PAC-Bayes theory.** We prove that the learned distributed solver generalizes to new problems sampled from the distribution of training problems via PAC-Bayes theory and describe how this generalization bound provides a numerical certificate on its performance. The proposed bounds are entirely different than the existing ones for learning-to-optimize methods in the literature. In particular, [R3] recently presented generalization bounds for learned optimizers but their approach is completely different relying on a binary error function. The bounds we are presenting are on the expected optimality gap on the final iterate which we believe is a more informative measure of progress. In addition, the works in [R4] and [R5] have also investigated generalization bounds for learned optimizers, but under the restrictions that the update function as a gradient step or a multi-layer perceptron, respectively. In contrast, it important to highlight that the bounds we are proposing are applicable to any learning-to-optimize method and do not assume any specific structure in the updates.
>
>     With the updated introduction and related work sections in our revised paper, along with this detailed exposition, we sincerely hope that the reviewer recognizes and appreciates the significant and multi-faceted novelty of our work. We remain fully available to provide any additional clarification the reviewer may require.

---

> ### Author Response · Authors · 2024-11-29
> **Response to Reviewer EEUg (part 4/4)**
>
> 3. We really thank the reviewer for pointing out that this part was not well clarified in our initial submission. In fact, Figures 5 and 6 (left) compare algorithms that do have the **same per-iteration complexity.** This is because **DeepQP is the deep unfolded version of OSQP, and DeepDistributedQP is the deep unfolded version of DistributedQP.**
>
>      - In particular, the most computationally expensive part of OSQP is solving Eq. (139) with the indirect method. The same stands of course for DeepQP as its only extra computation is forward propagating through the feedback MLPs of size 2x16. These operations are of course extremely cheap matrix multiplications and element-wise operations – that are highly parallelizable. As a result, the per-iteration complexity of both DeepQP and OSQP is identical.
>
>      - Similarly, DeepDistributedQP and DistributedQP have the same peri-iteration complexity coming from the update (11). Again, the only extra computation the former requires is the forward propagation of 2x16 which is negligible compared to update (11).
>
>     **Consequently, the compared pairs of algorithms have identical per-iteration complexities, and as a result, using the iterations number in the horizontal axis is a fair comparison.** Again, we are really thankful to the reviewer for pointing out that this part was not clarified well enough in the original version of our paper. To avoid any such confusion, we are now explicitly stating in Section 6 that these pairs of algorithms have the same per-iteration complexity.
>
> ## Questions
>
> We are really grateful to the reviewer for highlighting that the motivation and novelty of our work were not explained clearly enough in the original submission. In the updated version of our paper as well as in the above responses, we have emphasized the **paramount importance of developing effective methods for large-scale QP problems** and the great need for **addressing key challenges in distributed constrained optimization through deep unfolding**. We have also highlighted in a much clearer manner the significant and multi-faceted novelty of our work through presenting **a new distributed QP optimizer**, **a novel deep learning-aided architecture for large-scale quadratic optimization that is trained exclusively on small problems**, and **novel performance guarantees for learning-to-optimize methods through PAC-Bayes theory.**
>
> Overall, the reviewer's comments were illuminating for recognizing that the motivation and novelty of our work were not adequately explained in our initial submission. In addition, the reviewer's feedback on the scale of the presented problems was truly valuable for better demonstrating the applicability of the proposed framework on addressing large-scale problems with up to 150K variables and 50K in our revised submission. As a result, the provided review had a great impact in improving the quality of our paper.
>
> Given that the revised paper and our responses have fully addressed the points raised in your review, we would be truly grateful if you would reconsider the evaluation of our paper and recommend it for acceptance. Again, we are really thankful for your thorough and constructive criticism and we remain at your disposal for any additional clarifications you might require.
>
> [R1] Noah, Y., & Shlezinger, N. (2024). Distributed Learn-to-Optimize: Limited Communications Optimization over Networks via Deep Unfolded Distributed ADMM. IEEE Transactions on Mobile Computing.
>
> [R2] Boyd, S., Parikh, N., Chu, E., Peleato, B., & Eckstein, J. (2011). Distributed optimization and statistical learning via the alternating direction method of multipliers. Foundations and Trends® in Machine learning, 3(1), 1-122.
>
> [R3] Sambharya, R., & Stellato, B. (2024). Data-Driven Performance Guarantees for Classical and Learned Optimizers. arXiv preprint arXiv:2404.13831.
>
> [R4] Sucker, M., & Ochs, P. (2023, April). PAC-Bayesian learning of optimization algorithms. In International Conference on Artificial Intelligence and Statistics (pp. 8145-8164). PMLR.
>
> [R5] Sucker, M., Fadili, J., & Ochs, P. (2024). Learning-to-Optimize with PAC-Bayesian Guarantees: Theoretical Considerations and Practical Implementation. arXiv preprint arXiv:2404.03290.

---

> > ### Author Response · Authors · 2024-12-02
> > **Follow-up to Reviewer EEUg**
> >
> > As we are approaching the end of the discussion period today, we would appreciate the reviewer to clarify if the revised version of our paper and our responses have sufficiently addressed their concerns. We deeply value the reviewer's insightful feedback, which has been instrumental in identifying key areas of our paper that required greater emphasis. In our response, we have highlighted the vital importance of developing effective large-scale QP methods, emphasized the significant and multi-faceted novelty of this work, demonstrated the scalability of our framework to high-dimensional problems with up to 50K variables and 150K constraints,  and addressed all additional points and questions raised. If the reviewer agrees that we have satisfactorily addressed their feedback, we would greatly appreciate their reconsideration of our rating to reflect these improvements.

---

### Official Review · Reviewer_Y4sR · 2024-11-01

**Soundness:** 3
**Presentation:** 2
**Contribution:** 2
**Rating:** 3
**Confidence:** 4

**Summary:**

This paper combines the interpretability of distributed optimization with the strong empirical performance of deep neural networks to enhance optimization efficiency. The authors propose a distributed method for QPs with theoretical convergence guarantees. They then unfold this method into a deep neural network, leveraging it to accelerate QP solutions. The paper includes both theoretical proofs and numerical results to demonstrate effectiveness. The author claim that the proposed approach outperforms conventional optimization softwares.

**Strengths:**

The mathematical aspects of the paper are technically correct.

**Weaknesses:**

The primary concern revolves around the novelty of the unfolded network.
A similar approach can be found in [1], which also predicts parameters for ADMM in conventional OSQP using reinforcement learning, where this work seems to be a straightforward substitution of tuning efforts with basic MLPs.
This also raises questions about the fairness and completeness of the experiments, as the authors only compare their method against OSQP without considering other learning-based frameworks that could potentially yield better performance.
Additionally, the presentation in Section 6 is somewhat difficult to follow. For example, the implementation of the performance metrics is not clearly explained, and it would be helpful for the authors to provide more details on how improvements are quantified. Another minor issue is that some notations are quite confusing and not adequately introduced

[1]Jeffrey Ichnowski, Paras Jain, Bartolomeo Stellato, Goran Banjac, Michael Luo, Francesco Borrelli, Joseph E. Gonzalez, Ion Stoica, and Ken Goldberg. Accelerating quadratic optimization with reinforcement learning, 2021..

**Questions:**

Questions:
In addtion to those mentioned in weakness, there are some specifc questions regarding this paper.
1. What does r^k_i in Figure 1 and subsection **Learning feedback policies** represent? This notation is not defined elsewhere in the paper. Additionally, a complete update formulation would greatly aid in understanding the unfolding process.
1. The author mentions that distributed methods are effective for large-scale decision-making; however, the tested problems are relatively small. Can the authors discuss the feasibility of extending the numerical results to larger problem sizes?
3. The specific number of layers used in the neural network is not mentioned. From the description, I assume it corresponds to $K$. Do these layers share the same weights? Additionally, how does DEEPDISTRQP perform with varying numbers of layers?
4. Why does Figure 6 use the relative optimality gap as a metric, while other figures utilize the absolute optimality gap? What is the rationale behind this choice?

---

> ### Author Response · Authors · 2024-11-28
> **Response to Reviewer Y4sR (part 1/4)**
>
> We wish to sincerely thank the reviewer for their valuable feedback on our work, which has been instrumental in improving our paper. In the revised version and the responses below, we have placed greater emphasis on the novelty of our contributions, significantly scaled our experiments to address much higher-dimensional problems and thoroughly addressed all the points and questions raised by the reviewer.
>
> ## Response to Weaknesses
>
> > The primary concern revolves around the novelty of the unfolded network. A similar approach can be found in [1], which also predicts parameters for ADMM in conventional OSQP using reinforcement learning, where this work seems to be a straightforward substitution of tuning efforts with basic MLPs.
>
> We really thank the reviewer for their insightful comment, as it has helped us recognize the need to better articulate the novelty of our work and clearly differentiate it from existing approaches.
>
> We would like to emphasize that the main contribution presented in our paper, DeepDistributedQP, is fundamentally distinct from the RLQP method presented in [R1].
>
> In particular, **DeepDistributedQP is a distributed learned optimizer that is built upon the unfolding of the newly introduced DistributedQP optimizer – and not OSQP.** On the other hand, RLQP aims in accelerating the standard OSQP solver and as a result it is specifically targeted at **centralized quadratic programming.** While RLQP employs a multi-agent MDP setup, this is solely to facilitate its reinforcement learning problem formulation. In other words, this multi-agent setting does not allow the execution of RLQP to proceed in a distributed manner. **In contrast, DeepDistributedQP focuses on distributed quadratic optimization.**
>
> As a result, if any analogies are to be drawn between RLQP and our work, they would pertain to DeepQP, the centralized version of our approach, which is derived by unfolding OSQP. However, it is important to note that DeepQP is only a byproduct of this paper included for completeness. It is not the central contribution of our work. The primary focus of this paper lies in the development of a **deep learning-aided architecture for distributed constrained optimization**, which is embodied in the DeepDistributedQP framework.
>
> On top of that, we wish to further emphasize that **DeepQP and RLQP are fundamentally different approaches as well.**
>
> - **DeepQP is much more efficient in terms of training since it is a supervised learning framework.** By leveraging a supervising signal, our approach eliminates the need for extensive exploration of the state-action space, enabling more efficient policy learning. In contrast, formulating the parameter learning for an unfolded QP solver as a RL problem – as in RLQP -  introduces the challenge of adequately exploring the state-action space to derive good policies. Consequently, RLQP requires vast amounts of data and computational resources. This limitation was acknowledged by the authors of RLQP, who reported that training their networks took multiple days on a system with five NVIDIA V100 GPUs and 256 GB of RAM. In contrast, as shown in the newly added Table 2 in the Appendix, DeepQP required a maximum of 1 hour and 14 minutes to be trained on our system (RTX 4090 GPU 24GB and 64GB of RAM), highlighting the training efficiency of our approach.
>
> - **DeepQP explicitly captures the design requirement of finding good solutions within a fixed number of iterations.** This is well suited for many application settings where there is a limited computational budget. Similar performance may be achieved by tuning the discount factor in RLQP, but there is no explicit way to encode this design requirement in that framework.
>
> - **Finally, our framework also allows for establishing performance guarantees through the presented PAC-Bayes bounds.** It is unclear how RLQP would be accompanied by any such performance guarantees.
>
> Overall, we conclude that the RLQP method presented in [R1] is an entirely different framework than the main contribution of our paper, DeepDistributedQP. While some analogies can be drawn between DeepQP and RLQP, we hope that the above analysis clearly demonstrates that these are fundamentally distinct methodologies as well.
>
> [R1] Ichnowski, J., Jain, P., Stellato, B., Banjac, G., Luo, M., Borrelli, F., ... & Goldberg, K. (2021). Accelerating quadratic optimization with reinforcement learning. Advances in Neural Information Processing Systems, 34, 21043-21055.

---

> ### Author Response · Authors · 2024-11-28
> **Response to Reviewer Y4sR (part 2/4)**
>
> We acknowledge that we should have better emphasized these differences in our original manuscript. To address this, we have revised the relevant part in the Related Work (Section 2), to ensure that readers understand that the RLQP approach is only marginally related to the proposed methodologies. We sincerely hope that the reviewer finds our additional comments helpful in clarifying the significant distinctions between our work and RLQP. Again, we are really thankful to the reviewer for prompting us to further elaborate on these differences.
>
> > This also raises questions about the fairness and completeness of the experiments, as the authors only compare their method against OSQP without considering other learning-based frameworks that could potentially yield better performance.
>
> To our best knowledge, there is no learning-to-optimize framework for distributed constrained quadratic programming. Moreover, we are not aware of any learning-to-optimize framework that unrolls distributed constrained ADMM, making our work the first to enhance this widely used method through deep unfolding. Consequently, our comparisons for DeepDistributedQP are as thorough as possible given the novel setting considered.
>
> To further emphasize the computational benefits of DeepDistributedQP, we have included a wall-clock time comparison against OSQP (Fig. 1, 6, and Table 6), which demonstrates the orders-of-magnitude improvements our method achieves as problem scale increases.
>
> Furthermore, as mentioned in the preceding comparison, a benchmark between RLQP and DeepQP could be possible. However, RLQP is significantly less data-efficient than DeepQP, and such a comparison would take weeks of training RLQP on the available hardware. Finally, we reiterate that the main contribution of this work is DeepDistributedQP, not DeepQP, which was included primarily for completeness and illustrative purposes.
>
> >Additionally, the presentation in Section 6 is somewhat difficult to follow. For example, the implementation of the performance metrics is not clearly explained, and it would be helpful for the authors to provide more details on how improvements are quantified.
>
> We are really thankful for the valuable feedback of the reviewer that the illustration of our results was not sufficiently clear in our initial submission. To address this, we have significantly revisited the writing of Section 6, ensuring that all necessary details regarding the training and testing of the methods are now explicitly provided either within that section or in Section I of the Appendix. Additionally, we have included an additional quantification of improvements by comparing the required wall-clock time needed to achieve the same level of accuracy, further emphasizing the practical benefits of our approach. We hope these revisions address the reviewer’s concerns and provide a clearer and more comprehensive presentation of our results.
>
> > Another minor issue is that some notations are quite confusing and not adequately introduced
>
> We sincerely thank the reviewer once again for their thorough reading of our work and their valuable feedback. In the revised version of our paper, we have provided detailed clarifications for any notation that might not have been adequately addressed.
>
> ## Response to Questions
>
> 1. We thank the reviewer for pointing out that the primal and dual residual terms were not properly defined in our initial submission. We have added their detailed expressions in Appendix D. The updates of DeepDistributedQP correspond to the updates of DistributedQP, i.e., Eq. (3)-(8), unfolded over K iterations. These variables are then used within the residuals, now explicitly defined in Appendix D to obtain the algorithm parameters for the next updates, and so on. We hope that this explanation, along with the addition of the residual terms, clarifies the exposition of the updates. We remain available to the reviewer for any further clarifications needed.

---

> ### Author Response · Authors · 2024-11-28
> **Response to Reviewer Y4sR (part 3/4)**
>
> 2. We wish to really thank the reviewer for prompting us to place more emphasis on scaling our framework to larger problem sizes. In the revised version of our paper, **we have effectively applied DeepDistributedQP to problems with up to 50K variables and 150K constraints.** The detailed dimensions of the problems are provided in Tables 1 and 6. For convenience, we also repeat Table 1 here:
>
> | Problem Class            | N (train)   | n (train)    | m (train)    | nnz(Q, A) (train)  | N (test) | n (test) | m (test) | nnz(Q, A) (test) |
> |--------------------------|-----|------|-------|-------------|---------------|---------------|---------------|-----------------------|
> | Random QPs              | 16  | 160  | 120   | 4,000       | 1,024         | 10,240        | 9,920         | 300,800              |
> | Random QPs w/ Eq. Constr.| 16  | 160  | 168   | 4,960       | 1,024         | 10,240        | 9,920         | 300,800              |
> | Coupled Pendulums        | 10  | 470  | 640   | 3,690       | 1,000         | 47,000        | 64,000        | 380,880              |
> | Coupled Osc. Masses      | 10  | 470  | 1,580 | 4,590       | 300           | 28,200        | 47,400        | 141,180              |
> | Network Flow             | 20  | 100  | 140   | 600         | 5,000         | 25,000        | 35,000        | 150,000              |
> | Distributed LASSO        | 10  | 1,100| 3,000 | 29,000      | 500           | 50,100        | 150,000       | 1,450,000            |
>
> where on the left, we illustrate the dimensions used for training and on the right, the maximum dimensions used for testing. The terms N, n and m correspond to the total number of nodes, variables and constraints, respectively. The metric nnz(Q,A) represents the total amount of non-zero elements in the cost matrix Q and constraint matrix A.
>
> **These results highlight the ability of DeepDistributedQP to generalize well for large-scale problems, while being trained exclusively on much lower-dimensional ones.**
>
> We have also included a wall-clock time comparison in Fig. 1, 6 and Table 6 comparing the total computational time needed from DeepDistributedQP, DistributedQP and OSQP for reaching the same level of accuracy. For convenience, we repeat a part of Table 6 here, e.g. for the network flow problem:
>
> | N    | n      | m      | nnz(Q, A)   | DeepDistributedQP (Ours)        | DistributedQP (Ours)        | OSQP (indirect)      | OSQP (direct, 1st iter.) |
> |------|--------|--------|-------------|-------------|-------------|------------|------------------|
> | 20   | 100    | 140    | 600         | 6.80 ms     | 10.68 ms    | 9.51 ms    | **0.59 ms**      |
> | 50   | 250    | 350    | 1,500       | 7.81 ms     | 13.17 ms    | 14.81 ms   | **1.30 ms**      |
> | 200  | 1,000  | 1,400  | 6,000       | **12.08 ms**| 17.61 ms    | 208.19 ms  | 61.93 ms         |
> | 500  | 2,500  | 3,500  | 15,000      | **13.63 ms**| 19.73 ms    | 425.7 ms   | 745.2 ms         |
> | 1,000| 5,000  | 7,000  | 30,000      | **20.51 ms**| 31.62 ms    | 8.73 s     | 11.59 s          |
> | 2,000| 10,000 | 14,000 | 60,000      | **29.86 ms**| 47.22 ms    | 51.6 s     | 73.9 s           |
> | 5,000| 25,000 | 35,000 | 150,000     | **61.23 ms**| 85.99 ms    | 558 s      | *Out of memory*  |
>
> **The provided exhibition clearly demonstrates the superior scalability of DeepDistributed method as the dimension of the problems increases.** In particular, the proposed distributed learning-to-optimize framework can effectively address problems that OSQP would require a substantially higher amount of time or memory for reaching the same accuracy.

---

> ### Author Response · Authors · 2024-11-28
> **Response to Reviewer Y4sR (part 4/4)**
>
> 3. We thank the reviewer for their valuable feedback and questions. For clarity, we have explicitly added the number of layers K in Tables 2 and 3 in Appendix I – they can also be inferred from the iterations in the x-axis of Fig. 5 and 6.
>
>    These layers have different weights as indicated by the fact that the weights of the layers \theta_i^k have a superscript k. However, during the same iteration, the weights might be the same over all nodes (shared policies) or different (local policies). We have also added a comparison in the results section (Fig. 7) highlighting the difference in performance between the two approaches.
>
>     We are also thankful for the insightful suggestion to examine the performance of DeepDistributedQP with a varying number of layers. To address this question, we have included additional experiments in Section J.3 of the Appendix where we are repeating the last layer of DeepDistributedQP for extra iterations on the studied distributed problems. As shown by the results, the learned optimizer maintains its superior performance against the standard optimizer for all problems when using closed-loop policies and for 4 out of 6 problems when using open-loop ones. This shows that **this straightforward adaptation can work sufficiently well in practical scenarios where extra iterations are required**. In future work, we wish to also consider repeating the same layer over the last iterations during training, so that we further robustify performance for such cases.
>
> 4. Thank you very much for requesting this clarification. In Fig. 7 [right] (Fig. 6 in initial submission), we are using the relative optimality gap to measure progress since this is the progress metric on which the generalization bounds of Section 5 are derived on. We have added a clarification in the results section referring to this figure to ensure it is absolutely clear to the reader.
>
> We would like to sincerely thank the reviewer for their insightful comments, which helped us identify areas of our work that required further emphasis, such as clearly articulating its novelty and demonstrating the strong performance of DeepDistributedQP on much higher-dimensional QP problems than the ones used for training. As a result, the reviewer's feedback has directly contributed to a significant improvement in the quality and clarity of our paper. Given that we have addressed all points and questions raised, we would be truly grateful if you could reconsider the evaluation of our paper and recommend it for acceptance. We remain at your disposal for any additional clarifications you might require.

---

> > ### Author Response · Authors · 2024-12-01
> > **Follow-up to Reviewer Y4sR**
> >
> > As we are approaching the end of the discussion period, we would appreciate the reviewer to clarify if the revised version of our paper and our responses have sufficiently addressed their concerns. We deeply value the reviewer's insightful feedback, which has been instrumental in identifying key areas of our paper that required greater emphasis. In our response, we have highlighted the minimal similarity between our work and RLQP, demonstrated the scalability of our framework to high-dimensional problems with up to 50K variables and 150K constraints, shown the robustness of DeepDistributedQP while varying the number of layers, and addressed all additional points and questions raised. If the reviewer agrees that we have satisfactorily addressed their feedback, we would greatly appreciate their reconsideration of our rating to reflect these improvements.

---

### Official Review · Reviewer_fvD1 · 2024-11-03

**Soundness:** 3
**Presentation:** 2
**Contribution:** 3
**Rating:** 6
**Confidence:** 3

**Summary:**

The authors present an optimization technique for solving quadratic programming problems distributed across multiple compute nodes. The methods use a extend OSQP methodology to a distributed framework which has similarities to a two-block ADMM approach. The authors identify that selecting hyperparameters in this methodology can greatly influence convergence rates. As a result, the authors propose learning the hyperparameters through a deep learning model that has the iterations of the distributed OSQP algorithm embedded as layers.

**Strengths:**

Using solved QP problems to train the deep learning model, the authors are able to select optimal hyperparameters that greatly improve the convergence for solving new distributed QP problems.

**Weaknesses:**

A limitation of this approach seems to be the amount of training data required to train the neural network.

**Questions:**

1.	 Can the authors provide specific details about the dataset size used in each experiment, and comment about how much data would be needed for a user to implement this strategy.
2.	Can the authors discuss if they have separate models for each experiment, or if a single model was used across all problems. If multiple models are used, can the authors comment on the ability of models to generalize to other problems.
3.	Theorem 3 depends on a set of problems drawn from a distribution. Can the authors provide any empirical evidence or discussion on how well the bounds hold for different types of problems?
4.	The authors require knowing the maximum number of iterations, K, apriori. Generally, the maximum number of iterations is quite large with the intention that the optimization solver finds a solution before reaching the limit. Is choosing an appropriate value of K required to construct a deep neural network model that is memory and time efficient?
5.	In many of the experiments the authors evaluate the optimality gap up to the maximum number of iterations. However, often we care about the number of iterations to reach the optimum. Does this change the methodology implementation?

Some minor points are:
1.	The authors should define some terms like $r$, $\theta$ in equation (10).
2.	Can the authors provide more detail about the known mapping G on page 6, line 282.
3.	Spelling mistake on “non-distrubuted” on page 6 line 290.

---

> ### Author Response · Authors · 2024-11-28
> **Response to Reviewer fvD1 (part 1/4)**
>
> We sincerely thank the reviewer for their time and effort in thoroughly reviewing our paper and for the really interesting questions they have posed. We are grateful that the reviewer recognizes the strengths of our work and appreciates its importance. The comments and suggestions provided have been pivotal in identifying key parts that required more emphasis and clarifications, so that we further improve our paper.
>
> In the revised version of our paper and the responses provided below, we have thoroughly addressed all the points and questions you have raised.
>
> ## Response to Weakness
>
> We thank the reviewer for encouraging us to investigate and provide further clarification regarding the amount of training data required for our framework to perform effectively.
>
> To begin with, we have made the following additions and modifications with respect to that:
>
> - **Centralized experiments (DeepQP, Section 6.1).**  We have included an additional experimental analysis in Section J.1 of the Appendix which compares the performance of DeepQP using 500, 1000 and 2000 problems as training data. The provided results (Fig. 8) highlight the robust performance of DeepQP even with a limited amount of training data. As a result of this analysis, we have also replaced our previous experiments in Section 6.1 (Fig. 5) with using a training dataset size of 500 for all problems, except for the random QPs where we have maintained 2000 to present optimal performance.
>
> - **Distributed experiments (DeepDistributedQP, Section 6.2).** After observing a similar pattern when training DeepDistributedQP, we have reduced the amount of training data used for the experiments in Section 6.2 (Fig. 6) from 2000 samples to 500. The results demonstrate that the performance of DeepDistributedQP remains robust and highly effective, even with this reduced dataset. We retained 1000 training samples only for the first out of the six classes of problems (random QPs without equality constraints), as slightly better performance was observed with more data. For completeness, we have also included an additional figure (Fig. 9) in Section J.1 of the Appendix comparing performance with 500 and 1000 training samples for this problem, showing that even when using 500 training samples, the performance of DeepDistributedQP still remained significantly superior to the one of the traditional optimizer.
>
> **Consequently, the proposed methodology demonstrates high effectiveness and robustness, even when trained with a limited amount of data.**
>
> Furthermore, we wish to highlight the following quite important point. In the updated experiments of Section 6.2, training DeepDistributedQP takes place on much smaller problems than the ones, the framework successfully scales for during evaluation. For convenience, we also repeat Table 1 here:
>
>
> | Problem Class            | N (train)   | n (train)    | m (train)    | nnz(Q, A) (train)  | N (test) | n (test) | m (test) | nnz(Q, A) (test) |
> |--------------------------|-----|------|-------|-------------|---------------|---------------|---------------|-----------------------|
> | Random QPs              | 16  | 160  | 120   | 4,000       | 1,024         | 10,240        | 9,920         | 300,800              |
> | Random QPs w/ Eq. Constr.| 16  | 160  | 168   | 4,960       | 1,024         | 10,240        | 9,920         | 300,800              |
> | Coupled Pendulums        | 10  | 470  | 640   | 3,690       | 1,000         | 47,000        | 64,000        | 380,880              |
> | Coupled Osc. Masses      | 10  | 470  | 1,580 | 4,590       | 300           | 28,200        | 47,400        | 141,180              |
> | Network Flow             | 20  | 100  | 140   | 600         | 5,000         | 25,000        | 35,000        | 150,000              |
> | Distributed LASSO        | 10  | 1,100| 3,000 | 29,000      | 500           | 50,100        | 150,000       | 1,450,000            |

---

> ### Author Response · Authors · 2024-11-28
> **Response to Reviewer fvD1 (part 2/4)**
>
> In addition, we include a portion of the newly added Table 6 from the Appendix, which presents the total wall-clock time required by DeepDistributedQP, DistributedQP, and OSQP (using either direct and indirect method) to achieve the same accuracy, e.g. for the network flow problem:
>
> | N    | n      | m      | nnz(Q, A)   | DeepDistributedQP (Ours)        | DistributedQP (Ours)        | OSQP (indirect)      | OSQP (direct, 1st iter.) |
> |------|--------|--------|-------------|-------------|-------------|------------|------------------|
> | 20   | 100    | 140    | 600         | 6.80 ms     | 10.68 ms    | 9.51 ms    | **0.59 ms**      |
> | 50   | 250    | 350    | 1,500       | 7.81 ms     | 13.17 ms    | 14.81 ms   | **1.30 ms**      |
> | 200  | 1,000  | 1,400  | 6,000       | **12.08 ms**| 17.61 ms    | 208.19 ms  | 61.93 ms         |
> | 500  | 2,500  | 3,500  | 15,000      | **13.63 ms**| 19.73 ms    | 425.7 ms   | 745.2 ms         |
> | 1,000| 5,000  | 7,000  | 30,000      | **20.51 ms**| 31.62 ms    | 8.73 s     | 11.59 s          |
> | 2,000| 10,000 | 14,000 | 60,000      | **29.86 ms**| 47.22 ms    | 51.6 s     | 73.9 s           |
> | 5,000| 25,000 | 35,000 | 150,000     | **61.23 ms**| 85.99 ms    | 558 s      | *Out of memory*  |
>
> Note that the computational cost of constructing 500 sample problems with N = 20 (using for example OSQP with the indirect method) would be
>
> 500 x 9.51 ms =  4.755 s.
>
> As result, constructing the training dataset is extremely cheap computationally.
>
> Nevertheless, solving the high-dimensional problems our framework can tackle with OSQP instead (for example N = 5000) would either be extremely slow (558 s using OSQP indirect just to solve one problem) or impossible due to memory overflow issues.
>
> **As a result, we wish to also emphasize that constructing the training data for our setup is an extremely cheap process, and in addition DeepDistributedQP is able to effectively tackle much higher-dimensional problems than the low-dimensional ones used for training.**
>
> Finally, in case the reviewer refers to the dataset size of 15,000 samples used for obtaining the generalization bounds, using a high amount of data is a prerequisite for establishing meaningful PAC-Bayes bounds. See for example the frameworks in [R1] and [R2] that are using training datasets of size up to 10,000 and 50,000 respectively. Nevertheless, this amount of data is only required if one is interested in establishing the PAC-Bayes bounds.
>
> **Overall, our framework exhibits very effective performance in practice on tackling high-dimensional problems, while being trained with limited and small-scale problems. This fact highlights the efficiency of DeepDistributedQP in terms of training.**
>
> ## Response to Questions
>
> 1. Please see our response to the related weakness. The amount of data used for all experiments are detailed in Tables 2 and 3. As noted in Section I.3 in the Appendix, the training data are always obtained through solving sample problems with OSQP – as their dimension is low. For the testing data, beyond some scale, we use DistributedQP to construct the data - as it is much faster than OSQP (see Table 6) -  after verifying that the obtained solution is satisfying optimality conditions. This fact also highlights the importance of having established the convergence of DistributedQP to optimality earlier in Theorem 1 of Section 3.3.
>
> 2. We thank the reviewer for this really insightful question. The models we are training are specific for each problem class. Here, we wish to first emphasize that the primary focus of the learning-to-optimize field has been so far to learn optimizers that tailored for a specific problem (in terms of both type and dimension) [R3].
>
>    Consequently, we wish to first highlight the following: **The proposed DeepDistributedQP framework goes beyond this assumption as it is trained on small-scale problems and then successfully applied on much larger ones (see Tables 1 and 6). This already constitutes a really significant step towards generalizing learning-to-optimize methods for addressing large-scale optimization problems without requiring to be trained on them.**
>
> [R1] Majumdar, A., Farid, A., & Sonar, A. (2021). PAC-Bayes control: learning policies that provably generalize to novel environments. The International Journal of Robotics Research, 40(2-3), 574-593.
>
> [R2] Sambharya, R., & Stellato, B. (2024). Data-Driven Performance Guarantees for Classical and Learned Optimizers. arXiv preprint arXiv:2404.13831.
>
> [R3] Shlezinger, N., Eldar, Y. C., & Boyd, S. P. (2022). Model-based deep learning: On the intersection of deep learning and optimization. IEEE Access, 10, 115384-115398.

---

> ### Author Response · Authors · 2024-11-28
> **Response to Reviewer fvD1 (part 3/4)**
>
> 2. (Continued) Regarding testing policies on different problems in terms of both type and dimension, we have included two new experiments in Section J.2 of the Appendix. In the first experiment, we have trained a policy on equality-constrained random QPs and applied it on random QPs without equality constraints. We observe that the resulting performance is quite strong, as there is a degree of similarity in the structure of these problems. In the second experiment, we apply the same policy on a coupled pendulums problem where we observe that the performance becomes suboptimal. Overall, as expected, performance will remain strong as long as there is a degree of similarity between the classes used for training and testing. If these two classes are completely irrelevant, then it is expected that performance will decrease.
>
>      In future work, we wish to consider training our framework on diverse data so that we can arrive to models that better generalize for a broader range of problems in practice.
>
> 3. We thank the reviewer for this very interesting question as well. Regarding the PAC-Bayes generalization bound, the presented bound only applies to problems from the same distribution as the training problems. There are some techniques in the literature to adapt PAC-Bayes bounds to quantify performance on a test distribution different from the distribution generating the training set [R4], but all such techniques result in a new term in the bound which is capturing the distance between the two distributions. In practical applications, it is not clear how one would evaluate this term without having significant information about the test distribution. As such, PAC-Bayes bounds are typically not employed in this fashion.
>
>  4. We thank once again the reviewer for another insightful question. Indeed, this is a matter of important interest, if a user for example needs to run such an approach for long iterations.
>
>      First of all, we wish to underline the following. It is very often the case in many applications, that the computational budget might be restricted. In other words, it might be the case it is not possible to exceed a specific predefined number of iterations. This is exactly the setup deep unfolding (and hence our method as well) addresses as it aims in optimizing the performance of the solver under a limited amount of iterations.
>
>     Nevertheless, we agree that there might be cases where we need to run an optimizer for longer iterations. To cover this case, we have included additional experiments in Section J.3 of the Appendix where we repeated the last layer of DeepDistributedQP for extra iterations on the studied distributed problems. As shown by the results, the learned optimizer maintains its superior performance against the standard optimizer for all problems when using closed-loop policies and for 4 out of 6 problems when using open-loop ones. This shows that this straightforward adaptation can work sufficiently well in practical scenarios where extra iterations are required. In future work, we wish to also consider repeating the same layer over the last iterations during training, so that we further robustify performance for such cases.
>
>     In terms of training efficiency, the number of layers directly affects the required memory and time for training. Yet, we wish to highlight that our approach is quite memory and time efficient thanks to employing implicit differentiation as explained in Section 4.2 and training on small-scale problems.
>
> 5. We are again thankful for this thoughtful question. We would argue that "reaching the optimum" is equivalent in practice with "reaching an acceptable optimality gap" where of course the value of the latter depends on the application. As a result, if a user requires a more accurate solution and the available computational budget allows for additional iterations, then as explained in our response to question 4, an effective adaptation would be to simply repeat the last layer/iteration of the optimizer until the desired accuracy is achieved.
>
> [R4] Germain, P., Habrard, A., Laviolette, F., & Morvant, E. (2020). PAC-Bayes and domain adaptation. Neurocomputing, 379, 379-397.

---

> ### Author Response · Authors · 2024-11-28
> **Response to Reviewer fvD1 (part 4/4)**
>
> Regarding the minor points raised by the reviewer:
>
> 1. We thank the reviewer for noticing. We have included a detailed description for these terms in Section D of the Appendix.
>
> 2. We are thankful to the reviewer for asking for further clarification on the mapping between the local and global variables. We have included additional details and an extra figure (Fig. 2) in Section 3.1 which explain how this mapping works. We have adopted this formulation from the exposition of consensus ADMM in the highly influential tutorial on ADMM from Boyd et al [R5]. This formulation is particularly convenient as it does not explicitly assume a network structure in problems. We hope that the reviewer finds the added explanation valuable, so that the reader better understands how this mapping works.
>
>  3. Thank you very much for noticing. We have fixed this typo.
>
> Overall, we have found your comments and suggestions extremely helpful for better explaining fundamental parts of our work and highlighting its applicability and potential extensions. Given that we have significantly improved our manuscript in its revised version and that we have addressed all your points, we would be truly grateful if you considered re-evaluating our paper and recommending it for acceptance. We also remain at your disposal for any additional questions you might have.
>
> [R5] Boyd, S., Parikh, N., Chu, E., Peleato, B., & Eckstein, J. (2011). Distributed optimization and statistical learning via the alternating direction method of multipliers. Foundations and Trends® in Machine learning, 3(1), 1-122.

---

> > ### Author Response · Authors · 2024-11-30
> > **Follow-up to Reviewer fvD1**
> >
> > As we are approaching the end of the discussion period, we would appreciate the reviewer to clarify if our responses have sufficiently addressed their raised concern about the required amount of training data, as well as the rest of their questions. If so, and given the rest of the significant improvements in the revised version of our paper (better clarifying motivation and novelty, scaling our framework to much higher-dimensional problems with up to 50K variables and 150K constraints), we kindly request the reviewer to reconsider their rating.

---

> > > ### Comment · Reviewer_fvD1 · 2024-12-01
> > >
> > > Thank you for providing these new results. The questions that remain for the reviewer are:
> > > 1. How does learning on, for example, a NetworkFlow problem, and then applying to distributed lasso work. Furthermore, for a network flow problem, how does learning work for different topologies and problem structures within that class of problems?
> > >
> > > 2. The reviewer finds the timing analysis interesting. It seems as though the, from a complexity standpoint, the proposed method would introduce additional computation as its adds the neural network layer on top of the regular OSQP problem. However, it seems that the proposed method is actually quicker. Can the authors comment what is the main cause of the reduction of the wall-clock time.
> > >
> > > 3. Also in the rebuttal, the authors claim that "Note that the computational cost of constructing 500 sample problems with N = 20 (using for example OSQP with the indirect method) would be 500 x 9.51 ms = 4.755 s. As result, constructing the training dataset is extremely cheap computationally." However, this is magnitudes greater than the benefit timing benefit of their proposed model.  In future works, it might be interesting to study how the benefit evolves as similar problems (e.g., Network flow) are performed over time
> > >
> > > 4. When the authors scale the problem in their results to demonstrate a reduced training set (Tables 1 and 6), is the problem structure changed at all? More specifically, do the number of variables simply increase while the problem structure is repeated, or does scaling the problem also introduce new topologies of the problem?
> > >
> > > Thank you for your new results. These new results have improved the paper quality paper, and as a result, the reviewer has increased the score.

---

> > > > ### Author Response · Authors · 2024-12-02
> > > > **Response to Reviewer's fvD1 follow-up comment (part 1/2)**
> > > >
> > > > Thank you so much for recognizing the importance of our work and for raising your evaluation score from 5 to 6. We sincerely appreciate the thoroughness of your comments and the valuable insights you have been providing, which have significantly contributed to improving the quality of this paper.
> > > >
> > > > Below, we provide our responses to your additional questions:
> > > >
> > > > 1. We are really thankful for this insightful question. When the policies defined in Eq. (10) are *shared*, all nodes learn the same policy. This approach makes the policies transferable across different networks with varying numbers of nodes or even across entirely different problems, as these policies are based solely on the local primal and dual residuals and therefore remain agnostic to the underlying problem structure. On the other hand, if these policies are *local*, then each node learns its unique policy. While this approach is likely to enhance performance (as shown in the shared vs local policies comparison in Section 6.2), it is inherently limited to a specific network. Consequently, using a shared policy provides a significant advantage in versatility, enabling its training and application on diverse networks and problem classes. We hope this explanation addresses both scenarios you are inquiring about. Please let us know if you require any further clarification.
> > > >
> > > > 2. Thank you very much for requesting this clarification. As emphasized in our response to Reviewer EEUg and also explicitly stated in the revised version of our paper, DeepDistributedQP and DistributedQP share the *same per-iteration complexity*. This is due to the fact that their most computationally expensive operation is solving Eq. (11) using the indirect method (with conjugate gradient). The only additional computation introduced by DeepDistributedQP is the forward propagation through the feedback MLPs of size 2x16. However, these operations are extremely cheap matrix multiplications and element-wise operations - that are also highly parallelizable. As a result, their computational cost is in practice negligible compared to solving Eq. (11). Therefore, the extra time required by DistributedQP to achieve the same accuracy as DeepDistributedQP is simply attributed to its extra iterations (of equal complexity). This observation can be further verified by noticing that the ratio of their iterations closely aligns with the ratio of their wall-clock times.
> > > >
> > > > 3. We are really thankful for requesting this clarification and we apologize if our original response was not sufficiently clear. It is important to note that the dataset construction and training processes are taking place *offline*. Typically, there is some tolerance on the time such processes can take since they only happen once, before deploying the model on time-critical applications. On the other hand, deploying a distributed learned optimizer on new large-scale optimization problems is an *online* process, which is really important to accelerate. This is because the optimizer is expected to be used over and over again on applications with strict restrictions on the allowed computational time/iterations. As a result, comparing these two times is not an "apples-to-apples" comparison as reducing the computational time during the online deployment of the solver is of much greater importance. Consequently, what we intended to convey with this comment was that DeepDistributedQP is not only highly efficient in terms of training requirements - both in dataset size and the small scale of the training problems - but that the cost of constructing this dataset is also extremely low. At the same time, the benefits realized during the online deployment of the solver when applied repeatedly to targeted applications will far outweigh the modest computational effort spent on offline dataset construction and training. These benefits can be interpreted either as computational savings or as improvements in the solution quality the solver provides within the context of each application. We hope that this clarification sufficiently addresses your question. Please let us know if further elaboration is needed.

---

> > > > > ### Author Response · Authors · 2024-12-02
> > > > > **Response to Reviewer's fvD1 follow-up comment (part 2/2)**
> > > > >
> > > > > 4. Thank you once again for another important question. Scaling the problem dimension refers to increasing the number of nodes, which as a result leads to changes in the network topology. The number of variables and constraints in the local problems remain the same as we increase the number of nodes.
> > > > >
> > > > > We wish to sincerely thank you once again for your high-quality reviewing and overall commitment to this process. Your constructive comments and insightful questions have been really valuable in identifying key parts of the paper that required greater emphasis and clarity for the reader. Unfortunately, we need to also acknowledge with gratitude that despite today being the last day of the author-reviewer discussion phase, you remain the only reviewer that has responded to our rebuttal and revisions. Consequently, we are especially thankful for your time and for recognizing the critical importance and potential impact of this work in developing scalable and effective distributed learned optimizers for large-scale quadratic programming - one of the most significant classes of optimization problems in modern applications. We hope that our new response is able to further strengthen your recommendation for accepting our paper.

---

### Author Response · Authors · 2024-11-30
**General Message to Reviewers and Area Chair after Revision (part 1/2)**

We sincerely thank all three reviewers for their time and effort in thoroughly reviewing our work. We deeply appreciate their constructive feedback and valuable suggestions, which have significantly contributed to improving the quality of our paper. In the revised version, we have carefully considered the suggestions of the reviewers on expressing the motivation for addressing large-scale QP problems and the novelty of the proposed DeepDistributedQP framework more clearly. Additionally, we have addressed the concerns of the reviewers regarding the scalability of our framework by significantly increasing the dimension of the illustrated problems to ones with up to 50K variables and 150K variables and showing the orders-of-magnitude improvements in wall-clock time that our approach offers compared to conventional solvers such as the state-of-the-art OSQP method.

We provide a summary of our revisions and responses to the main concerns expressed by the reviewers below:

- **Motivation (Reviewer EEUg).** We agree with Reviewer EEUg that the initial version of our paper did not sufficiently emphasize the critical importance of developing effective methods for tackling large-scale QP problems. To address this concern, we have revisited the introduction section so that we more clearly articulate why it is vital to develop scalable and effective methods for solving such problems. In summary, QP serves as a fundamental cornerstone in optimization, with applications spanning a wide range of fields. On top of that, it also frequently serves as a subroutine in many advanced non-convex optimization methods. Hence, as the scale of modern applications is continuously increasing, the development of scalable and effective methods for addressing large-scale QP problems is becoming increasingly urgent. We hope that the reviewer recognizes the importance of the problem this work is addressing.

- **Novelty (Reviewer Y4sR and EEUg).** We thank the reviewers for encouraging us to explain more clearly the novelty of our work.

   -   *Similarity with RLQP (Reviewer Y4sR)*. We appreciate reviewer’s Y4sR feedback regarding the need to better clarify any similarity between our work and RLQP. In our response to the reviewer, we have emphasized the minimal connection between the proposed DeepDistributedQP and RLQP by clearly explaining the distinct differences between these approaches. We have also more clearly expressed these fundamental differences in the related work section to ensure that any such unnecessary confusion is avoided. Again, we are really thankful to the reviewer for pointing out that these differences were not clearly stated in the original version of our paper.

    - *General novelty (Reviewer EEUg)*. We are really thankful to reviewer EEUg for their feedback to place more emphasis on explaining the novelty of our work. We have addressed this concerns through both revisiting the statement of contributions in the revised version of our paper and in our individual response to the reviewer. In summary, the novel aspects of this work are the following:

         1) We introduce DistributedQP, a ***novel decentralized optimization method for tackling large-scale QP problems.***
         2) We propose DeepDistributedQP, a ***novel deep learning-aided distributed architecture for large-scale QP, which relies on unfolding DistributedQP as supervised learning framework and learning policies for the algorithm parameters***. To our best knowledge, this is the ***first work to present a learning-to-optimize method for distributed QP***. In addition, despite the widespread applicability of ADMM for deriving distributed methods, this is also the ***first work to present a deep unfolded framework of distributed ADMM for large-scale constrained optimization*** towards addressing some of its main shortcomings such as its need for rigorous tuning, lack of generalization capabilities and potential computational/communication limitations.
         3) We establish ***novel performance guarantees for learned optimizers on unseen problems through PAC-Bayes theory.***
         4) Our framework can ***effectively handle large-scale QP problems with up to 50K variables and 150K constraints, while being trained exclusively on much lower-dimensional ones.*** To our best knowledge, ***this is the first work to present a distributed learning-to-optimize method with such generalization capabilities.***  Hence, these results constitute a key step towards the future development of effective distributed learning-to-optimize methods that can be trained on low-dimensional problems and be applied on much higher-dimensional ones.

       We truly hope that the reviewer is able to appreciate the novel aspects and importance of the results of this work.

---

> ### Author Response · Authors · 2024-11-30
> **General Message to Reviewers and Area Chair after Revision (part 2/2)**
>
> - **Scaling to high-dimensional problems (Reviewer Y4sR and EEUg).**  We sincerely thank both reviewers for encouraging us to further expand the scale of the addressed problems compared to our initial submission. In our updated results, we have placed much greater emphasis on demonstrating the scalability of our framework. In particular, we show that DeepDistributedQP can effectively address problems with up to 50K variables and 150K constraints, while being trained exclusively on much lower-dimensional ones (Fig. 6 and Tables 1, 6). We have also added a total wall-clock time comparison with the state-of-the-art OSQP method which highlights the orders-of-magnitude improvements DeepDistributedQP offers computationally (Fig. 1, 6 and Table 6). We hope that the reviewers appreciate the scale of the newly presented problems and recognize the ability of our framework to scale for high-dimensional problems.
>
> - **Required amount of training data (Reviewer fvD1).** We have provided an additional experimental analysis which highlights that our framework only requires a limited amount of training data to be effective. In addition, we have emphasized that DeepDistributedQP can effectively handle large-scale problems, while being trained exclusively on much lower-dimensional ones. Consequently, DeepDistributedQP is characterized by significant training efficiency both in terms of the amount and the size of the problems required for training.
>
> Overall, the feedback from all three reviewers has been invaluable in highlighting aspects of our work that required further emphasis. In the revised version of our paper, as well as in our individual responses, we have thoroughly addressed all concerns raised by the reviewers and have incorporated their suggestions for improvements. We believe that this work marks a foundational step in advancing distributed learning-to-optimize methods for large-scale constrained optimization and overcoming the current shortcomings of standard (non-learned) distributed optimization methods. Consequently, we sincerely hope that both the reviewers and the area chair recognize the importance of this contribution and recommend our paper for acceptance. We remain at the disposal of the reviewers and the area chair for any additional clarifications.

---

### Author Response · Authors · 2024-12-03
**Follow-up Message to Reviewers and Area Chair - Last Day of Author-Reviewer Discussion Phase**

As today is the last day of the discussion period, we kindly request the remaining reviewers (Reviewers Y4sR and EEUg) to clarify whether their concerns have been addressed. We would also like to express our gratitude to Reviewer fvD1 for recognizing the significance of our work and raising their score.

For convenience, we provide a summary of the raised concerns from Reviewers Y4sR and EEUg, as well as how we have addressed them through the revised version of our paper and responses:

-  **Reviewer Y4sR**: While the primary concern of the reviewer was a potential similarity between our work and RLQP, we have clearly emphasized the minimal connection between RLQP and the proposed DeepDistributedQP framework. Furthermore, we have addressed the concerns of the reviewer regarding scaling to high-dimensional problems, through our new experiments which showcase that DeepDistributedQP can effectively address problems with up to 50K variables and 150K constraints, while being trained exclusively on much lower-dimensional ones. In addition, we have illustrated the effectiveness of our framework under a varying number of layers per request of the reviewer. Given that we have fully addressed all the concerns of the reviewer, we would greatly appreciate the reconsideration of their rating.
- **Reviewer EEUg**: The primary concern of the reviewer was on the motivation and novelty of this work. We hope that the revised version of the introduction of our paper as well as our response have revealed both the vital importance of developing effective methods for large-scale quadratic programming (QP) problems, as well as the significant and multi-faceted novelty of our work.

   - In terms of motivation, QP serves as a fundamental cornerstone of optimization, appearing in a wide array of applications and frequently acting as a subroutine in many advanced optimization methods. As the scale and complexity of modern applications continue to grow, it is urgent to develop effective methods for solving large-scale QP problems efficiently.

   - In terms of novelty, the contributions of this work include the following:
       1. We introduce DistributedQP, a novel decentralized optimization method for tackling large-scale QP problems.
        2. We propose DeepDistributedQP, a novel deep learning-aided distributed architecture for large-scale QP, which relies on unfolding DistributedQP. To our best knowledge, this is the first work to present a learning-to-optimize method for distributed QP. In addition, despite the widespread applicability of ADMM for deriving distributed methods, this is also the first work to present a deep unfolded framework of distributed ADMM for large-scale constrained optimization towards addressing some of its main shortcomings.
       3. We establish novel performance guarantees for learned optimizers on unseen problems through PAC-Bayes theory.
       4. Our framework can effectively handle large-scale QP problems with up to 50K variables and 150K constraints, while being trained exclusively on much lower-dimensional ones. To our best knowledge, this is the first work to present a distributed learning-to-optimize method with such generalization capabilities.

       We sincerely hope that this exposition effectively conveys the importance of the motivation and the multi-faceted novelty of this work, and is sufficient to address the reviewer’s concerns.

Overall, we truly appreciate the insightful comments of all reviewers, as they have highlighted parts of our work that required further emphasis. As a result, their comments have significantly contributed to improving the clarity and quality of our paper.

If Reviewers Y4sR and EEUg agree that we have fully addressed their concerns, we would greatly appreciate the reconsideration of their rating to reflect these improvements. We also remain at the disposal of the reviewers and the area chair for any additional clarifications.

---

### Meta-Review · Area_Chair_bzcg · 2024-12-21

**Metareview:**

This paper introduces a novel framework, DeepDistributedQP, which combines distributed optimization techniques with deep learning to address large-scale Quadratic Programming (QP) problems. The key strength of this work lies in its innovative integration of the Operator Splitting QP (OSQP) method into a distributed architecture (DistributedQP) and subsequently unfolding this method as a deep learning framework (DeepDistributedQP). The proposed approach demonstrates significant improvements in scalability, achieving solutions for problems with up to 50K variables and 150K constraints, while maintaining rigorous convergence guarantees. Experimental results highlight the practicality and computational efficiency of the method, achieving orders-of-magnitude improvements in wall-clock time over traditional solvers like OSQP.
The primary contributions of the paper are multi-faceted and impactful: (1) it pioneers a learning-to-optimize approach tailored for distributed constrained optimization, (2) it introduces a novel methodology for tackling large-scale QP problems using limited training data, and (3) it extends the applicability of distributed optimization to high-dimensional problems with significant computational savings.
The paper makes valuable theoretical and practical contributions to the optimization and machine learning community.

**Additional Comments On Reviewer Discussion:**

The discussion phase helped to improve the paper by addressing important points raised by reviewers. The authors worked hard to reply to concerns about scalability and novelty by including larger-scale experiments and better explaining how their work is different from other methods, like RLQP. Even though the authors made these changes, some reviewers still felt that the paper might fit better in an optimization-focused venue because of its focus on classical optimization problems. However, the authors showed how their method could be useful for bigger problems and efficient computation, which are also important for machine learning. The discussions between authors and reviewers were helpful and made the paper clearer and more complete.

---

### Decision · Program_Chairs · 2025-01-22

Accept (Poster)